# Mapping the fracture network in the Lilstock pavement, Bristol Channel, UK: manual versus automatic

Christopher Weismüller[1], Rahul Prabhakaran[2,3], Martijn Passchier[4], Janos L. Urai[4], Giovanni Bertotti[2], Klaus Reicherter[1]

[1] Neotectonics and Natural Hazards, RWTH Aachen University, Aachen, Germany
[2] Department of Geoscience and Engineering, Delft University of Technology, Delft, the Netherlands
[3] Department of Mechanical Engineering, Eindhoven University of Technology, the Netherlands
[4] Structural Geology, Tectonics and Geomechanics, RWTH Aachen University, Aachen, Germany

*Correspondence to* Christopher Weismüller (c.weismueller@nug.rwth-aachen.de)

**Abstract.** The 100,000 m² wave-cut pavement in the Bristol Channel near Lilstock, UK, is a world-class outcrop, perfectly exposing a very large fracture network in several thin limestone layers. We present an analysis based on manual interpretation of fracture generations in selected domains and compare it with automated fracture tracing. Our dataset of high-resolution aerial photographs of the complete outcrop was acquired by unmanned aerial vehicle, using a survey altitude optimized to resolve all fractures. We map fractures and identify fracture generations based on abutting and overprinting criteria and present the fracture networks of five selected representative domains. Each domain is also mapped automatically using ridge detection based on the complex shearlet transform method. The automatic fracture detection technique provides results close to the manually traced fracture networks in shorter time, however, with a bias towards closely spaced Y over X nodes. The assignment of fractures into generations cannot yet be done automatically because the fracture traces extracted by the automatic method are segmented at the nodes, unlike the manual interpretation in which fractures are traced as a path from fracture tip to tip consisting of several connected segments. This segmentation makes an interpretation of relative age impossible because the identification of correct abutting relationships requires the investigation of the complete fracture trace following a clearly defined set of rules. Generations one and two are long fractures that traverse all domains. Generation three is only present in the southwestern domains. Generation four follows an ENE-WSW striking trend, is sub-orthogonal to generations one and two and abuts on them and generation 3, if present. Generations five is the youngest fracture set with a range of orientations, creating polygonal patterns by abutting at all other fracture generations. Our mapping results show that the northeastern domains only contain four fracture generations, thus the five generations of the outcrop identified in the southwestern domains are either not all present in each of the five domains or vary locally in their geometry, preventing the interpreter to link the fractures to their respective generation over several spatially separate mapping domains. Fracture intensities differ between domains where the smallest is in the NE with 7.3 m/m² and greatest in the SW with 10 m/m², coinciding with different fracture orientations, and distributions of abutting relationships. Each domain has slightly different fracture network characteristics and greater connectivity occurs where the development of later, shorter fractures is not affected by the stress shadowing of pre-existing, longer fractures.

## 1. Introduction

Recent technological advances allow us to collect large amounts of remote sensing and outcrop data, e.g. using LiDAR, unmanned aerial vehicles (UAV), and structure from motion (SfM) tools (Bemis et al., 2014; Bisdom et al., 2017; Hansman and Ring, 2019; Müller et al., 2017; Niethammer et al., 2012; Vasuki et al., 2014; Weismüller

et al., 2019; Westoby et al., 2012). These high-resolution datasets can be acquired in short periods of time, optimizing resolution, and enabling the detailed digitalization of km-scale outcrops in sub-cm resolution. The mapping and interpretation of features in the datasets is time-consuming because these very large areas must be mostly interpreted manually. Automated tools can aid during the interpretation, but do not yet match the quality and reliability of manual interpretations, and thus must be used with care (Duelis Viana et al., 2016; Vasuki et al., 2014).

Several sampling techniques have been developed to extract data about fracture sets that do not necessarily require a complete mapping of the whole area, e.g. line sampling (Priest and Hudson, 1981), polygon or areal sampling (Wu and D. Pollard, 1995), circular scanline sampling (Mauldon et al., 2001; Rohrbaugh et al., 2002; Watkins et al., 2015) or rectangular window sampling (Pahl, 1981).

In this work, we use a UAV to collect aerial images of the fractured limestone at the Lilstock coast in the Bristol Channel, UK. Sample pictures and tests from varying height in combination with ground-truthing show that we can resolve all visible mode I barren fractures in the pavement, which are the target of this study. With the right compromise between flight altitude and resolution, we photographed the whole outcrop of 400x150 m within two days with similar lighting conditions, creating a set of raw images depicting a fracture network which is too large to be manually interpreted. Our aim in this study is to provide a complete interpretation and fracture network analysis of the selected sub-areas of the outcrop, providing a benchmark of our analyses. We hand-interpret several domains and use them as basis for supervised automatic tracing to create a large fracture dataset. We present our interpretation of several fracture generations in the selected areas to review the evolution of the fracture network in time steps, using topological branch and node analysis (e.g. Dimmen et al., 2017; Morley and Nixon, 2016; Nyberg et al., 2018; Procter and Sanderson, 2018; Sanderson and Nixon, 2015). By comparing these areas, we give further estimates about the spatial variation within the larger fracture network. In a companion paper (Passchier et al., in prep) we present a manual fracture mapping study of the whole area, discuss the criteria for identifying different fracture generations, and study the spatial heterogeneity of different fracture generations.

## 2. Study area

### 2.1 Geology

The study site is a wave-cut platform on the southern coast of the Bristol Channel in Somerset (Fig. 1a), close to the hamlet Lilstock. The outcrop exposes joints, faults, and fractures in a lower Jurassic sedimentary sequence of bituminous shale, marl, and limestone (Fig. 1b). The site is renowned among structural geologists and the subject of various publications studying faults, fractures, fracture relationships, or basin inversion (Crider and Peacock, 2004; Dart et al., 1995; Gillespie et al., 2011; Glen et al., 2005; Peacock et al., 2018; Peacock and Sanderson, 1991; Procter and Sanderson, 2018; Rawnsley et al., 1998). We focus on a single fractured limestone layer that has been previously referred to as "the bench" (Loosveld and Franssen, 1992) or "Block 5" (Engelder and Peacock, 2001).

Several tectonic phases are inferred from the structures found in the Bristol Channel basin, starting with two events of N-S extension, in the Jurassic to Early Cretaceous and Late Cretaceous to Oligocene, followed by Alpine N-S contraction from the late Oligocene to Miocene and the progressive relaxation during the Late or post-Miocene (Rawnsley et al., 1998). These events are recorded as conjugate E-W striking normal faults (Brooks et al., 1988) caused by the extension and conjugate strike slip faults along with inverted normal faults, which resulted from the

subsequent compression (Dart et al., 1995; Glen et al., 2005; Kelly et al., 1999; Nemčok et al., 1995). This study focuses on mode I joints in the limestone layers, which have been studied by Agheshlui et al. (2018), Azizmohammadi and Matthäi (2017), Belayneh et al. (2006a, 2006b), Belayneh and Cosgrove (2004), Engelder and Peacock (2001), Gillespie et al. (2011), Loosveld and Franssen (1992), Matthäi et al. (2007), Matthäi and Belayneh (2004), and Peacock and Sanderson (1991, 1994, 1995). According to Engelder and Peacock (2001), minor tectonic events post-date the inversion and resulted in the jointing of the limestone: the first joint set of the Bristol Channel was possibly caused by a stress field consistent with second Alpine event during the late Oligocene to Miocene (Rawnsley et al., 1998). A second phase of jointing that follows a propagation sequence consistent with an anticlockwise shift (NW-SE to NE-SW) of the regional maximum horizontal stress (Loosveld and Franssen, 1992) and a period of E-W compression (Hancock, 1969). Unfilled joints are caused by basin-wide relaxation of the Alpine compression (Rawnsley et al., 1998). Engelder and Peacock (2001) further point out that the youngest joint set correlates with the contemporary tectonic stress field (Bevan and Hancock, 1986; Hancock and Engelder, 1989), and that the youngest NW-striking joint sets in NW Europe could be caused by exhumation in a late stage Alpine stress field (Hancock and Engelder, 1989).

## 2.2 Data set

We selected five 140 m² large rectangular domains (Fig. 1b) to be mapped and interpreted, both manually and automatically, from the generated ortho-mosaics that have a spatial resolution of ~ 1 cm/pixel. Three domains are located on the south-western part of "the bench", referred to as SW1, SW2, and SW3 (from W to E), and domains on the north-eastern part, referred to as NE1 and NE 2 (from NW to SE) (Fig.1b). The reasons for selecting these specific regions are:

i. all domains are within a single fractured limestone layer "the bench", allowing us to compare the spatial variation of the fracture network within this single layer,

ii. fractures in "the bench" have erosion-enhanced aperture and hence are easily detected in UAV-photographs because of the high contrast between fracture and rock,

iii. "the bench" offers the largest exposure of a sub-horizontal single layer,

iv. a qualitative review indicated that the fracture network complexity changes over short distances in the southwestern part of "the bench",

v. our domains NE1 and NE2 were chosen to partly overlap with the area studied in Gillespie et al. (2011), allowing a comparison of the results,

vi. in the area overlapping with the one of Gillespie et al. (2011) a qualitative review reveals a radial pattern of fractures that converge towards the fault (see Fig. 1b) S of the domain, which may impact the local fracture network geometry and topology,

vii. areas with preferably little erosion of the surface of the pavement help to reduce the number of false positives during the automatic extraction,

viii. areas with no obvious voids in the network which may be caused accumulations of remaining seaweed covering the fractures or missing parts of the pavement due to erosion.

The size of the domains (140 m² each) was chosen to be as large as possible, to prevent over-censoring of longer
fractures, while still allow manual mapping in an appropriate amount of time.

## 3.  Methods

### 3.1  UAV

We used a DJI Phantom 4 UAV with a 12 MP camera to capture several sets of overlapping photographs of the
pavement from an orthogonal perspective to create ortho-rectified mosaics. These mosaics were used as a raster
layer for detailed mapping of the fracture network. To optimize the quality of the raw image data, the aerial
photographs were acquired on days with suitable weather conditions: low wind velocities, no rain, sunny
conditions with almost clear skies and warm temperature, that lead to the complete evaporation of the seawater on
the surface, but not in the cavities of the fractures, further increasing the contrast between fracture and matrix.
Because the near-flat topography does not cast shadows, sunny conditions in combination with the contrast due to
the wet fractures are superior to the typically recommended cloudy skies that usually result in a better image quality
as diffuse light casts fewer shadows with lower dynamic range. The setting along the coast of the Bristol Channel
necessitates data acquisition at low tide, since most of the pavement is covered by the sea during high tide and the
outcrop is washed clean but dries quickly on a clear, warm day. To optimize the volume of data acquisition during
a single day, we undertook as many flights as possible during low tide in the morning and again in the afternoon.
Apart from the cliff, the coast is flat without major obstacles, enabling us to automate the UAV mapping procedure
by using Pix4Dcapture to pre-generate flight routes for the UAV. The automation provides better control on the
degree of overlap between each photograph. We used settings of at least 80% frontlap and 60% sidelap with
vertically downwards facing camera, resulting in 8-9 perspectives per point in the center of our models that are
adequate to create a 2D map of a low topography surface. We used the SfM photogrammetry software Agisoft
PhotoScan to match the photographs, calculate camera perspectives, and create dense point clouds. Using the point
clouds, digital elevation models (DEMs) were calculated achieving resolutions < 2.3 cm/pixel for flight altitudes
below 25 m above ground. These DEMs were used as reference surfaces for the calculation of the ortho-mosaics
with resolutions ~ 1 cm/pixel respectively. Since we focus on the 2D geometry of the fracture networks, we were
able to omit the placement of ground control points to increase the geo-referencing accuracy. The onboard GPS
receiver of our UAV allows absolute horizontal positioning in the range of few meters and we only refer to the
relative not absolute positioning. Quality control and ground truthing were done by field observation and
measurements on objects of known size in our models. Examples showing that we were able reproduce
measurements in the field with measurements taken in the resulting DEMs and ortho-mosaics are provided in S1.

To identify the optimum between flight altitude and resolution, we took sample photographs starting at 7 m up to
100 m (Fig. 2). We evaluated the spatial resolution in the photographs with respect to fractures that we can identify
in the outcrop. An altitude ranging between 20 - 30 m offers the best balance between flight altitude vs image
accuracy, while being able to cover larger areas in less time at a sufficient spatial resolution to capture all fractures
in the photographs (Fig.2). Photographs taken at the greater altitude of 100 m (Fig. 2a) resolve only the larger
fractures, which is possible because of the high contrast between fracture and rock (cf. Gillespie et al., 2011), but
failing to resolve the smaller fractures sufficiently (Fig.2). By contrast, flying at the lower elevation of 10 m
produced DEMs with greater resolution of 8 mm/pix and ortho-mosaics with 4 mm/pix. Yet, no other fractures

can be identified in the data from these flights as compared to the data from flight altitudes of 25 m (e.g., Fig. 2b vs. 2e). Thus, we opted for a maximum flight altitude of 25 m to reduce survey time/battery and data volume.

An overview of the entire set (Fig. 1b) was created from photographs taken from a flight altitude of 100 m. The spatial resolution in the ortho-mosaic from 100 m is 4.21 cm/pix, which is slightly better than the 5 cm/pixel achieved by Gillespie et al. (2011), (their figures 8 and 9), who used a large-scale camera for an aerial photographic survey by airplane.

### 3.2 Fracture trace mapping

Manually tracing all fractures of the outcrop is a very time intensive task because of the large amount of fractures present in the dataset. Thus, an automatic fracture mapping technique for more rapid data characterization is needed. However, the reliability along with the strengths and weaknesses of automatic fracture detection must be addressed first. To do so, we use both manual and automatic techniques for the same domains, allowing a direct comparison of results. This comparison helps us to identify potential human bias (e.g. Andrews et al., 2019;

Peacock et al., 2019) and gives us a measure of quality of the automatic fracture detection technique, which then can be optimized to map the whole outcrop in future work.

In the manual approach, fractures are traced as polylines from tip to tip, utilizing the digital imagery provided by the UAV survey as base map in ArcGIS. We traced the widely eroded fractures along their medial axis creating as few vertices as possible, while still maintaining the original fracture geometry. Abutting fractures were mapped as

polylines and snapped at the abutting intersections. Snapping is not necessary for cross-cutting fractures, but it is for intersections where a younger fracture converges to an existing junction. Tracing one short segment between two junctions of fractures involves identifying its' start and end points to providing the input in the software to create the trace. Based on our experience, this process takes at least five seconds when the case is clear to the interpreter. Extrapolated to an amount of ~ 7000 segment traces, which roughly represents the amount of segment

traces required to map one domain in the SW, the mapping time sums up to ca. 10 hours. Yet, the estimated time does not include quality and integrity checks or corrections and reinterpretations, which may easily add another five to six hours to the required minimum time of 10 hours to trace the fractures of one domain manually.

An approach to semi-automatically map this outcrop has been published by Gillespie et al. (2011), who used ant

tracking on a lower resolution dataset, resulting in a good overview of the fracture geometry, but incomplete fracture detection (their Fig. 9), which is insufficient for a detailed analysis of crosscutting relationships. To overcome these limitations, we use the technique of Prabhakaran et al. (2019), that allows detailed detection of fractures in our data. The automated trace extraction is based on ridge detection using the complex shearlet transform method as described by Reisenhofer et al. (2016) and extended to fracture trace extraction from drone

photogrammetry by Prabhakaran et al. (2019). The automatic processing consists of serial image processing steps applied to the UAV-images, including ridge ensemble computation, segmentation, skeletonization, and polyline fitting (Fig. 3). The automatic trace extraction requires a selection of parameters pertaining to the shearlet systems. For the Lilstock dataset, we chose a set of shearlet systems that capture fractures of multiple scales and applied it to all image tiles. Subsequently, for each image tile, a single set of shearlets was used and the intensity threshold

(Fig. 3) adjusted to convert ridge ensemble maps into binarized images by visual comparison with the source image as explained in detail in Prabhakaran et al. (2019) . Such a step was necessary owing to the spatially varying fracture intensity.

To preprocess the ortho-mosaics for the automatic trace detection we sliced them into tiles of 1000 x 1000 pixels resulting in six to seven tiles per map domain. Depending on the available hardware the processing time of the feature detection may vary and took us ca. 20 minutes per tile. However, this time does not account for the preparation of shearlet systems, trial and error taken in deciding the intensity cutoff for each image, removal of potentially produced false positives, and manually joining the detected features at the edges of the tiles. Taking these necessary steps into account, the automatic trace extraction of one tile may require one up to two hours in total and between ca. 6 and 14 hours for a map domain.

### 3.3 Network analysis

To analyze the fracture networks, we chose a sampling strategy comparable to the polygon sampling of Nyberg et al. (2018) which allows two-dimensional areal sampling of fracture networks by in a manually defined area or subareas therein. This approach allows us to analyze the domains entirely as areas and to identify possible spatial heterogeneities between and within the map domains.

We used ArcGIS as main platform in combination with the NetworkGT plugin (Nyberg et al., 2018). The ArcGIS environment enables us to measure fracture length and strike directions based on the traced segments of the fractures that are depicted in the georeferenced and orthorectified mosaics.

These 2D fracture networks may be described in terms of vertices and edges that constitute a spatial graph (e.g. Sanderson et al., 2019), in which the set of vertices comprises the intersections of e.g. abutting or crosscutting fractures as well as the terminating points of isolated fracture tips. The sinuosity of the fracture is approximated using piecewise linear edges. Because the intersection points between these edges are also considered as vertices, we will follow the usage of e.g. Nyberg et al. (2018) and refer to intersections of fractures as nodes.

The nodes are classified in terms of node degree, i.e., the number of edges that pass through it, and the edges are classified based on the type of branches. Sanderson and Nixon (2015) specify three node types, isolated (I), abutting (Y), and crosscutting (X) whose relative proportions define the topology of the graph. In our interpretations, the nodes of fractures censored by the boundary of the sample area are classified as end nodes (E). The fracture segments connecting the nodes and referred to as branches may be classified into three types. The branch types are I – I when both nodes at the branch tips are I-nodes, C-I when connected at one tip (via an abutting or crosscutting node) but isolated at the other, and C-C when both tips end at a Y or X node.

The manually-traced fractures were split into segments between intersections to apply these topological analysis tools from the NetworkGT plugin (Nyberg et al., 2018). This step is necessary for the manually-traced fractures because they have been traced along a path from fracture tip to fracture tip and do not yet include information of intersections with other fractures along that path. Furthermore, the traces cannot be processed using NetworkGT in their initial form, because the toolbox requires singlepart features instead of multipart features (i.e. every segment with an own ID instead of several connected segments with a shared ID to represent the complete fracture trace). For the automatically generated traces that are already segmented at intersections as output from the technique of Prabhakaran et al. (2019), this processing step is not necessary.

NetworkGT only supports nodes of the types I, Y, and X nodes of higher order, i.e. more than four branches intersecting at one point are returned as error nodes due to the missing implementation of such cases. To assign the correct order of the node to the returned error nodes, the spatial join function from the ArcGIS Analysis Toolbox was used to count the number of intersecting branches at the specific nodes, allowing us to reassign their

type accordingly as special cases of X nodes, where five (Penta), six (Hexa), seven (Hepta) or eight (Octa) branches intersect. This is possible, because the spatial join function allows matching the branches to the target error node based on their relative spatial locations. The tips of the branches are at the same spatial position as the node, which resembles an intersection, and the spatial join function returns a new field in the error nodes attribute table including the number of intersecting branches at that exact position. Furthermore, linkage to an unknown type of node yields errors in the recognition of branch connectivity. These errors were corrected using SQL queries to select undefined branches to change their connectivity accordingly to their number of connected ends.

### 3.4 Manual classification of fractures into generations

To identify different generations of the manually-traced fractures, we used abutting and cross-cutting relationships as discussed in Peacock et al. (2018), aided by general fracture attributes such as length and strike. Fracture generations were assigned following the criteria below:

Gen. 1: Longest fractures that traverse through the study areas. Absolute orientations are not used as criterion because they may change in converging patterns as present in the NE of "the bench".

Gen. 2: Subparallel orientation to gen. 1, shorter in length and abuts on to gen. 1.

Gen. 3: Shorter than gen. 1 and 2, oriented at an acute angle to gen. 1 and 2 and abuts on to them.

Gen. 4: ENE-WSW striking, sub-orthogonal to gen. 1 and 2 and abutting gen. 1 and 2, and both abuts and crosscuts gen. 3

Gen. 5: Shortest fracture traces with a large range of orientations and abuts all other generations sub-orthogonally, creating polygonal shapes

Distinguishing generations 1 and 2 by length is not possible for all fractures because large portions of individual traces continue outside of our domains. Thus, we qualitatively investigated the continuation of the fracture trace towards the outside of the map domain to aid the interpretation. A fracture set analogous to generation (gen.) 3 in the SW domains was absent in the NE domains. Our interpretation of fracture generations is consistent with the four main joint sets identified in Rawnsley et al. (1998) for the northeastern areas. Other interpretations of joint populations in other locations at Lilstock (Belayneh and Cosgrove, 2004; Engelder and Peacock, 2001; Loosveld and Franssen, 1992) identified up to six generations of joints.

### 4. Results

### 4.1 Fracture trace segments

We traced fractures in five different domains of 140 m² each (Fig. 1b) manually and automatically. To make both datasets comparable, the manual traces were split into segments between nodes to resemble the automatically extracted trace segments. This step is necessary because the automated extraction results in traces segmented at the fracture intersection nodes. The number of overall trace segments in automatically traced networks is, therefore, greater than the number of fracture traces as would be described by an interpreter mapping complete fractures that consist of several segments along a path from fracture tip to tip, which makes a direct comparison of both methods difficult.

The number of automatically-extracted segments is greater than the number of the manually-traced ones in all domains (Table 1), whereas the difference of traced segments within the domains is smaller in the NE domains and larger within the SW domains. The cumulative length distribution of segments and associated plots of log-normal standard deviations for automatically-extracted traces (Fig. 4) and manually-traced segments (Fig. 5) for each domain are similar to each other when the methods are compared qualitatively, despite the aforementioned difference in the number of segments. For both methods, the cumulative length distribution and log-normal standard deviation plots in fig 4 and fig. 5 show that fracture traces of all domains resemble the characteristic negative power law associated with fracture traces. A qualitative comparison of the histogram distributions within both, the automatically-extracted segments (Fig. 4) and the manually-traced segments (Fig. 5), shows a greater similarity within the groups of spatially close domains in the SW or the NE than a comparison between SW vs. NE. While the overall histogram distributions of automatically-extracted segments (fig. 4) and manually-traced segments (fig. 5) appear similar and the plots have peaks at comparable segment lengths and mean segment lengths (Table 1) when compared to each other, the distributions show that the automatic extraction results in a relatively higher amount of smaller segments.

The calculated mean lengths of the automatically-traced segments are similar to each other in the SW and range from 16 to 17 cm in the three domains, while the difference from 15 to 21 cm is greater in the NE domains. The variation of the mean lengths between the SW and NE domains for the manually-traced segments is similar to the automatically-extracted ones and the mean lengths of manually-traced segments are consistently greater by a difference of 1 to 4 cm (Table 1). Both methods show that the longest segments are in NE2 as compared to the other domains, which is consistent over the greater values for the NE2 25%, 50% and 75% quartiles (Table 1).

The cutoff length for image sampling for all windows is 1 cm, therefore values smaller than 1 cm are not significant. The maximum lengths of the segments may be censored by the sampling windows. Apparent maximum lengths of the segments are largest in SW2 and shortest in NE1 which is consistent in both tracing methods, even though manual maximum segment lengths may be up to 16 cm larger than their automatically-traced counterparts (Table 1). The covariance is positive and similar in all domains and both methods, ranging from the lowest in NE1 to the highest in SW1. The kurtosis in all domains and for both methods is positive, largest in SW1 and overall larger in the SW domains than in the NE domains, which suggests that length-frequency-distributions are closest to that of a half-normal distribution (kurtosis = 3) in SW1 with a decreasing trend towards the NE, what suggests that segment lengths towards the NE have fewer outliers in the form of longer segments. This characteristic of the length-frequency distribution is shown further by the positive skewness in all domains and both methods with a decreasing trend towards the E as well, which further indicates that the branch distribution in the NE domains is more symmetric, while the distributions in the W are more asymmetric with a tail towards the right. This characteristic of the distribution suggests that the SW domains tend to have more segments that are longer than most of the segments traced or extracted in the respective domain as compared to the domains in the NE.

To compare the resulting networks of both methods and all domains spatially we calculate fracture trace segment density and fracture trace segment intensity. Therefore, we use the $P_{ij}$ system (Dershowitz and Herda, 1992) as denotation. In the $P_{ij}$ system, "i" gives the dimension of the sample which is "2" for two-dimensional maps of fracture traces. The second index "j" gives the dimension of the measurement, which is "0" for fracture density that quantifies the number of fractures per unit dimension, or "1" for fracture intensity that measures the total

fracture persistence per unit dimension (Dershowitz and Herda, 1992). In the case of our results for both, $P_{20}$ and $P_{21}$, it must be noted that these do not represent actual fracture density and fracture intensity in the domains, but density and intensity of the fracture segments.

### 4.1.1    Fracture trace segment densities

To provide examples and a direct comparison of the manually- and automatically-extracted traces we selected the $P_{20}$ plots of the domains SW3 (Fig. 6 and NE2 (Fig. 7) that show the density of fracture segments windows of 0.5 x 0.5 m within the domains and the network of traces in the background. Therefore, these figures allow us to make
a direct comparison of the number of segments traced in a certain part of the domain between the two methods. Plots of $P_{20}$ and $P_{20}$ absolute and relative differences for all domains are provided in the supplement S2 and S3.

In domain SW3 (Fig. 6) qualitative comparisons of the manual and automatic traces show an overall similar spatial distribution of areas with relatively higher and lower $P_{20}$ values, except for an area between 4 and 6 m in the x-direction and 3 – 4 m in the y-direction . Plots of the $P_{20}$ absolute and relative difference at the bottom (Fig. 6)
show that the automatic trace extraction resulted in a higher $P_{20}$ value in this area which can be verified visually by the greater amount of automatically-extracted short segments of the underlying fracture trace network which are not represented in the manually-traced segments. Overall, $P_{20}$ is higher (58.9 m$^{-2}$) for the automatically-extracted segments than for the manually-traced segments (49.8 m$^{-2}$) (Table 1).

A similar comparison made for domain NE2 (Fig. 7) shows fewer differences in the segment density for both
methods when compared to SW3 (Fig. 6). Again, the plots of the $P_{20}$ absolute and relative difference at the bottom of fig.7 show that the automatic method extracted small segments that are not represented in the manually-traced network, mostly at areas at 4 and 8 m along the x-axis and 0.5 m along the y-axis and 16 m along the x-axis and 5 m along the y-axis. The greater similarity of $P_{20}$ for both methods in this domain is also evidenced by the overall $P_{20}$ values of 30.9 m$^{-2}$ for the manually-traced network and 34.5 m$^{-2}$ for the automatically-extracted network (Table
1).

### 4.1.2    Fracture trace segment intensities

The $P_{20}$ plots presented in the preceding section allow us to compare the spatial distribution of the number of segments per unit area and further enable us to isolate areas where the resulting number of trace segments differs between methods. However, the resulting networks derived from the two methods require another comparison than
just $P_{20}$, because arguably a single fracture trace can be mapped as a path consisting of a different numbers of traces by different methods, while the overall path geometry of the fracture from tip to tip along the segments may be the same. Therefore, $P_{21}$ allows us to compare the lengths of segments per unit area for both methods, where the similarity between methods suggests that the same fracture traces were recognized and extracted or traced with similar lengths regardless of the number of segments that represent the fracture trace.

The trace maps are depicted as an overlay on a $P_{21}$ fracture intensity plot (Dershowitz and Herda, 1992) for both methods and all domains in Fig. 8. To provide a good comparability to the $P_{20}$ plots in figures 6 and 7, the same cell size of 0.5 m * 0.5 m was chosen. A qualitative comparison between the resulting $P_{21}$ plots of automatic segment trace extraction and manually-traced segments shows a high similarity in all domains because the
distribution of cells with relatively higher or lower segment intensity within the respective domains is the same

for both methods. However, smaller areas within the domains can be identified in which $P_{21}$ is higher for the automatically-extracted traces (Fig. 8). An example of higher $P_{21}$ can be observed in SW3 between 4 and 6 m in x-direction and 3 – 4 m in y-direction (Fig. 8), where $P_{20}$ was also higher and overall more smaller fracture traces were extracted automatically than traced manually (Fig. 6.). To better isolate and quantify areas in which the two methods give different $P_{21}$ results, the difference between the manual and automatic interpretations is depicted as a spatial map of as $P_{21}$ fracture intensity difference in fig. 9. A qualitative comparison of the two methods between the domains again shows, that the differences are minor and not larger than ca. 4 m$^{-1}$ per cell. Examples of areas with highest differences are in SW3 between 6 and 8 m along the x-axis and 7 m in y-direction (fig. 9), which are again caused by an overall larger amount of traces extracted automatically than manually (see also fig. 8). In NE2 several neighboring cells at 14 m in x-direction and between 0 and 2 m along the y-axis show the opposite case (fig. 9), where the manual interpreter traced two parallel fractures which were extracted as one automatically.

Overall, resulting $P_{21}$ values for all domains are very similar for both methods with a minimum difference of 0.01 m$^{-1}$ between the methods in SW2 and NE2 and a maximum difference of 0.25 m$^{-1}$ m in NE1 (Table 1). Unlike the results of $P_{20}$, which are greater for the automatically-extracted trace segments than the for the manually-traced segments for all domains, the resulting $P_{21}$ values might be greater or smaller for either method when compared to the other, varying in between the domains (Table 1).

Analyses of the $P_{21}$ fracturing intensity for both methods shows, that $P_{21}$ is overall greater in the SW domains compared to the NE domains, and greatest in SW2 and smallest in NE2 (Table 1).

### 4.2 Network characteristics and spatial variation of the automatically-traced fracture segments

The traces resulting from the automatic technique are segmented, thus only the final cumulative network can be analyzed, because the correct identification of abutting and crosscutting relationships is a prerequisite for identifying age relationships, requiring a review of the complete fractures, not just segments. The networks resulting from the automated data collection depicted as branches and nodes did identify all visible fracture traces successfully (Fig. 10 and 11). The node topology statistics indicate that the network is well connected because it consists of only few isolated nodes and an abundance of abutting (Y) nodes (Table 2 and compare with Fig. 12 upper). Isolated nodes decrease in number from W to E. The number of Y nodes lies between 3919 in SW3 and 4390 in SW2 in the SW domains and has maximum and minimum values in the northeastern domains, where NE1 has the largest number (5100) and NE2 the smallest (2517). The number of X nodes along with Penta nodes increase from SW1 towards SW3 and NE1 to NE2, with NE1 having the smallest number of X nodes of all five domains. Length weighted rose plots (Fig. 13) show a significant variation in correlation of fracture length to orientation between NE and SW regions. In the SW study areas, the fracture pattern is similar in all three areas (Fig. 10). In the NE2 study area proximal to the single NE-SW-trending fault that transects the entire exposure, there are two predominant fracture sets (Fig.13). In NE1, the same two sets persist (Fig. 13), but the overall fracture pattern is polygonal in geometry (Fig. 11).

### 4.3 Manually interpreted fracture generations and their spatial variation

Fracture generations for the manually collected data were defined by using the rules in section 3.4. Figures S4 to S8 in the supplemental data enable the reader to compare these manually derived trace maps to the digital base maps.

### 4.3.1 Southwestern domains

The SW domains (Fig. 14) include five fracture generations that differ in their distribution through the domains. In SW1 and SW2, gen. 2 fractures are more numerous than gen. 1 fractures. In SW3, this relation is reversed. Overall, the generations 1 and 2 have similar geometry. Generation 3 is represented in all three domains with increasing abundance from W (SW1) to E (SW3), coinciding with an increasing fracture intensity. Generation 4 only represents a small number of fractures in each domain, while gen. 5 is present in large numbers. Average values of fracture length and strike for the domains in the SW are presented in Table 3. To visualize the relationship between fracture strike and length, their values were plotted for each fracture and color-coded according to the associated fracture generation (Fig. 15). Fractures of gen. 1 and 2 trend between 90° and 120°, reaching a maximum (censored) length of 18 m (Fig. 15). Generation 3 resembles a Gauss distribution around 80° with most fractures reaching lengths up to 2 m, but also including lengths up to a maximum of 10 m. Generation 4 plots around a mode of 060° with lengths up to 3 m. Fractures associated with gen. 5 cover a wide range of strike directions. Fractures in SW2 resemble the distribution of SW1 with no obvious differences. Fracture length-related distributions in SW3 (Fig. 15) show deviations from SW1 and SW2: the cluster with the longest fractures between 100° and 120° is dominated by gen. 1, while the respective clusters in SW1 and SW2 are interpreted as mostly gen. 2, coinciding with prior qualitative observations.

### 4.3.2 Northeastern domains

The network in NE2 (Fig. 16) mainly consists of many gen.1 fractures that converge beyond the southeastern edge of the sample window towards a fault in the SE, while NE1 has a less unimodal distribution of length-weighted fracture orientation for the fracture generations. Notably, the northeastern fracture networks lack gen. 3 fractures as compared to the SW sample windows. The gen. 2 and 4 fractures crosscut and abut on each other, so they do not have a clear relative age relationship. Like the domains in the SW, gen. 5 is present in both NE domains and abuts all older generations without a preferred orientation. Averages of fracture length and strike for NE1 and NE2 show some differences to the domains in the SW (Fig. 17 and Table 4). Generation 1 mostly occurs between 120° and 160° in both, NE1 and NE2, reaching almost 9 m maximum length. Generation 2 shows a wider cluster in NE1 with most fractures striking between 80° and 140° and a similar distribution as gen. 1 in NE2. The cluster associated with gen.3 in the SW (Fig. 15) is not present in NE1 or NE2 (Fig. 17). Another cluster at 60° is associated with gen. 4 in NE1 and NE2, which is the same in all three areas in the SW. Generation 5 is widely spread in all directions, especially in NE1, while a maximum of gen. 5 fractures between 20° and 60° is present in NE2.

### 4.4 Fracture network evolution

To analyze the impact of the fracture generations on the evolution of the overall network, fracture traces were split at intersections into branches and the intersections represented as nodes, to resemble the fracture networks generated from the automatic trace detection (Fig. 10 and 11). The results are presented for SW1 - SW3 in figures 18 - 20 and NE1 and NE2 in figures 21 and 22, showing the network evolution in time steps by adding the next generation to the subsequent subfigure. The following sub-sections summarize the major changes for every domain during the network evolution to guide the reader through figures 18 - 22 and tables 5 - 9.

### 4.4.1 SW1

The initial time step comprises three sub-parallel branches of gen. 1 fractures (Fig. 18a and Table 5). Their full lengths fall outside the mapped area and they are censored by the map boundary resulting in three isolated (I – I) branches of 3 m length on average. With gen. 2 (Fig. 18b) more sub-parallel branches are added, of which a large number abut at the map boundary, resulting in further isolated branches and an increased average length of 4.69 m. Some gen. 2 fractures abut at gen. 1 in acute angles, resulting in Y nodes at connected (C – C) branches. Generation 3 strikes at an angle towards gen. 1 and 2, either crosscutting the older fractures (X nodes) or abutting them (Y nodes) (Fig. 18c), leading to a shorter average branch length of 0.59 m. Few fractures interpreted as gen. 3 end in I nodes. Generation 4 is sparsely represented in SW1, so the impact on the fracture network as compared to the previous generations is minor (Fig. 18d). With gen. 5 (Fig. 18e) the network becomes more spatially dense, further reducing the average branch length to 0.2 m. The fractures may abut on older generations, but also tend to join in existing junctions, leading to the development of Penta and Hexa nodes.

### 4.4.2 SW2

In SW2, generation 1 is represented by four fractures, two of which abut on the mapping boundaries at each side and two end in I nodes (Fig. 19a and Table 6). Most gen. 2 fractures end in either I nodes or at the map boundary but can also connect to the tips of preexisting gen.1 fractures as observed on the largest gen.1 fracture in the center (Fig. 19b). Generation 2 leads to a decrease in average branch length from 7.98 m to 4.51 m. Generation 3 has a large impact on the network, further reducing the average branch length to 0.54 m due to many fractures crosscutting older fractures (Fig. 19c), resulting in a sharp increase of Y and X nodes. Generation 4 has a minor impact (Fig. 19d), not significantly altering the network parameters. Generation 5 (Fig. 19e) again reduces the average branch length to 0.2 m, while reducing the number of I nodes and strongly increasing the number of Y and X nodes.

### 4.4.3 SW3

Compared to SW1 and SW2, gen. 1 fractures are more numerous in SW3, however, they are still not well connected with most fractures ending in I or E nodes at the map boundary (Fig. 20a and Table 7). Generation 2 is less pronounced in this area with only a small impact on the network (Fig. 20b). The average branch length slightly increases by 0.11 m, as few long fractures are added. With the addition of gen. 3 to the network (Fig. 20c), the average branch length is reduced from 3.17 m to 0.59 m, again with a strong increase of Y and X nodes. With the addition of gen. 4 (Fig. 20d), I nodes are reduced in favor of Y and X nodes. Generation 5 (Fig. 20e) further reduces the average length to 0.2 m with a sharp increase of Y and X nodes.

### 4.4.4 NE1

Generation 1 consists of several sub-parallel branches that mostly end in I nodes or at the map boundary (Fig. 21a and Table 8). Abutting fractures are sparse, as only four Y and zero X nodes are observed, leading to an average branch length of 4.33 m. Generation 2 (Fig. 21b) strikes at an angle to gen. 1 leading to an increase of cross cutting and abutting nodes, that are present in almost equal numbers. Also, the number of I nodes is quadrupled. The average branch length is reduced to 0.79 m. With the addition of gen. 4 (Fig. 21c), the average branch length is

further reduced to 0.68 m, accompanied by an increase of abutting (Y) and cross cutting (X) nodes. Generation 5 (Fig. 21d) further reduces the average branch length to 0.24 m with a strong increase of Y nodes from 547 to 4135 and X nodes from 229 to 705.


### 4.4.5  NE2

In this subarea, gen. 1 is present as long subparallel branches that spread out radially from a fault outside of the southeastern map boundary (Fig. 22a and Table 9). The branches barely abut or intersect each other or create I nodes. Most of the branches run into the mapping boundaries at opposite sides, leading to an average branch length

of 4.14 m. Generation 2 crosscuts gen. 1 almost orthogonally (Fig. 22b) leading to an increase of Y and X nodes along with a reduced average branch length of 0.8 m. Generation 4 (Fig. 22c) further reduces the average branch length to 0.45 m, with a slight increase of X and Y nodes. With the addition of gen. 5 (Fig. 22d), the average branch length is reduced to 0.16 m along with the addition of a large number of Y and X nodes.

A direct comparison of the node distribution for all domains is shown in ternary plots highlighting the changing node distribution during the network development (Fig. 12 upper). Results of the latest network state (analog to gen. 1 – 5 for manually-traced fractures) from the automatic trace extraction are depicted in Fig. 12 lower for a direct comparison.

## 5.  Discussion

### 5.1 Manual vs automatic tracing

Manual tracing of the fracture network is comprehensive but time-consuming. At this quality of outcrop and resolution of imaging, human bias is minor (e.g. Andrews et al., 2019; Peacock et al., 2019), and similar results can be produced by different interpreters (e.g. Long et al., 2018). A manual interpreter's work can gain quality control from considering the work of another interpreter, while the reliability of the automatically mapped network

entirely depends on ridge detection and image post-processing parameters.
Time is an important variable for both methods and was estimated in section 3.2 to compare manual and automatic trace extraction. For a network of traces in one domain a minimum time of ca. 15 hours is required for the manual tracing when quality checks are incorporated, which is comparable to the maximum time required for the complete automatic trace extraction. During these estimated periods of time, the interpreter's attention is required all the

time in the manual method, whereas in the automatic method the interpreter only has to check the input parameters and the results. Thus, automatic mapping may save personal time of the interpreter, it is overall fast compared to manual mapping and additional time can be saved by parallelization of the process. However, the network needs to be checked for artifacts along with a general estimation of the reliability or capability of the method before it can be applied widely.

Based on an extrapolation of the time required to manually map the five domains, a complete interpretation of the whole dataset incorporating the so far unmapped areas of the bench and the other layers would take a manual interpreter several hundreds of hours of pure mapping time. An estimation of the required time for the automatic tracing to complete the whole outcrop based on the time required for the presented domains is not trivial because several aspects need to be considered. We selected areas of good outcrop quality and high fracture visibility for

our map domains. Other regions covered by the ortho-mosaics include areas where the water in the fracture cavities has already dried out, what locally reduces the contrast. To achieve results of good quality in those areas, more shearlet combinations and different thresholds and parameter settings are required. In highly eroded areas, more time is required to remove false positives. Therefore, the chosen image processing parameters greatly influence the automatic extraction and speed up the overall process when the extracted network has minimal artifacts. If the

automatic extraction returns many false positives and incorrect node connections, correcting them can be more time-consuming than an initially correct manual interpretation.

In either case, an advantage of the automatic tracing is the ability to reproduce results solely by choosing the same parameters, while manual reproduction requires the interpreter to follow a clearly defined set of rules that can

become excessive depending on the complexity of the dataset. Examples are widely eroded fractures that can be traced along their edges or the median of the cavity, which may also lead to different interpretations, especially at widely eroded junctions. These junctions can either be interpreted as two younger fractures closely abutting on an older one, or only one younger fracture that crosscuts the older one. This is of particular importance for unexperienced interpreters who have to decide whether to learn how to use the software and set up the rules as

explained above, or how to apply the automatic trace extraction code written for MATLAB. While the ability to learn either method strongly depends on the individual, the advantage for such users is that the automatic traces provide unbiased segments which can be used to guide the interpretation and maintain consistency and quality. However, using the fracture mapping code of Prabhakaran et al. (2019), it is only possible to generate quality segmented networks of branches and nodes for well exposed patterns. The interpretation of fractures longer than

a segment between two nodes and the association to a certain generation depending on crosscutting relationships still must be done manually.

For the data presented in this work, the results of both techniques are very similar, and to better highlight the dissimilarities of the traced fractures, the differences of the $P_{20}$ and $P_{21}$ analyses are presented in Figs 6, 7, 9 and

supplements S2 and S3).  $P_{21}$ plots are expected to have greater values for automatically-traced fractures, because the traces are expected to be more sinuous than the manually-traced ones, given that the automatic code generates traces based on the detected ridges, whereas a manual interpreter tends to trace using as few vertices as possible. Regions where $P_{21}$ is greater for manual traces such as in NE2 at 14 m in x-direction and between 0 - 2 m in y-direction (Fig. 9) show cases where the interpreter can draw a trace based on geological knowledge and no trace was identified automatically. More general examples for cases like this are thinning out or merging fracture

openings where the automatic detection stops but the human interpreter continues the trace, resulting in longer or more numerous traces (see also e.g. Fig. 23 mark 7). However, when using the $P_{ij}$ system that further subdivides the domains in smaller boxes to analyze the differences between two methods, it should be noted that apparent differences might also be cause by the position of the segments and nodes relative to the gird cells. The whole

network is clipped by the grid cells and intersection points of some fractures may fall into a different cell due to a different interpretation, e.g. at the widely eroded intersections of fractures. When one intersection point falls into another cell due to a different interpretation, it not only increases the count in one cell, but also reduces the count in the other which results in neighboring cells with one having a positive and the other a negative difference (e.g. in NE1 at 5 m in x and 5 m in y direction, fig. 9).

Fig. 23 illustrates a direct comparison of the manual interpretation and automatically extracted fracture networks. Increasing complexity of the fracture network causes more differences in the interpretations, mainly in the interpretation and number of nodes, while the average lengths of the branches only differ by a few centimeters (see also Table 10). The manual interpretation favors nodes of higher degree in direct comparison to the automatic interpretation in most cases (Fig. 23, see also Fig. 12). The biggest difference is the greater number of Y nodes

counted in the automatic network. This difference results from i) the overall larger number of fractures identified by the automatic process along with ii) the bias of the code towards nodes of lower degrees. This bias is caused by broadly eroded fractures that lead to inaccuracies when the code is tracing one of the edges instead the medial axis of the fracture, as a manual interpreter would do. This artifact leads to an overall higher number of nodes of smaller degree in the automatic traces for the same sample window as for manual traces (see Fig. 23 mark 4 and Table

10). More short traces are detected leading to a greater number of isolated nodes (Fig. 23 mark 5). Due to the erosion the limestone becomes rough at its surface and can develop a structure which may be interpreted as a fracture trace in the automatic extraction. These false positives are caused when the parameters of the code are chosen in a way, that they are too sensitive. In the manual interpretation, these structures were interpreted as erosional surface features and not as fractures. Thus, the difference here highlights the importance to find the right

parameter combination of the image processing parameters, to find the middle ground between the detection of wrong positives and false negatives. Compared to the ant-tracking method (Gillespie et al., 2011) applied to a much lower resolution (5cm/pixel) dataset, we deem our results more reliable because our spatial resolution of ~1 cm/pixel allows us to resolve details smaller than the observed width of the eroded fractures, that may add up to several cm. This resolution allows us to make interpretations based on features in the same scale, as it would be

done directly on the outcrop.

We infer that the automatic code at this stage represents a good option for creating an initial fracture trace map that only differs from a manual interpretation to a degree that is comparable to the deviation of two manual interpretations of the same fracture network (e.g. Long et al., 2018). More complex tasks, like an interpretation of age relationships based on abutting and crosscutting criteria still require manual input. Based on this, future work

can include the extension of the automatic mapping routine to the whole outcrop and use the manual interpretations to define criteria to combine automatically mapped branches into fractures and to assign fractures to predefined generations.

### 5.2 Classification into fracture generations

The classification of fractures into generations depends on the expertise of the interpreter. Locally in the sample

windows, assignment to one of several generations is possible for a single fracture trace. In those cases, the interpreter has to decide with possible human bias. Examples for cases where an interpretation of the generations is non-unique or different interpretations of the underlying geometry are possible because of a locally lower quality of the data (e.g. eroded areas) are provided in figure 24.

Figure 24a shows an example of a fracture interpreted as gen, 2 by an elimination process based on the predefined

rules and interpretations. During this process, the other generations were ruled out and gen. 2 remained as the most likely one in the eyes of the interpreter. We interpret these cases as the reason for outliers in figures 15 and 17, in which the plotted point representing a fracture outlies the rough distribution of the other fractures associated to the same generation. Even though rules for the interpretation of generations based on abutting and crosscutting criteria may be clearly defined, their implementation is not trivial as shown in Fig. 24a mark 1. At the shown location, a

clear identification of the abutting relationship of the 3 fractures is not possible because the junction is eroded, and several fractures appear to intersect at the same location. In other cases, the interpreter must decide whether a junction is the result of a splaying fracture or a younger fracture abutting at an older one (Fig. 24b mark 2). While the dataset of ortho-mosaics provides a good contrast between eroded fracture and rock in most areas, few cases where the location of the fracture tip is not clearly distinguishable are possible (Fig. 24b mark 3). The tracing of

fractures as one-dimensional lines leaves further room for the interpretation of the position of the fracture when the original fracture has been eroded widely. In those cases, the interpretation must be based on an area that envelops the actual fracture instead of clear trace of the fracture at the surface (e.g. Fig. 24c mark 4 and panel d mark 7). Intersections of fractures that are widely eroded can be interpreted as an intersection of all fractures, or several spatially close intersections of several fractures (Fig. 24c mark 5), which influences the results of analyses

of the network connectivity (c.f. Fig. 23). Other complex interpretations are required, when e.g. a long fracture matches the criteria for an old generation, but its trace appears to abut on a younger fracture, or the trace bends and continues with a geometry that qualifies the fracture for another generation (Fig. 24d mark 6). One possible explanation for this outcome is when old fractures are reactivated and abut younger fractures or fractures interact with local features, e.g. preexisting fractures, that may cause distortions in the geometry of later-formed fractures.

This effect is visible e.g. in figures 15 and 17 where fractures that overlap with the point clouds of the youngest generation in the plots were assigned to an older generation during the interpretation.

    Examples of the types of judgment calls that an interpreter may need to make for gen. 1 traces include accounting for the effect of censoring by sample windows on considerations of fracture lengths, assigning generation to a

fracture trace with generations share orientations, and deciding if nearly parallel fractures are one or two generations. For our early generations, fracture length is a biased parameter, because the fractures can be censored, such that they are longer than the respective dimension of our map domain. The circumstance of possible censoring of long gen.1 fractures highlights the impact of fracture geometry on possible interpretations and results when selecting location and dimension of a sampling window. The strike directions of gen.1 fractures in the SW (100°

- 120°) and NE (120° - 160°) differ, raising the question whether these are two independent sets. Considering the radial/converging pattern of gen.1, that can be observed best on a larger scale on the eastern part of "the bench". Here, we interpret the fractures with different trends as belonging to a larger structure, and hence a single generation. The underlying reason for this interpretation is that fractures that are subordinate to a larger structure like this fault (Fig. 1b) can form simultaneously but in different directions. Thus, one criterion cannot rule out the

other in this special case.

    In the southwestern domains generations 1 and 2 are very similar in length and strike, so they can only be distinguished when gen.2 fractures bend at the tips and abut on gen.1. Considering this observed geometry, another possible interpretation is to merge the first two generations in the SW into one, where the geometry is simply recording the order in which fractures of the same generation formed. We opted for two separate generations,

because they are interpreted as two consecutive generations and therefore this decision does not have an impact on the analysis of the succeeding network development.

    Gen. 2 is more distinct in NE1 than NE2 where gen. 1 has greater abundance with narrower spacing, restraining the development of younger fractures on one hand and causing them to appear like gen. 5 on the other hand, possibly leading to a mix-up in the interpretation of the generations.

Generation 3 is present in all domains in the SW but absent in the NE. The disappearance of such a distinct fracture generation over a relatively short distance of 200 meters can be explained by a wrong association of fractures with other generations or subject to reasons that are not within the scope of this study. The geometry and distribution of preexisting fractures strongly influences the development of younger generations, possibly creating local variations. Considering that generation 1 and 2 in the SW could belong to the same set, gen. 2 as mapped in the

NE could be equivalent to generation 3 in the SW. This assignment would mean that all domains incorporate the same generations but with local variations in their geometry. Based on our analysis with five spatially isolated domains, this determination cannot be easily made, but requires a continuous tracing of the fracture generations over the complete outcrop area. This interpretation does not have an impact on the network evolution analysis, because the generations are merged in the same order, but the situation highlights the necessity of a complete

automatic tracing and interpretation of the whole outcrop.

Given the consistency of gen. 4 traces across the five sample windows, we focus next on gen. 5 fractures, that created polygonal patterns with generally shorter traces than the other fracture generations (< 0.5 m) and are present in all areas. They barely show orientation modes, but a qualitative inspection of fig. 15 and 17 suggests that a large fraction of the gen. 5 fractures is oriented in N-S and E-W directions, which appears to be caused by

the influence of pre-existing fractures (e.g., in the NE2, where gen. 5 strikes between 20° and 60°, which is orthogonal to gen.1 fractures, representing the shortest connection between them).

For most cases, we expect and observe that younger fractures are shorter than older ones because pre-existing fractures acted as propagation barriers, constraining the maximum lengths of the younger fractures. Counterintuitive to that, our gen.1 fractures can be shorter than gen. 2, and gen. 3 fractures shorter than gen. 4

(Tables 3 and 4). We interpret this anomaly in the case of gen. 1 and 2 to relate the selection and orientation of our domains relative to the gen. 1 fractures, so that they are more strongly censored than gen. 2 fractures. The orientation of gen. 3 is subparallel to preexisting generations, thus the fractures are more likely to abut. This is not the case for gen. 4, which is sub-orthogonal to generations 1 and 2 und thus more likely to cut through existing fractures, because they were able to propagate through the older fractures for reasons beyond the scope of this

project (e.g., propagation stress conditions, existence of mineral fill in the older fractures, etc.).

## 5.3 Network analysis

Analyzing the distribution and geometry of the branches shows that both can change over distances of a few meters within the same limestone layer. These changes may follow a local trend, e.g. decreasing skewness of the segment distribution plots (fig.4 and 5, Table 1) from W to E that indicates that the branch length distribution in the NE

domains is closer to a symmetric one, while the distributions in the W are more asymmetric with a tail towards longer branches, suggesting that fractures in the W may consist of longer segments as in the NE. However, also strong fluctuations were observed, such as with the number of branches, influencing the magnitude of $P_{21}$ fracture intensity, which has a difference of almost a third from the greatest value in SW2 to the smallest in NE2. Node distributions are linked to the number of branches and the way they interact. The decrease of I nodes from SW to

NE suggests a consistent spatial trend. However, this observation of a trend does not apply to the numbers of Y nodes, which fluctuate over short distances, e.g. from 2517 to 5100 from NE2 to NE1 (Table 2), a percentage difference of 68%. This effect underlines the heterogeneity of the fracture network, even though it might appear relatively homogeneous when observed qualitatively, and the necessity for sampling representative domains, when it is not possible to map the complete fracture network. In this study, we selected the domains primarily based on

the quality of the data as explained in section 2.2. When the sampling areas are supposed to be selected in a way to represent the complete fracture network in its variety, preliminary investigations are required. First steps to identify those representative domains can consist of a qualitative analysis of the network followed by a sparse interpretation of the most prominent fracture sets to reduce the risk of their lack in the chosen domains. However, a reliable statement whether the network in a small domain is representative for the entire network can only be
made when the network has been analyzed entirely.

### 5.3.1 Network evolution

Topological analyses of the fracture network evolution show that average branch lengths decrease with additional
fracture generations. This outcome is expected for non-parallel fracture sets which will eventually abut or crosscut each other, contemporaneously increasing the count of Y or X nodes. We identified nodes with more than four branches intersecting at one point in both, manually and automatically extracted traces. Depending on the number of intersecting branches, these nodes are treated as special cases of X nodes: Penta- (5), Hexa- (6), Hepta- (7) or Octa- (8) nodes. Due to the widespread erosion of the fractures at the surface, it is not possible to tell
macroscopically, whether they are narrowly spaced X and/or Y nodes or true nodes of a higher degree. In a spatially dense and strongly connected fracture network, we consider it as possible.

Isolated (I) nodes are more numerous in the initial network stages, where fractures have more space to develop and propagate through the limestone without encountering stress shadows of pre-existing fractures. At later stages, most I nodes become connected to other fractures reducing their overall number. Compared to Y or X nodes, the
number of I nodes is much lower in general, except for the initial fracture generations. In some cases, initial I nodes of old fracture generations appear to abut to younger fractures. Reasons for this geometry might be reactivation of the fracture, or younger fractures connecting with the tip of the pre-existing ones (c.f. fig. 24). In these cases, a unique interpretation is not always possible because abutting criteria become unreliable and other criteria such as length and strike must be considered to aid the interpretation.

The number of Y nodes increases when generations of similar orientation interact with each other or short fractures connect larger ones. X nodes are often the result of intersecting fractures with orthogonal or sub-orthogonal orientation. Nodes of higher degree (5+ branches) are the result of X nodes, to which a younger fracture (mostly gen. 5) abuts.

The average branch lengths (Tables 5 – 9) show a trend of decreasing branch lengths with an increasing number
of nodes. These trends are caused by an increasing number of younger fractures that crosscut or abut on the older ones.

Tables 5-9 show the development of branch lengths have about the same order of magnitude over all sample windows. Especially in the final stage of the network, the average branch lengths are very similar in all sample windows. This outcome indicates that the last fracture generation has a strong impact on the overall network
topology. Older fracture generations have a larger influence on network geometry, because pre-existing fractures influence the geometry of the later fracture generations in terms of possible fracture lengths and distribution. However, the topology can be very similar when younger fracture generations develop and infill the network.

**6 Conclusions**

We used an UAV to take several sets of overlapping images from different altitudes to create orthorectified mosaics
of the fractured limestone pavement on the coast near Lilstock in the Bristol Channel, UK. Based on these orthorectified mosaics, we selected 5 domains based on their outcrop quality and traced the fractures therein using two methods, an automatic trace segment extraction code from Prabhakaran et al., (2019), and manual tracing of the fractures. This allows us to compare both methods in terms of time usage, similarity of the resulting segment traces and network topology. Using the manual interpretation of fracture generations, we further analyze the
evolution of the network connectivity and discuss spatial variations within the larger network based on the results of both techniques to highlight differences between the five domains. The main findings of this study are listed below.

- A comprehensive dataset of the fractured pavement with a so far unpreceded resolution was created using
UAV photogrammetry.

- Automatic trace extraction of the fractures in the dataset is faster than manual tracing, when the parameter combinations used for the automatic extraction are chosen correctly. Furthermore, the manual tracing requires the interpreter's attention throughout the complete process, while the automatic trace extraction only needs supervision during some of the steps and further reduces the time that the interpreter actively
spends on the task.

- When the parameter combinations are chosen improperly, the automatic method may produce a great number of artifacts that require manual corrections, that may take more time than an initially correct manual interpretation.

- Automatic trace extraction results in a greater number of overall segments as seen in the fracture trace
density (Fig. 6 and 7 and supplement S3, Table 1). However, the overall identified fracture traces are similar in both methods, as suggested by similar fracture trace intensities in all domains (Fig, 8 and 9, Table 1).

- Resulting network topologies are similar for both methods, however, the automatic technique is biased towards a greater number of nodes of smaller degree, while a manual interpreter tends to create less
segments and connects more branches at a single node.

- Using the automatic method, an interpretation of relative age relationships between fractures is not yet possible, this requires a manual interpretation.

- The five inferred fracture generations are not equally distributed throughout our five selected areas, the spatial variation is significant in the same layer.

- The selected size of the mapping area can impact the measurements when the largest fractures are longer than the outlines of the map boundary.

- The connectivity of the fracture network increased through time. The contribution of different generations of fractures to the network connectivity depends on their number and orientation relative to pre-existing fractures.

- The network topology and connectivity in this area is strongly influenced by the last generation and varies between domains, in which the greatest connectivity was observed in the SW and the smallest in the NE.

*Code availability.* The code used for automatic fracture tracing is published in Prabhakaran, R., Bruna, P.-O., Bertotti, G. and Smeulders, D.: An automated fracture trace detection technique using the complex shearlet

transform, Solid Earth, 10(6), 2137–2166, doi:10.5194/se-10-2137-2019, 2019. The code is available on Github https://github.com/rahulprabhakaran/Automatic-Fracture-Detection-Code/tree/v1.0.0 (last access: 30 March 2020; see https://doi.org/10.5281/zenodo.3245452).

*Data availability.* Shapefiles of the fracture traces presented in this manuscript are provided in the supplement.

The image files are published under DOI: 10.18154/RWTH-2020-06903.

*Author contribution.* All authors contributed to the discussion and commented on the manuscript. CW acquired, processed, and evaluated the data and has written the manuscript with input of the co-authors. RP contributed the automatically-traced fractures and helped with the preparation of the manuscript. MP contributed to the manual

mapping of fractures and the initial interpretation of fracture generations. JLU gave the impulse to this work, provided the funding, discussed results at every stage of the project and helped to write the manuscript. GB contributed to the discussion and structure of the manuscript. KR provided input during the manuscript preparation.

*Competing interests.* The authors declare that they have no conflict of interest.

**Acknowledgements**

We would like to thank Roberto Emanuele Rizzo and William Dunne for their very constructive comments. We highly appreciate the time they have invested to carefully review this manuscript, helping us to improve it.

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

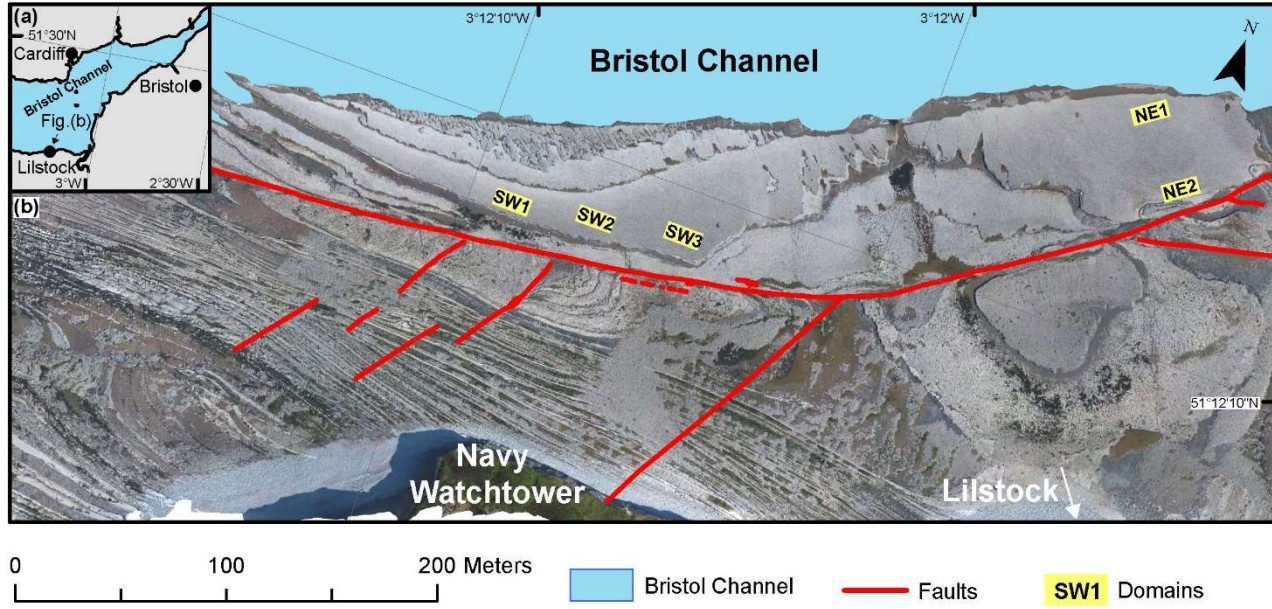

**Fig. 1: (a)** Location of the study area in the Bristol Channel, Great Britain. **(b** Ortho-rectified photo-mosaic of UAV-photographs taken from 100 m. The ortho-mosaic shows the coast at Lilstock during low tide, exposing the fractured limestone. The map domains on "the bench" are marked as yellow rectangles.


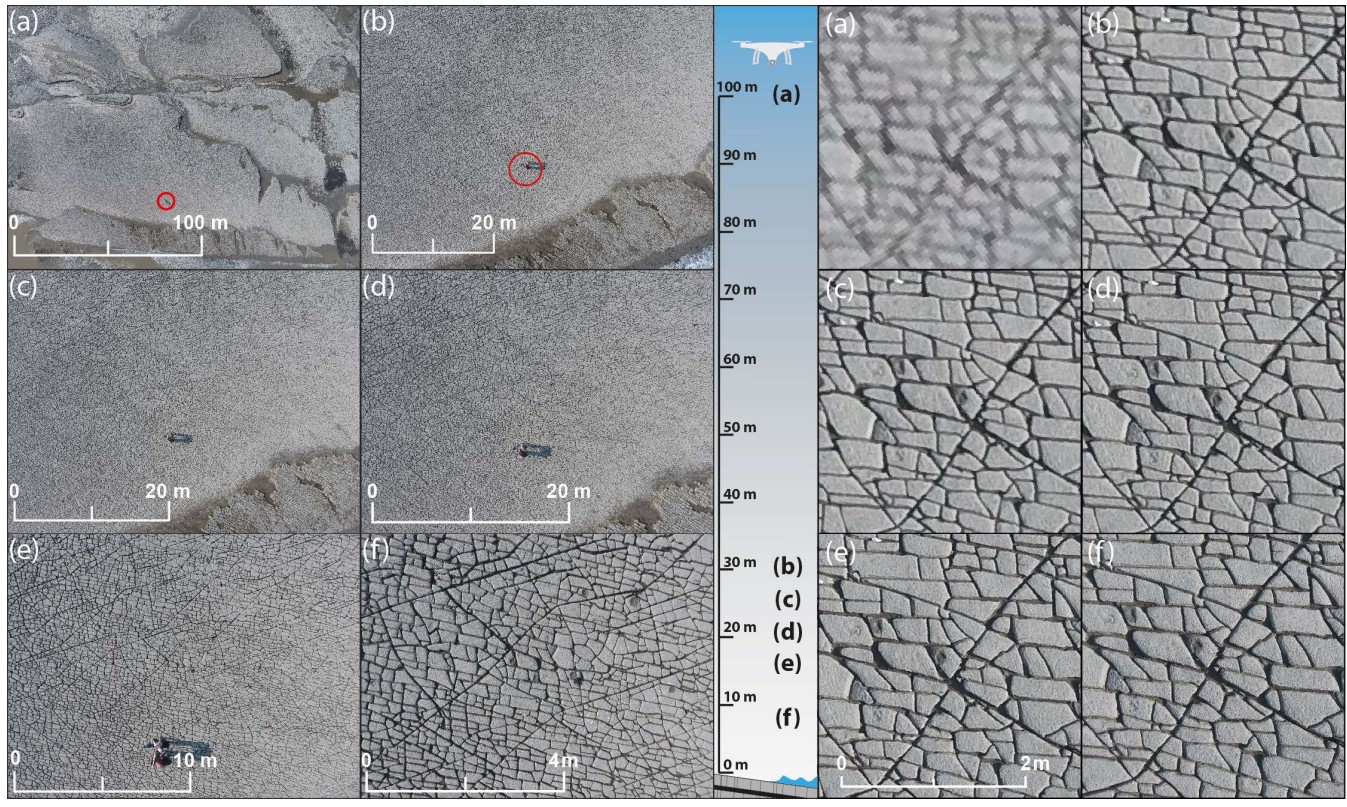

**Fig. 2:** The fractured limestone captured from above by UAV. Left: View from different altitudes with altitudes decreasing from (a) to (f). The exact altitudes from which the photographs have been taken can be viewed in the sketch in the center. Right: Same photographs as on the left, zoomed in to the same degree and location in every image. With increasing altitude, the spatial resolution of the image decreases. Note: persons for scale and encircled for better visibility in (a) and (b) on the left.


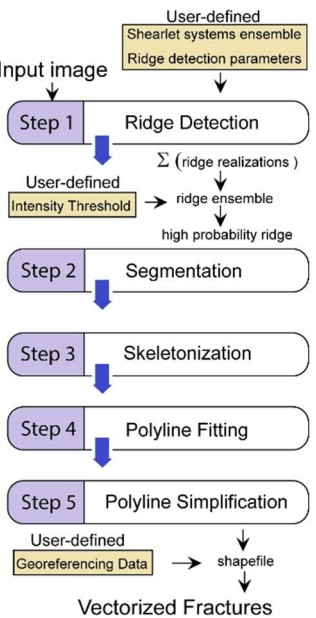


**Fig. 3: Overview of the automated fracture detection process (reproduced with permission from Prabhakaran et al, 2019)**

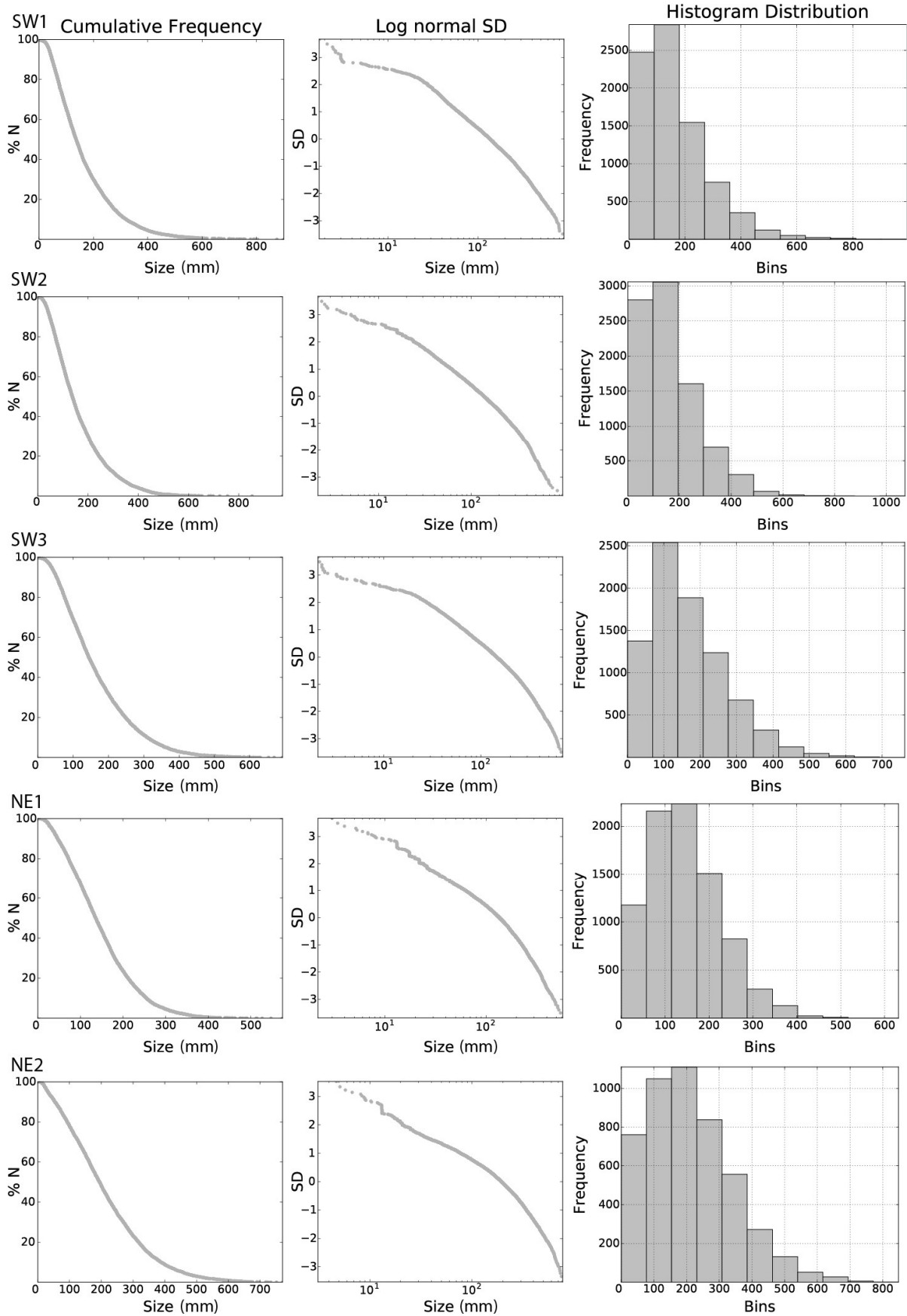

**Fig. 4: Cumulative length, log normal standard deviation and histogram distributions of the automatically-traced fractures (branches) in the five domains.**

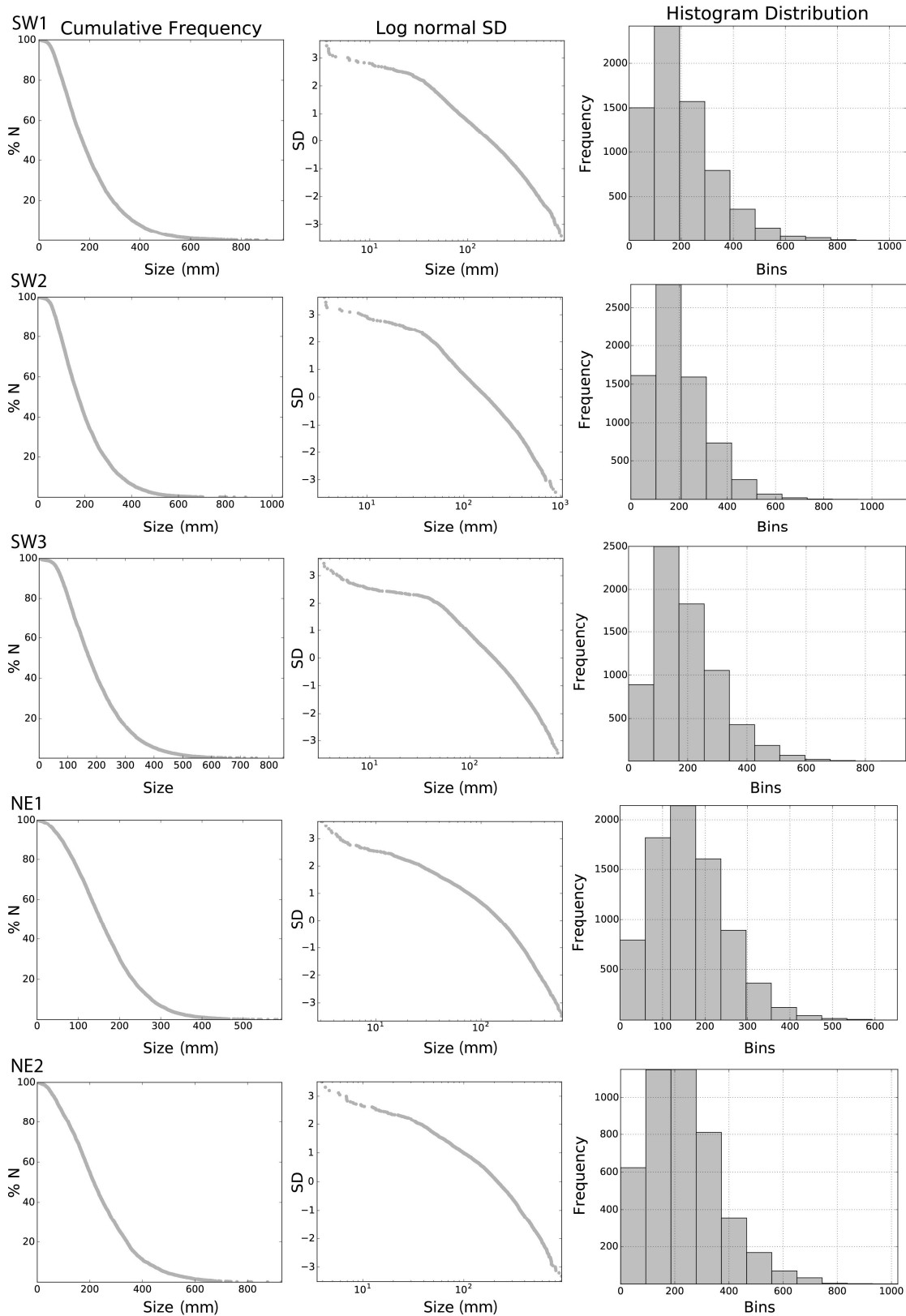

**Fig. 5: Cumulative length distribution, log normal standard deviation and histogram distributions of the manually mapped and segmented fracture traces (branches) in the five domains.**

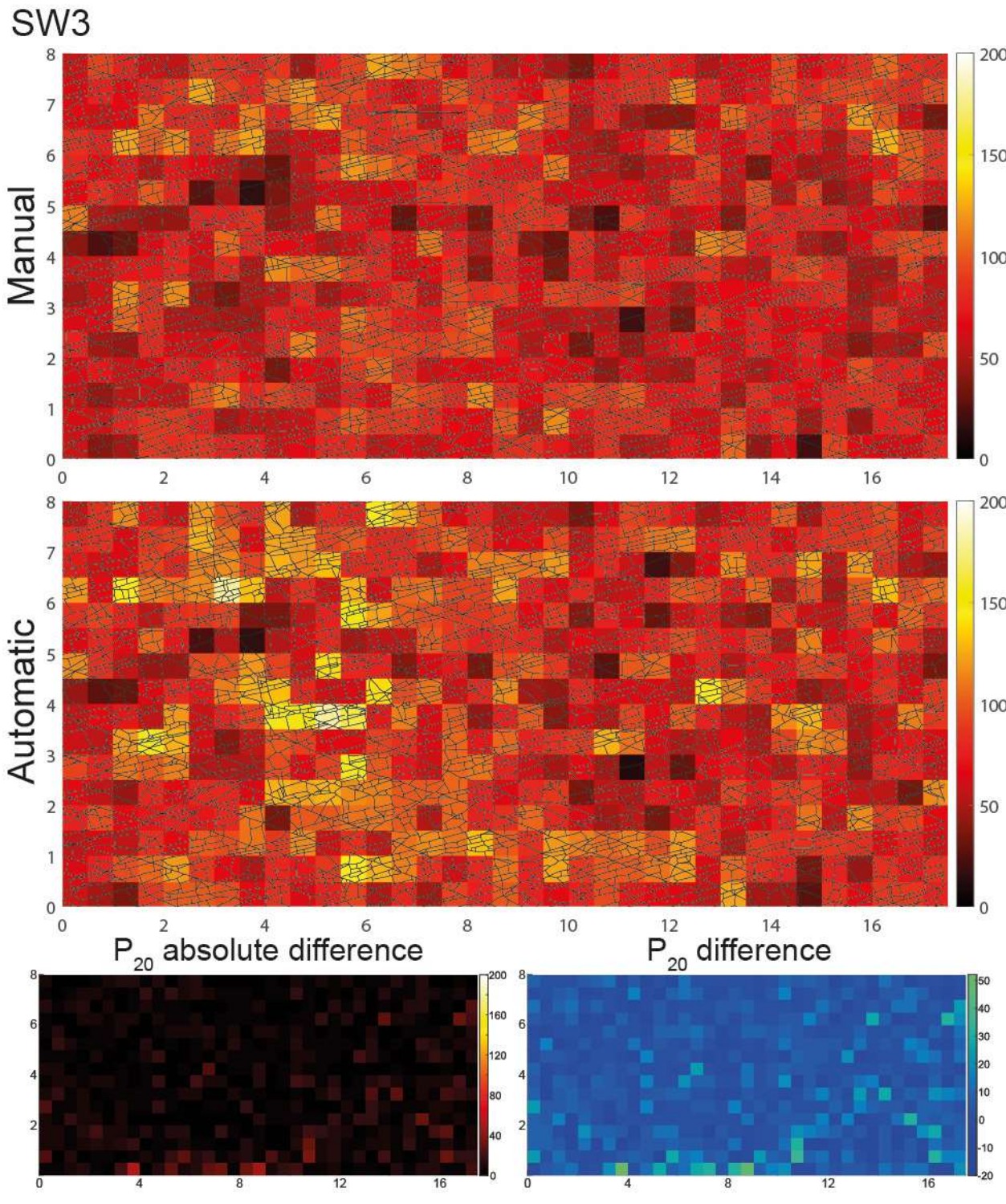

**Fig. 6: Comparison of the $P_{20}$ fracture segment density between manually an automatically-traced fracture segments for SW3. Absolute and relative differences are shown in the bottom line. Unit of the axes in (m), unit of the color bar in (1/m²).**

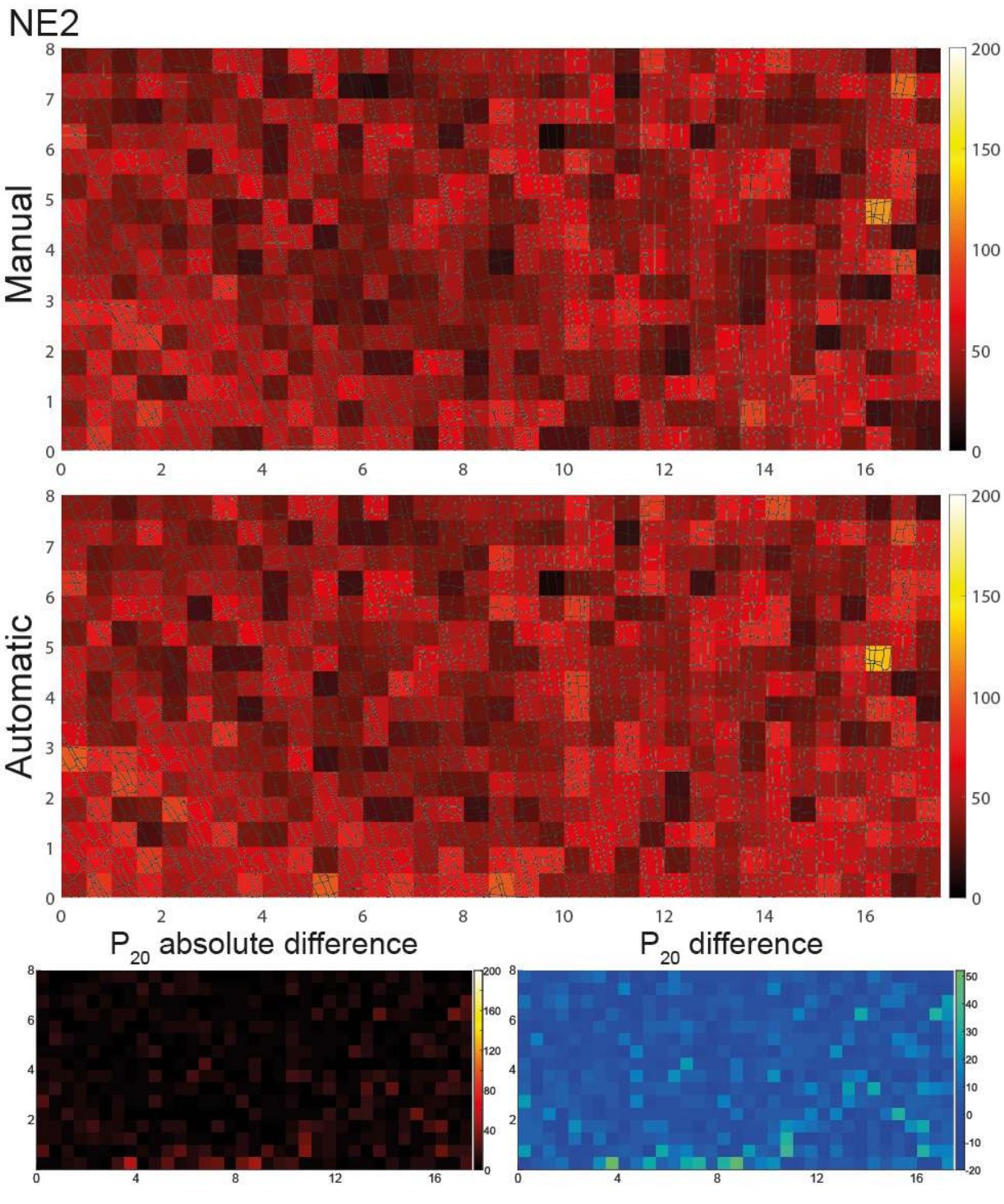


**Fig. 7: Comparison of the $P_{20}$ fracture segment density between manually an automatically-traced fracture segments for NE2. Absolute and relative differences are shown in the bottom line. Unit of the axes in (m), unit of the color bar in (1/m²).**

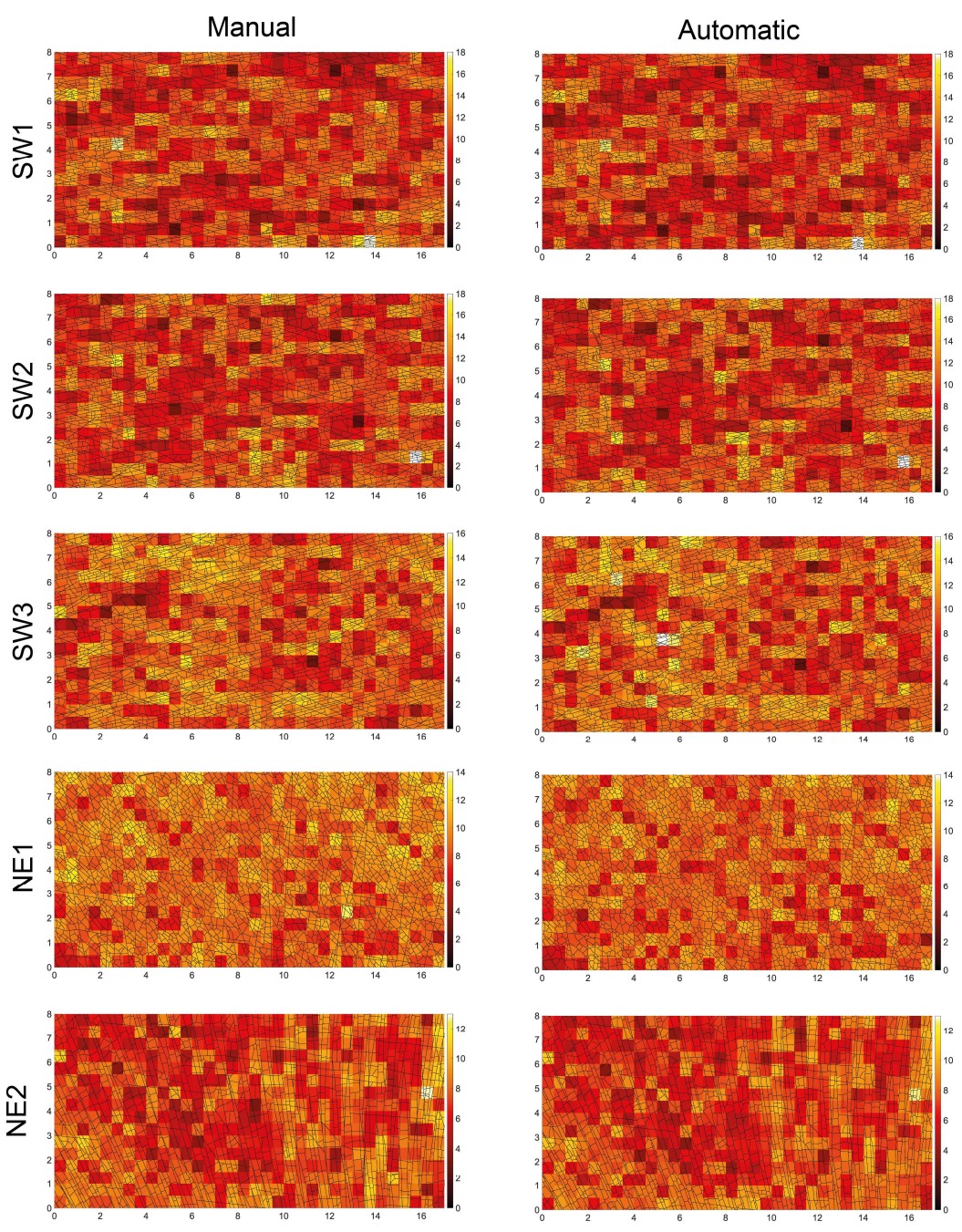


**Fig. 8: P$_{21}$ fracture intensity plots of the mapped fractures within the 5 subareas for the manual and automatic interpretations. Unit of the axes in (m), unit of the color bar in (m/m²), where lighter is more dense.**


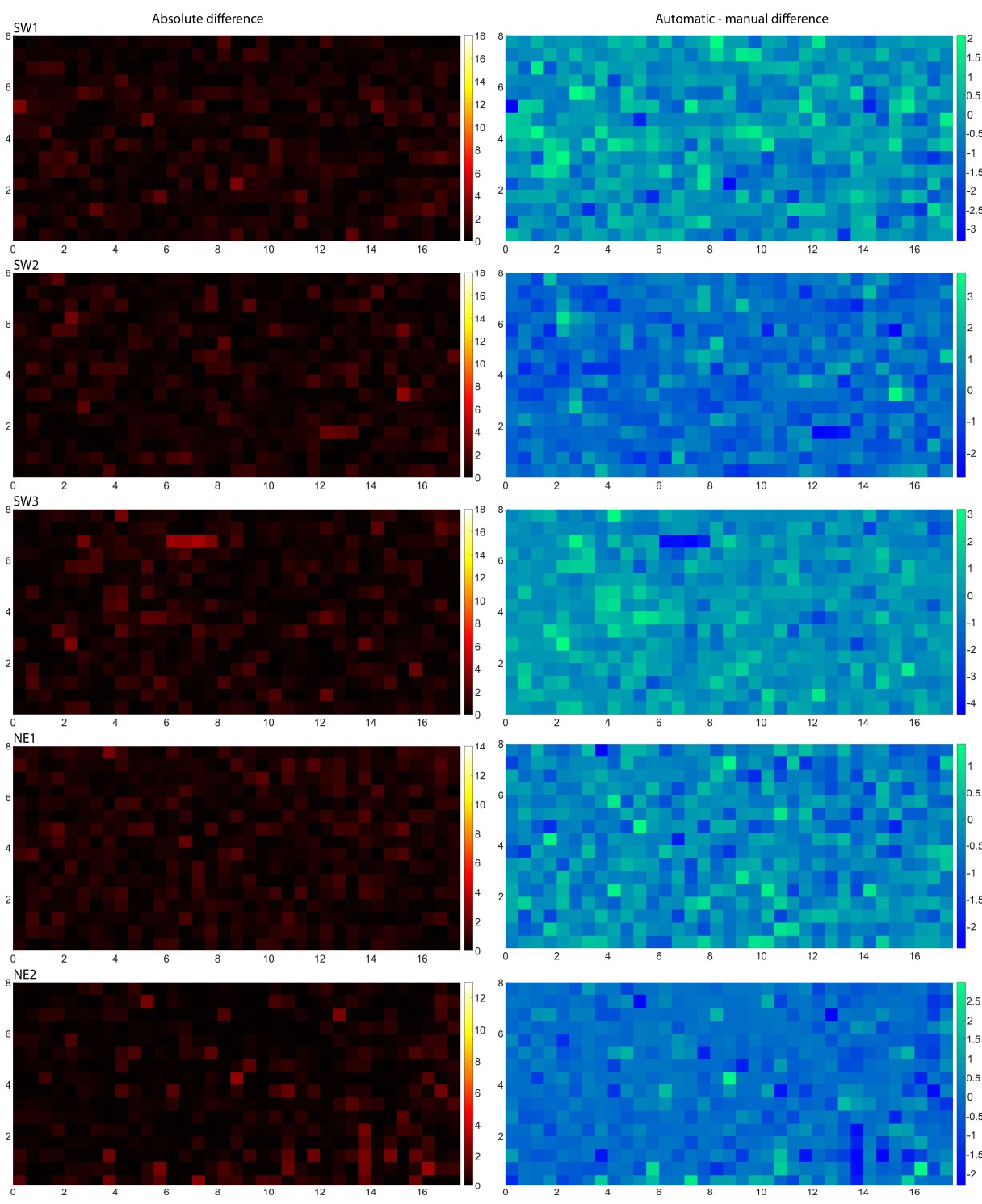

**Fig. 9: Differences of the $P_{21}$ fracture segment intensity between manually- and automatically-traced fracture segments for all five domains. Absolute differences are shown in the left column, relative differences in the right column. Unit of the axes in (m), unit of the color bar in (m/m²).**

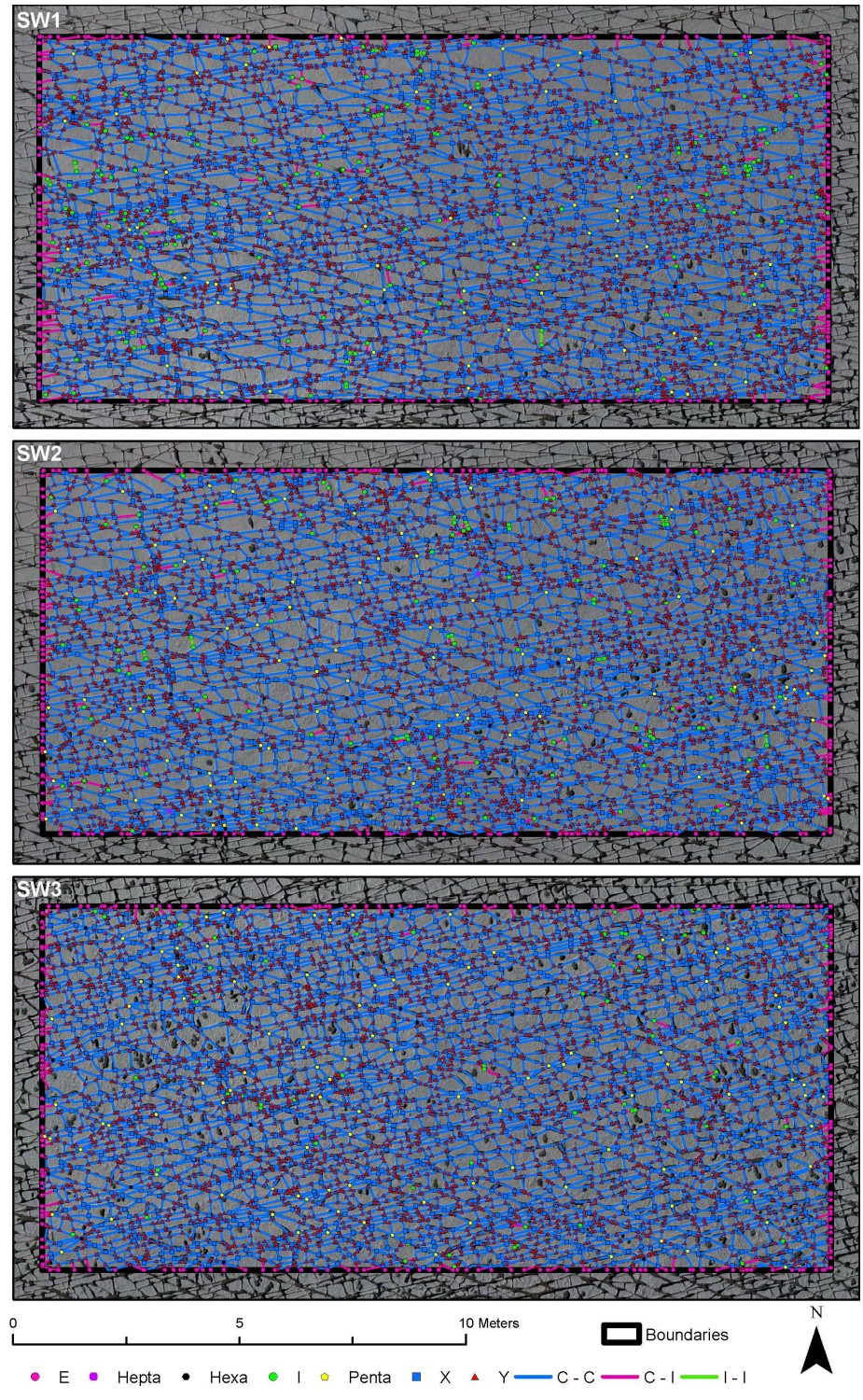


Fig. 10: SW fracture networks, automatically-traced and plotted as branches and nodes.

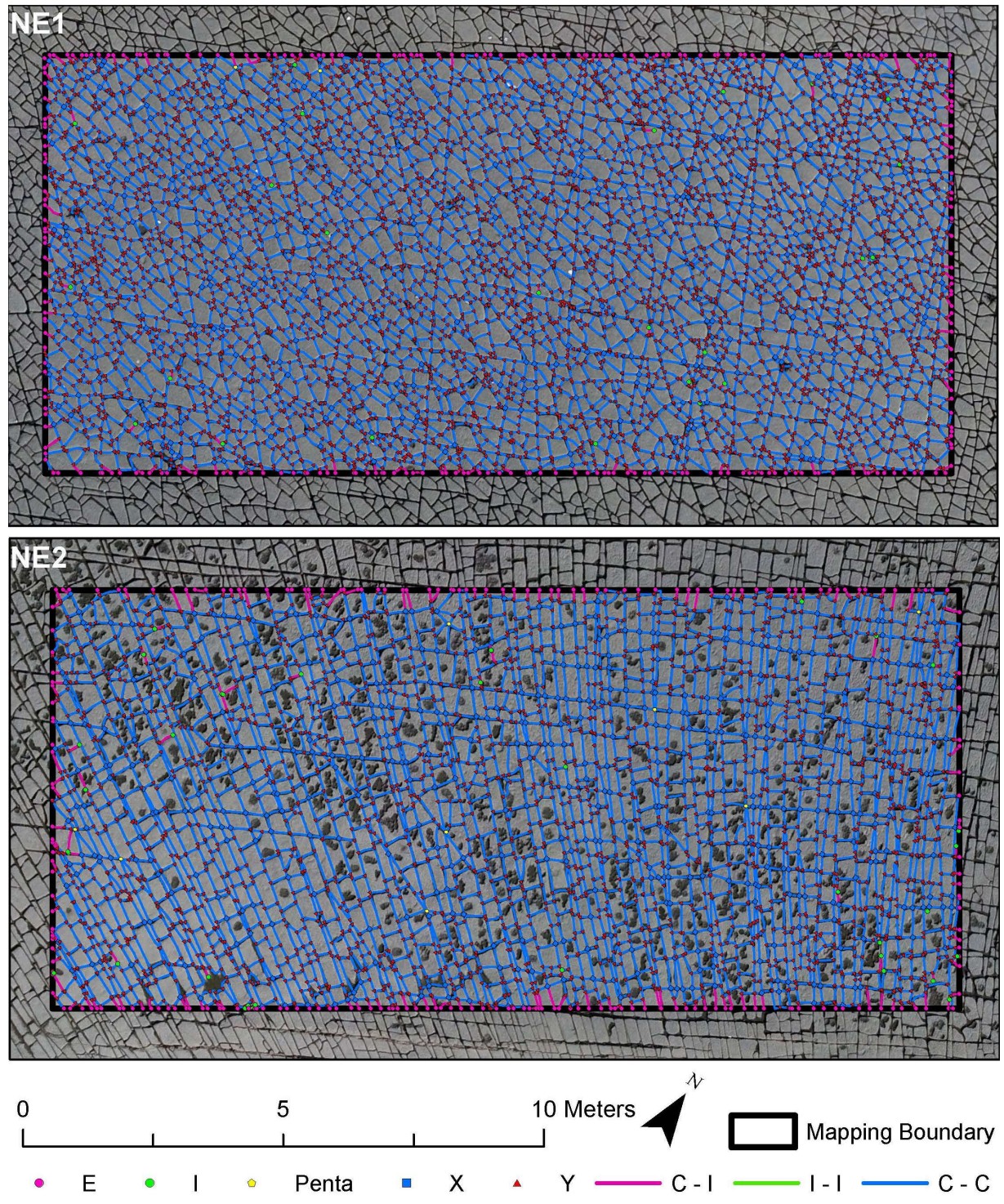

Fig. 11: NE fracture networks, automatically-traced and plotted as branches and nodes.

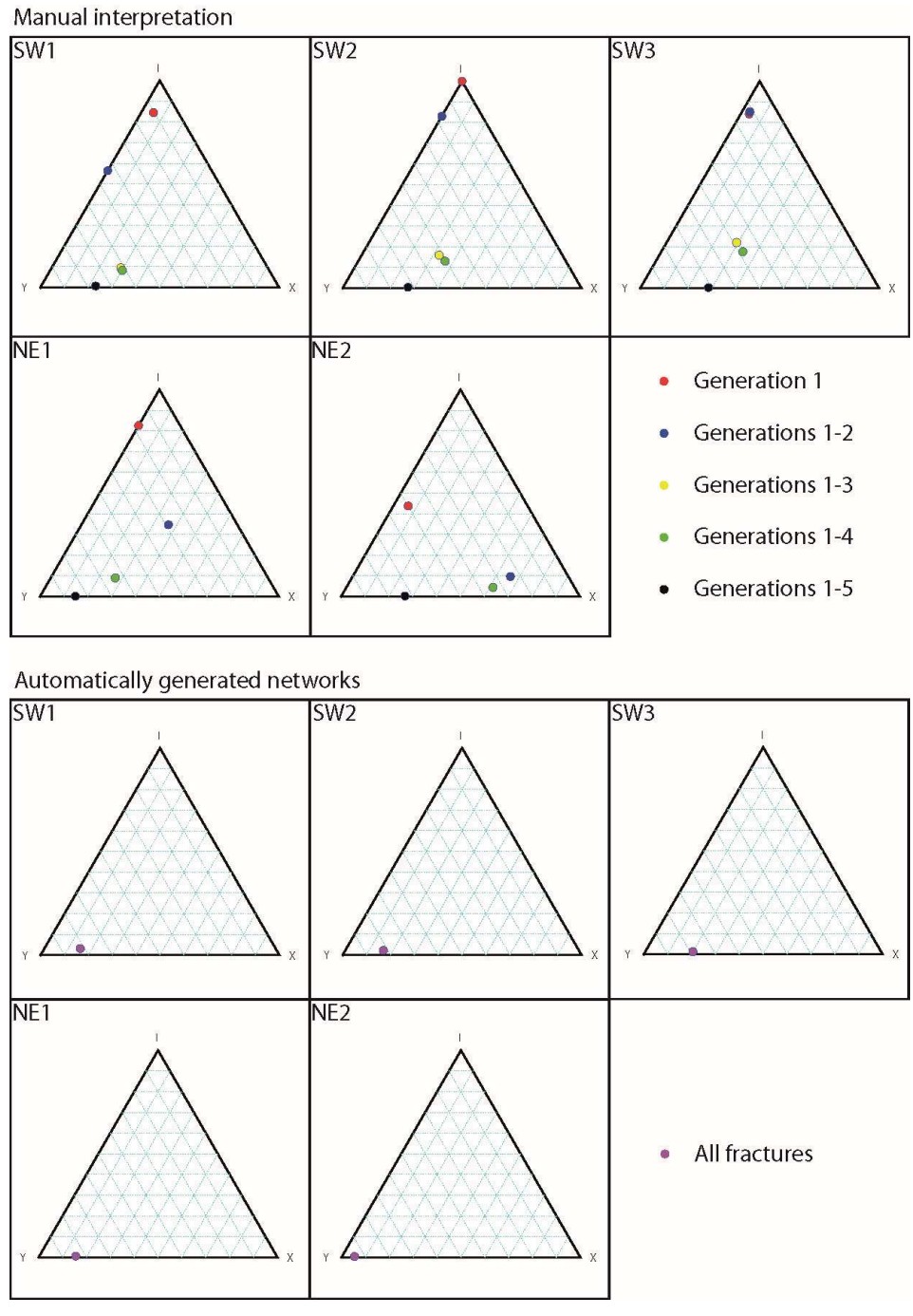

**Fig. 12: Ternary plots visualizing the node distributions. For the manual interpretations, the different network stages in terms of fracture generations are plotted. The automatic networks only show the final stage of the network, which corresponds to generations 1 – 5 in the manual interpretation. Nodes with four and more than four intersecting branches have been binned as type X nodes.**

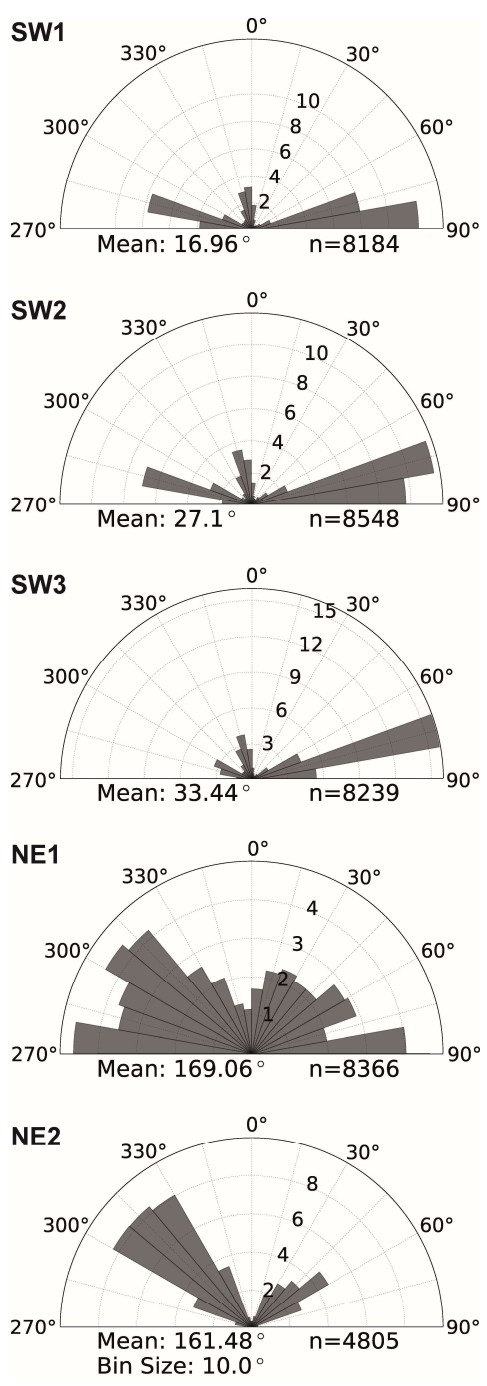


**Fig. 13: Length weighted rose plots of automatically extracted fracture trace segments (branches) in the five domains.**

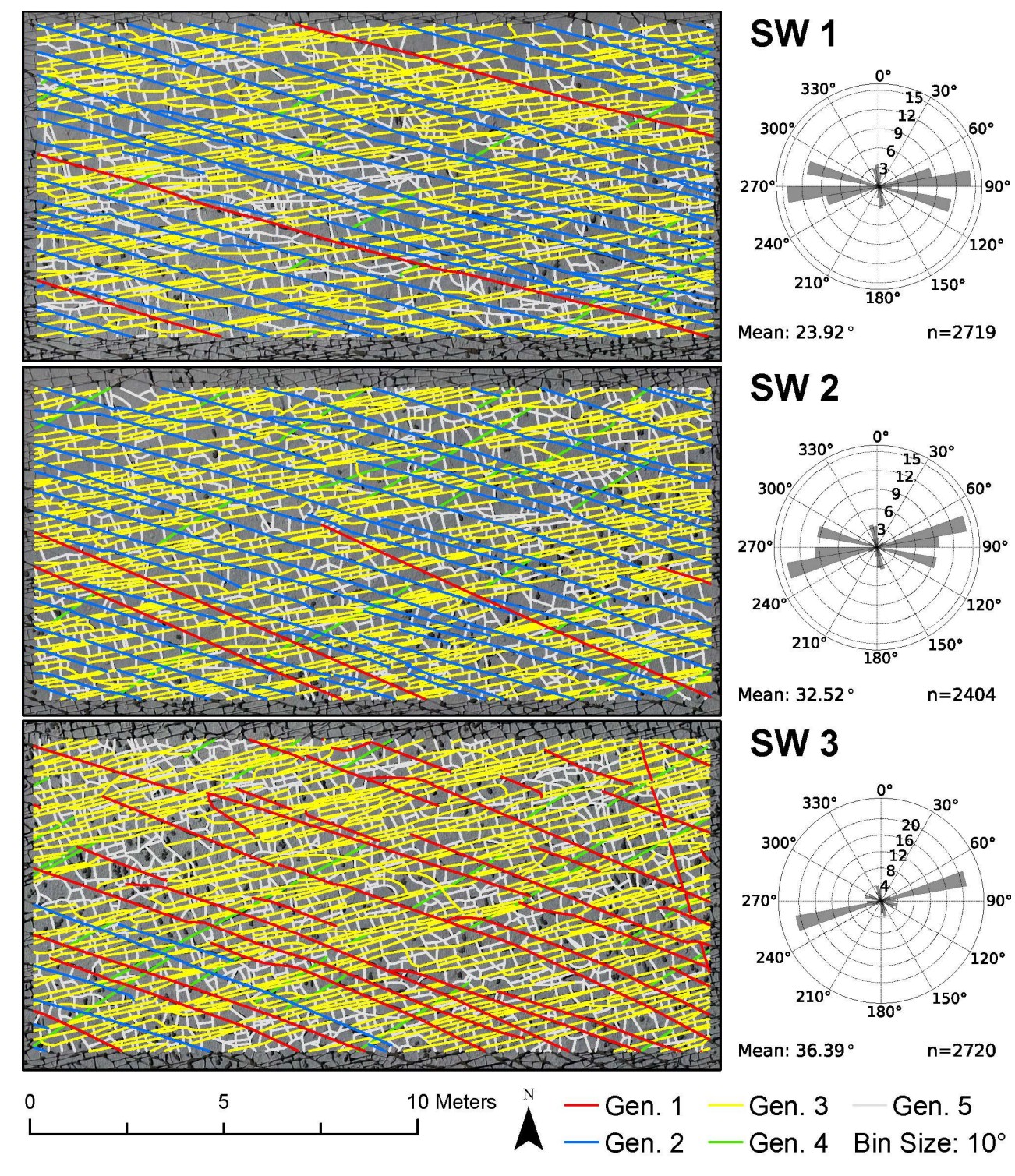

Fig. 14: Interpretation of fracture generations in manually interpreted data in the domains in the SW. Length-weighted rose diagrams showing fracture abundance as a function of the trends of fracture traces.


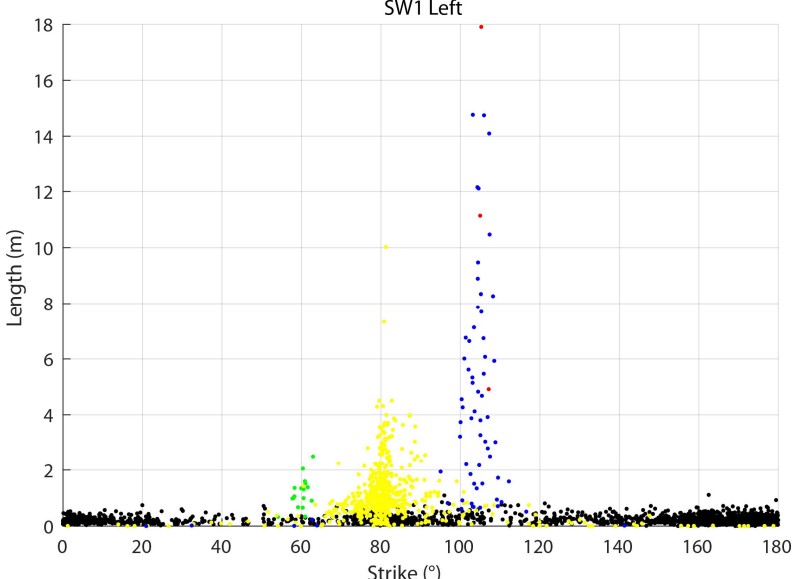

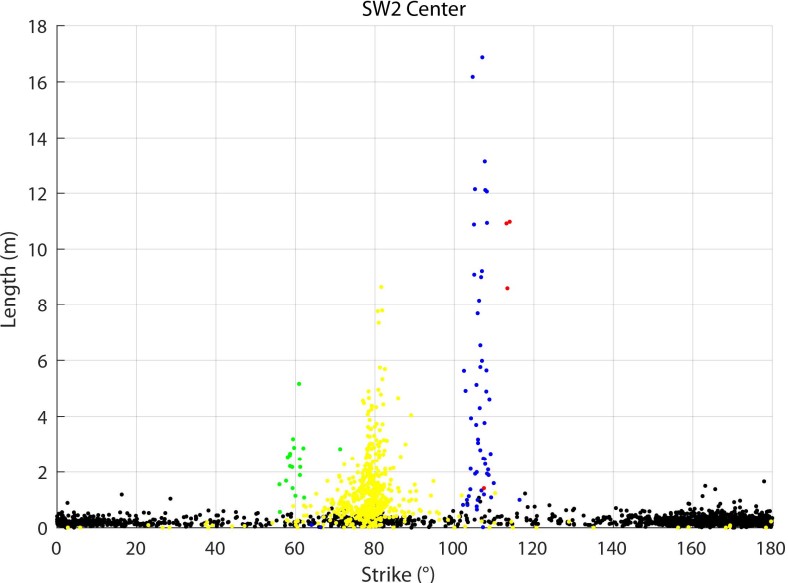

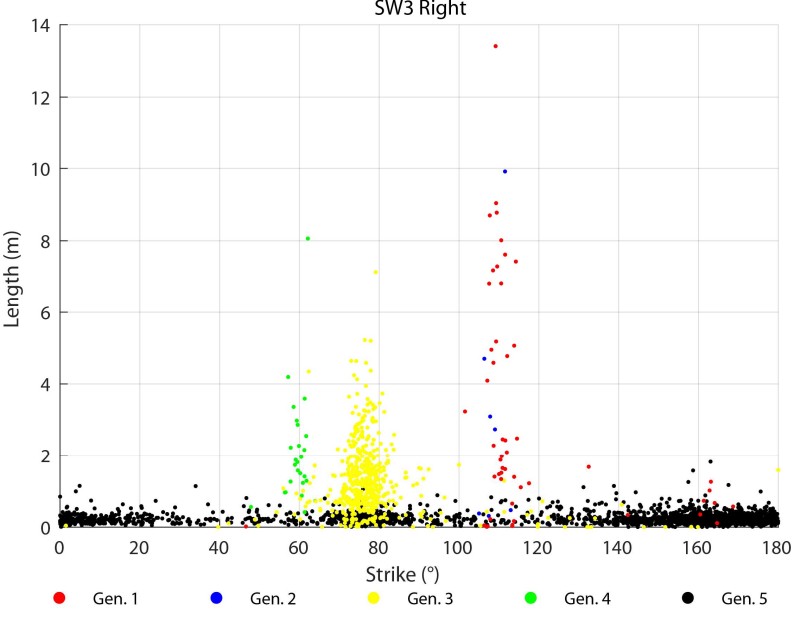

**Fig. 15: Plots of strike direction (°) on the x-axis and length (m) on the y-axis for every fracture mapped in the domains in the SW, color coded by generation.**

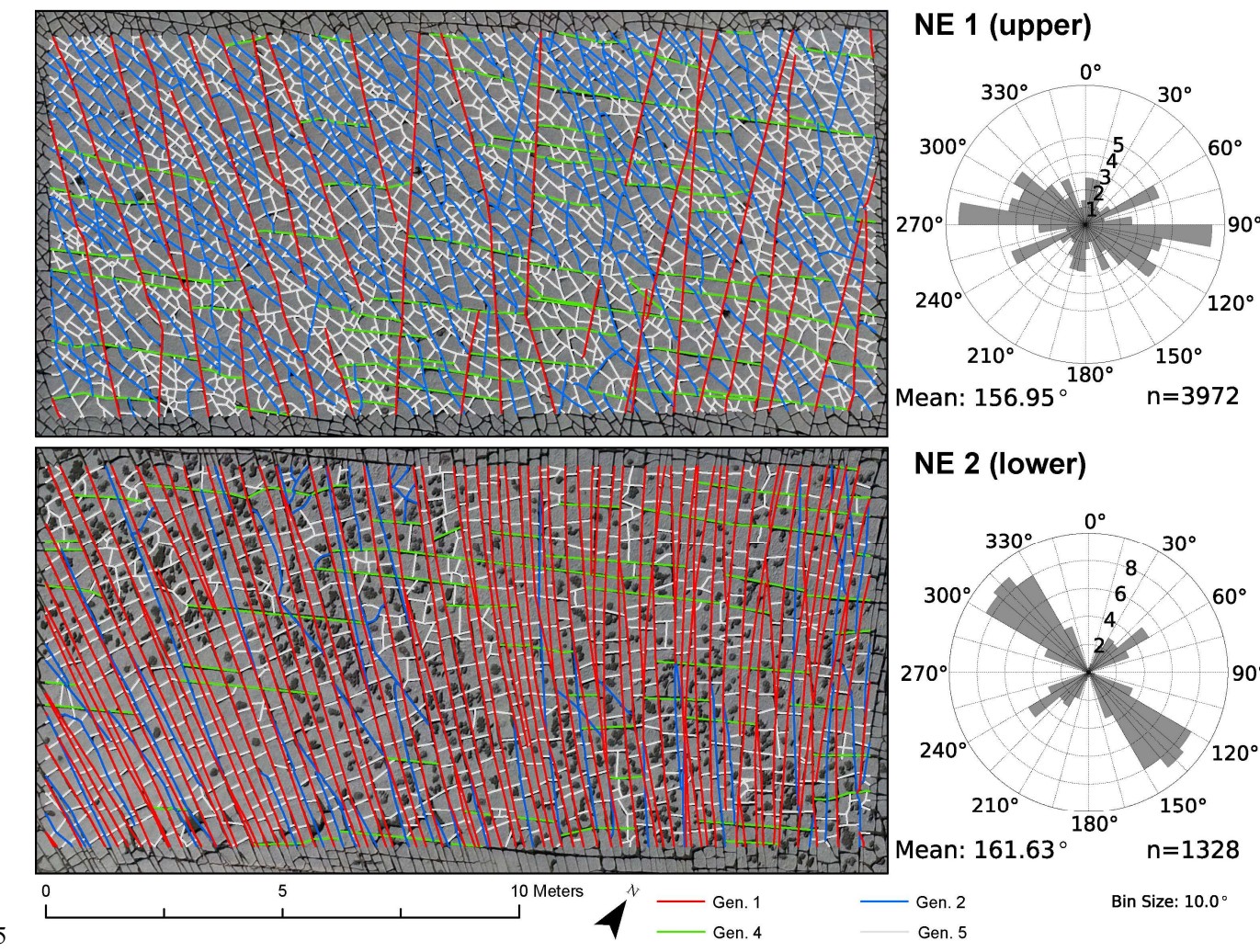

**Fig. 16: Interpretation of fracture generations in manually interpreted data in the domains in the NE. Length-weighted rose diagrams showing fracture abundance as a function of the trends of fracture traces. Note: the map domains have a NW (top) – SW (bottom) orientation as indicated by the north arrow.**

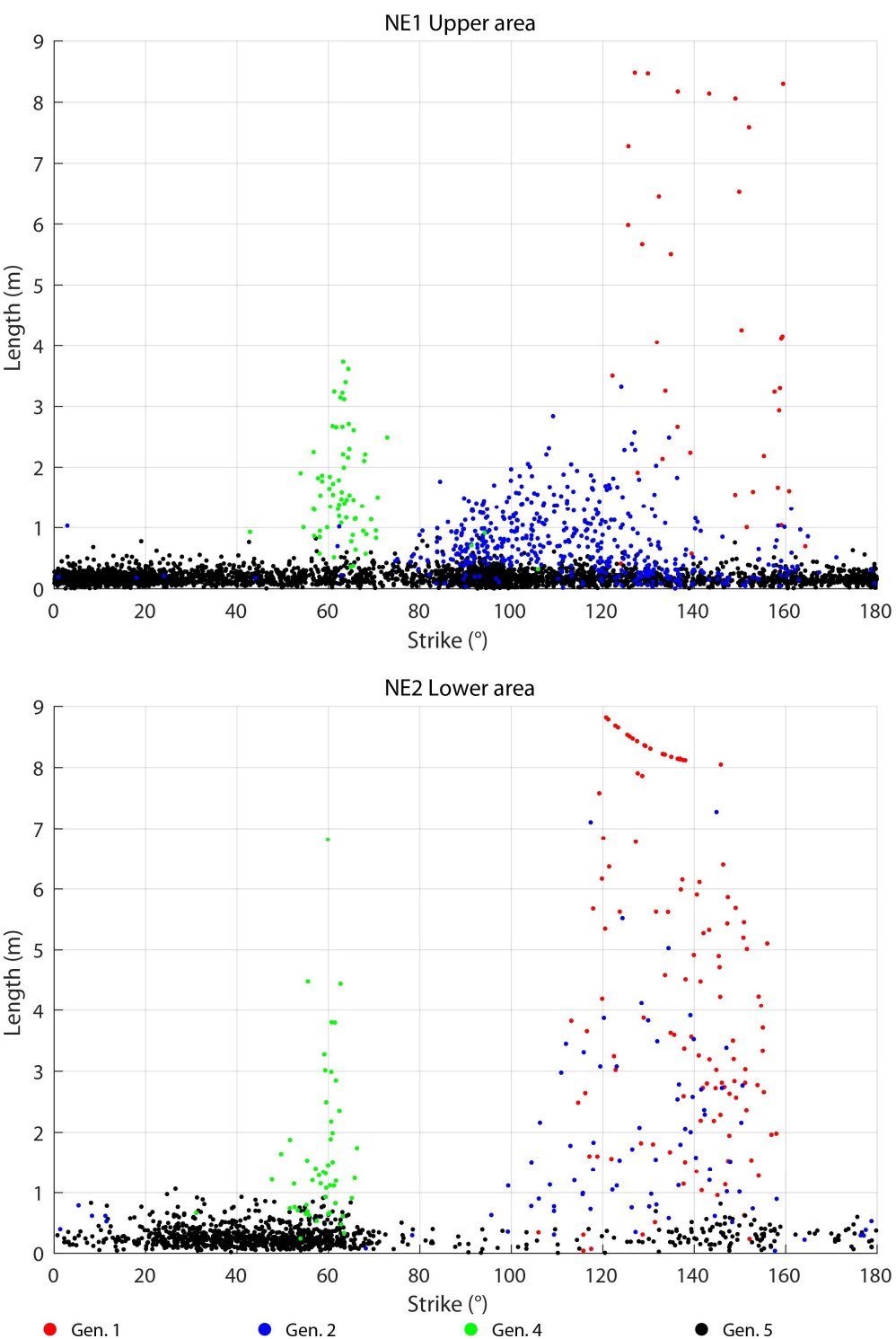

**Fig. 17: Plots of strike direction (°) on the x-axis and length (m) on the y-axis for every fracture mapped in the domains in the NE, color coded by generation.**

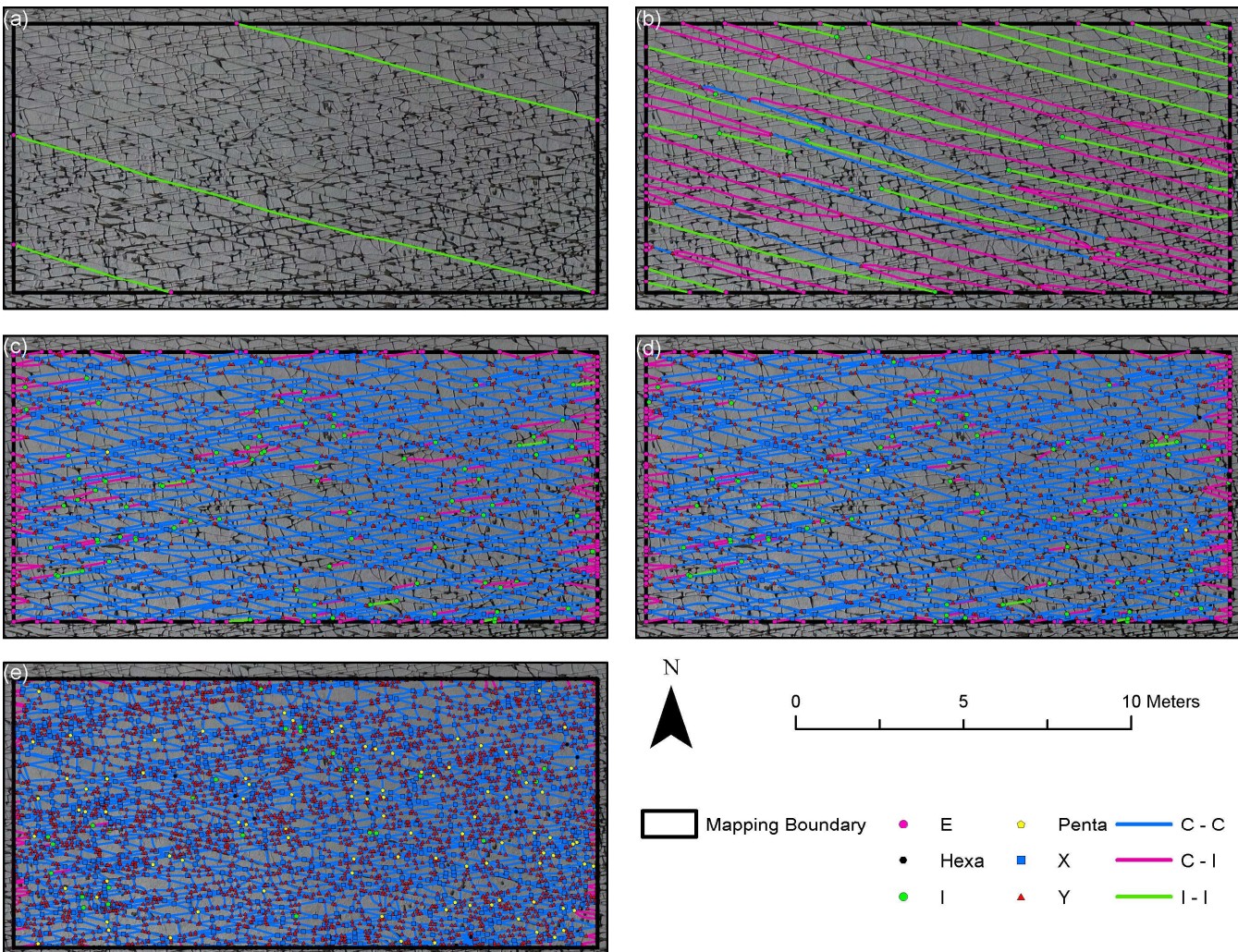

**Fig. 18:** Development of the fracture network in time steps for SW1 from generation 1 to 5, visualized as branches and nodes. The consecutive fracture generation is added for each panel from (a) to (e).


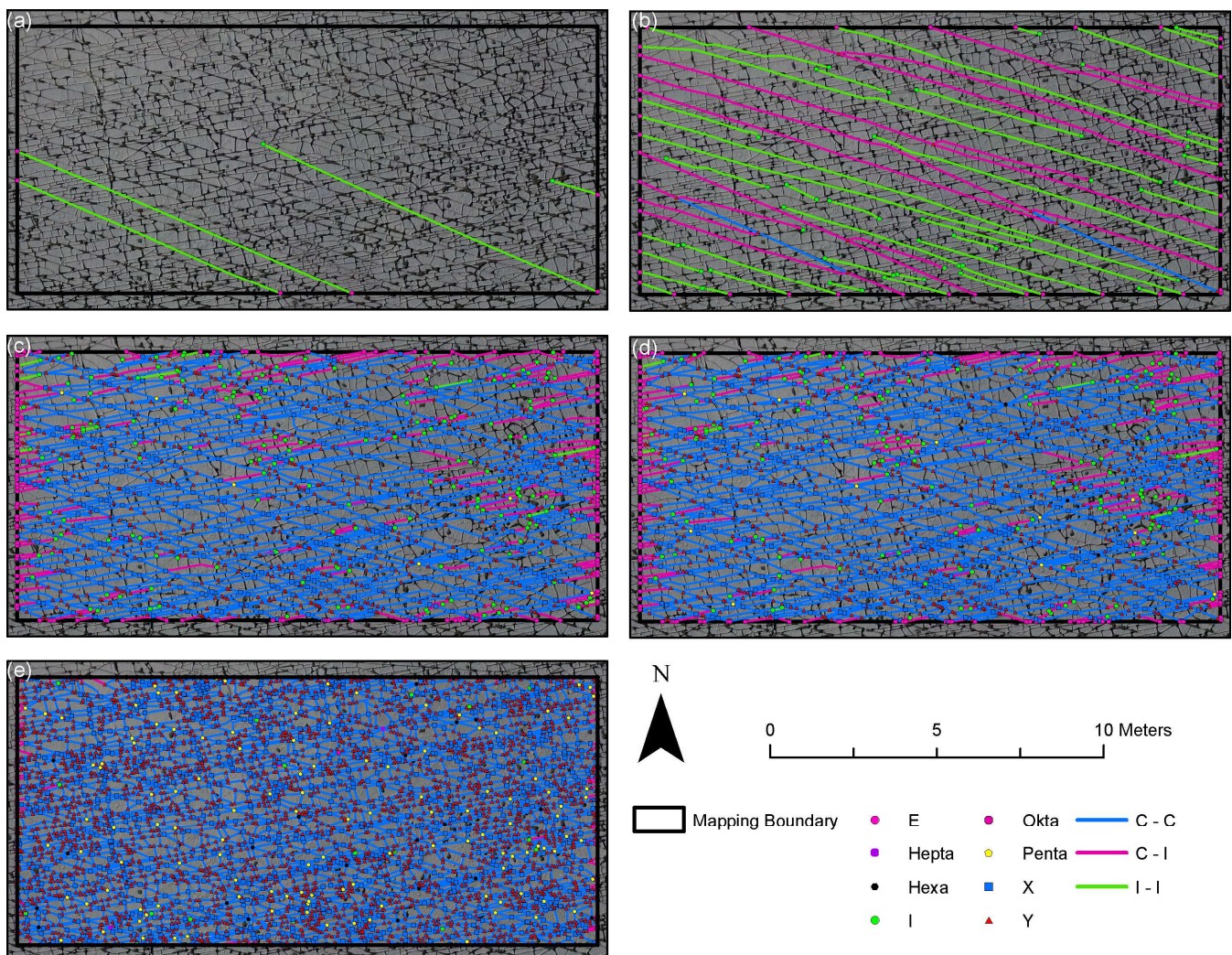

**Fig. 19: Development of the fracture network in time steps for SW2 from generation 1 to 5, visualized as branches and nodes. The consecutive fracture generation is added for each panel from (a) to €.**

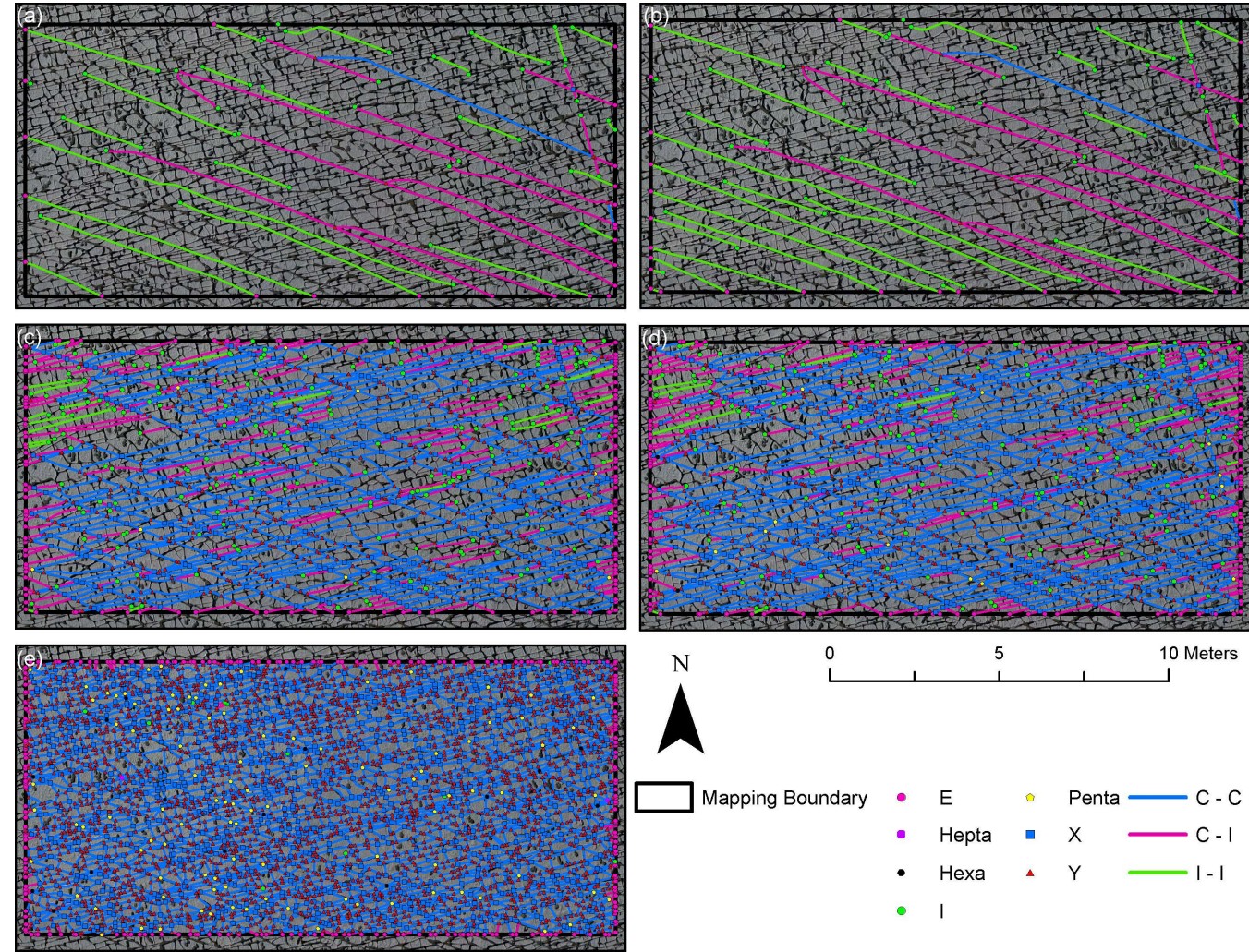


**Fig. 20: Development of the fracture network in time steps for SW3 from generation 1 to 5, visualized as branches and nodes. The consecutive fracture generation is added for each panel from (a) to (e).**

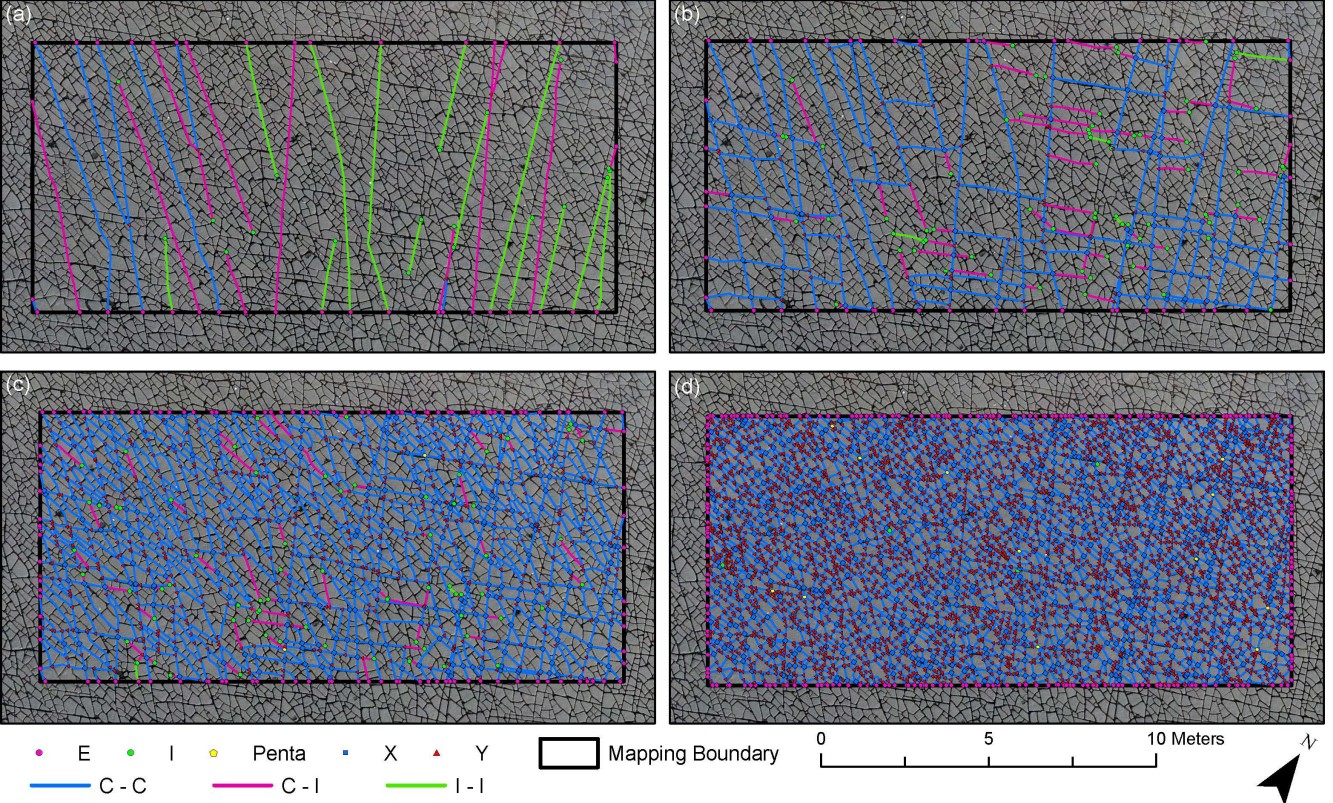

Fig. 21: Development of the fracture network in time steps for NE1 from generation 1 to 5, visualized as branches and nodes. The consecutive fracture generation is added for each panel from (a) to (d). Note that no fractures belonging to gen. 3 were identified in this domain.


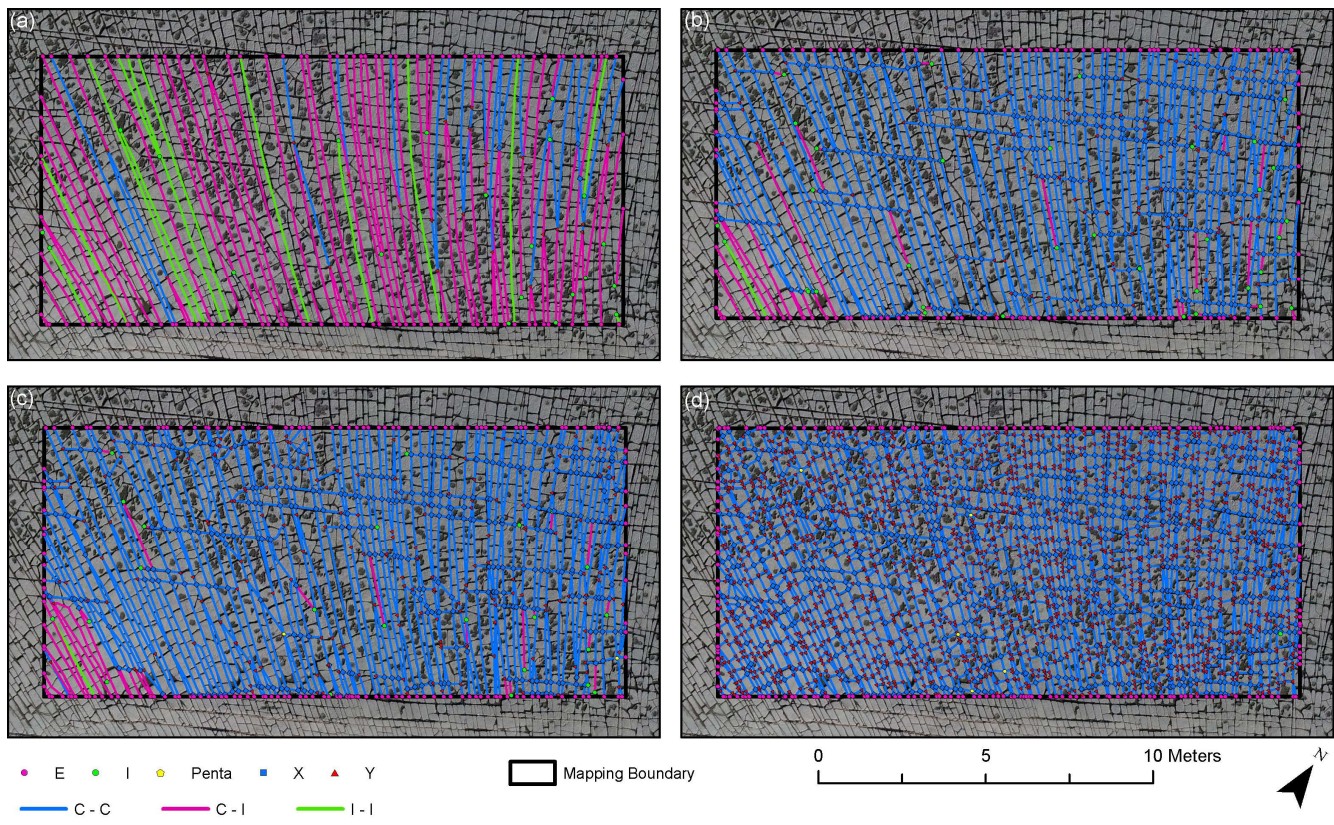

**Fig. 22: Development of the fracture network in time steps for NE2 from generation 1 to 5, visualized as branches and nodes. The consecutive fracture generation is added for each panel from (a) to (d). Note that no fractures belonging to gen. 3 were identified in this domain.**

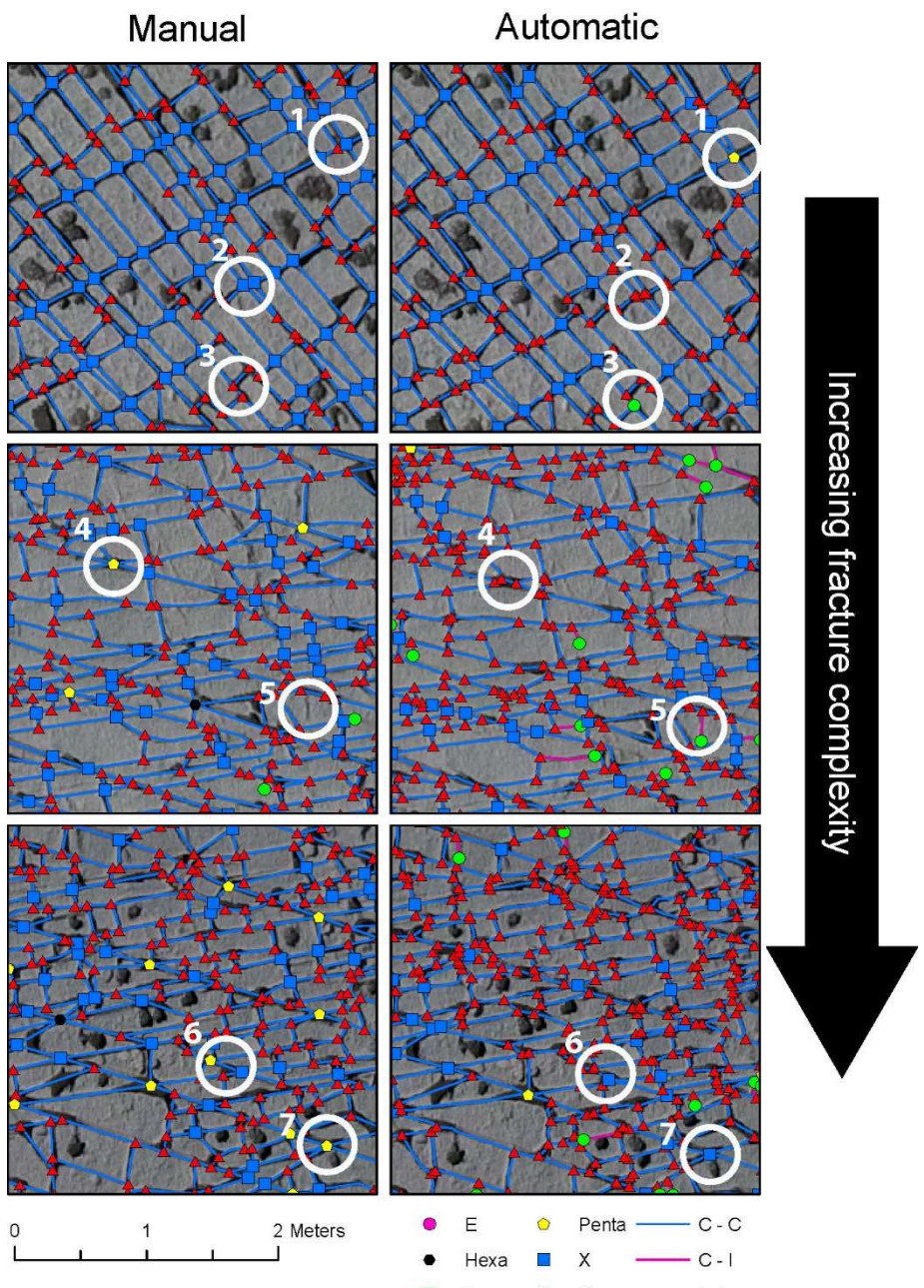


**Fig. 23: Comparison of the resulting network analysis from manual (left) and automatic (right) mapping. The complexity of the fracture patterns increases top to bottom; therefore, the quality of the results is assumed to decrease. Marks 1 - 6 show differences of the manual and automatic interpretation. 1) Two closely spaced Y and X nodes (manual) or one Penta node (automatic); 2) X nodes (manual) or closely spaced Y nodes (automatic); 3) Abutting fracture (manual) or I node (automatic); 4) One Penta node (manual) or three Y nodes (automatic); 5) Interpretation as surface erosion or a small fracture; 6) Penta and X nodes (manual) or X and Y nodes (automatic); 7) Interpretation as two close fractures resulting in a Penta node (manual) or as one fracture resulting in a X node.**


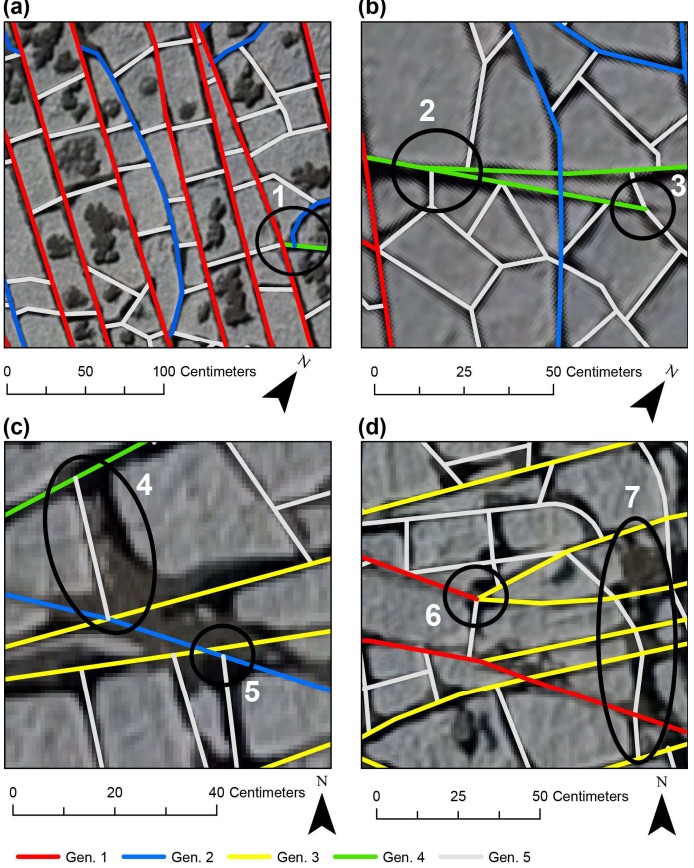

**Fig. 24: Details of the manual interpretation of fracture generations showing examples that allow for different interpretations of the generation and network geometry. (a) The fracture in the center of the image was assigned to gen. 2 (blue) because it abuts twice at the same gen.1, fracture, which is atypical for fractures of gen. 2 and was decided by elimination of the other interpreted generations. Mark 1 shows an interpretation where gen.2 appears to abut on gen. 4, which then abuts on gen. 1 towards the left. This is an example in which the initial tracing of the network was unclear because of the widely eroded intersection of the fractures and has been revised during the assignment of generations. (b) Mark 2 shows a splaying gen. 4 fracture, which is a rare case and cold also be interpreted as a younger fracture abutting on gen. 4. Mark 3 shows the tip of gen.4 which apparently abuts on gen. 5. The small darker in front of the tip suggests, that the fracture may continue, or the surface of the rock has been eroded in that place or is wet, in which case the interpreter decided to end the trace. (c) Mark 4 shows a widely eroded fracture which has been traced along one of the edges instead along its median axis. The reasoning behind this decision is the assumption, that the left edge sharper and straighter than the one on the right, which suggests a stronger erosion in that part. Therefore, the left edge was interpreted as the best representation of the fracture geometry in this case. Mark 5 shows a gen. 5 fracture abutting on gen.2. However, the wide erosion of the fracture itself and the junction of gen. 2 and gen. 3 towards the left, also allow the interpretation as a triple junction of generations 2, 3 and 5. (d) Mark 6 shows a junction where the gen.1 fracture was interpreted to stop, while the trace continues as gen.3 which initially follows the rough strike direction of gen. 1 and then bends towards the strike direction of gen. 3. Mark 7 shows another strongly eroded area that hinders the interpretation of fractures.**

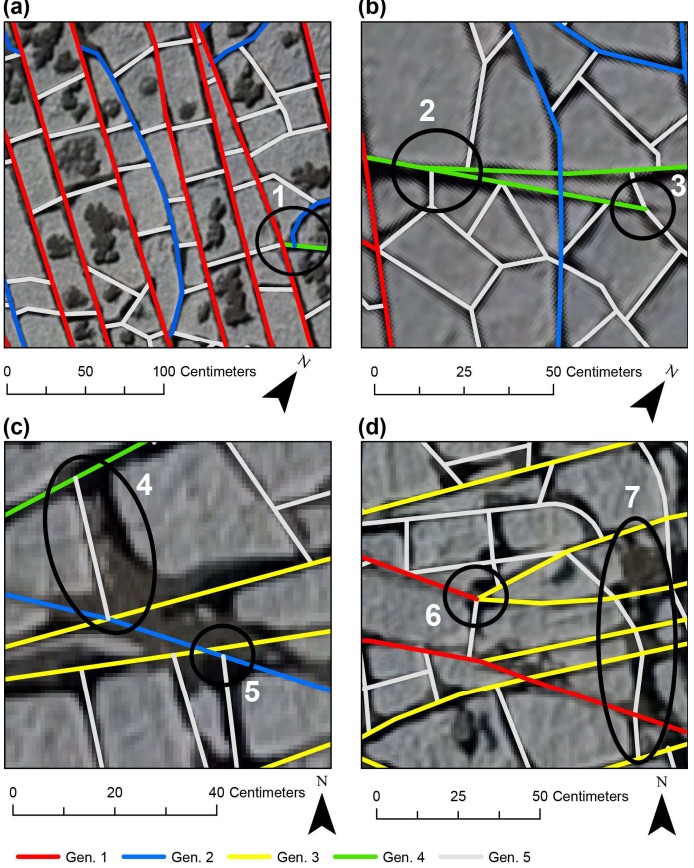

**Table 1: Statistics and results of the $P_{ij}$ analyses of the automatically and manually-traced fracture networks.**

| | Automatic | | | | | Manual | | | | |
|---|---|---|---|---|---|---|---|---|---|---|
| | SW 1 | SW 2 | SW 3 | NE 1 | NE 2 | SW 1 | SW 2 | SW 3 | NE 1 | NE 2 |
| Branches | 8184 | 8548 | 8239 | 8366 | 4831 | 6888 | 7094 | 6975 | 7789 | 4366 |
| $P_{21}$ (m$^{-1}$) | 9.7 | 10.01 | 9.84 | 8.75 | 7.31 | 9.8 | 10 | 9.7 | 9 | 7.3 |
| $P_{20}$ (m$^{-2}$) | 58.5 | 61.1 | 58.9 | 59.8 | 34.1 | 49.1 | 50.6 | 49.8 | 55.6 | 30.9 |
| Mean (m) | 0.16 | 0.16 | 0.17 | 0.15 | 0.21 | 0.2 | 0.2 | 0.2 | 0.16 | 0.24 |
| Geom. mean (m) | 0.13 | 0.13 | 0.13 | 0.12 | 0.17 | 0.16 | 0.17 | 0.17 | 0.14 | 0.19 |
| Standard deviation | 0.12 | 0.11 | 0.1 | 0.08 | 0.13 | 0.13 | 0.12 | 0.11 | 0.86 | 0.14 |
| 25% | 0.08 | 0.08 | 0.09 | 0.08 | 0.11 | 0.1 | 0.11 | 0.11 | 0.1 | 0.14 |
| 50% | 0.14 | 0.14 | 0.14 | 0.14 | 0.19 | 0.17 | 0.17 | 0.17 | 0.15 | 0.22 |
| 75% | 0.22 | 0.22 | 0.23 | 0.2 | 0.29 | 0.26 | 0.26 | 0.26 | 0.22 | 0.32 |
| Min. | < 1 cm | < 1 cm | < 1 cm | < 1 cm | < 1 cm | < 1 cm | < 1 cm | < 1 cm | < 1 cm | < 1 cm |
| Max. (m) | 0.9 | 0.97 | 0.69 | 0.57 | 0.77 | 0.97 | 1.05 | 0.85 | 0.59 | 0.93 |
| Covariance | 0.7 | 0.67 | 0.62 | 0.56 | 0.62 | 0.64 | 0.59 | 0.56 | 0.53 | 0.57 |
| Skewness | 1.46 | 1.21 | 1.01 | 0.74 | 0.77 | 1.31 | 1.21 | 1.06 | 0.72 | 0.88 |
| Kurtosis | 2.99 | 1.82 | 1.07 | 0.55 | 0.48 | 2.33 | 2.03 | 1.42 | 0.72 | 1.01 |

**Table 2: Nodes generated from the automatically-traced fracture networks.**

| | SW 1 | SW 2 | SW 3 | NE 1 | NE 2 |
|---|---|---|---|---|---|
| E | 248 | 262 | 268 | 236 | 279 |
| I | 166 | 117 | 62 | 19 | 17 |
| Y | 4250 | 4390 | 3979 | 5100 | 2517 |
| X | 713 | 757 | 885 | 282 | 448 |
| Penta | 62 | 87 | 103 | 2 | 8 |
| Hexa | 3 | 9 | 17 | | |
| Hepta | | 2 | | | |

**Table 3: Average (avg.) values of fracture length (m) and strike (°) for the domains in the SW.**

|      | Avg. length (m) | | | Avg. strike (°) | | |
| --- | --- | --- | --- | --- | --- | --- |
| Gen. | SW 1 | SW 2 | SW 3 | SW 1 | SW 2 | SW 3 |
| 1 | 3.06 | 7.98 | 3.06 | 106 | 112 | 117 |
| 2 | 4.31 | 4.67 | 3.09 | 101 | 105 | 109 |
| 3 | 1.12 | 1.25 | 1.30 | 83 | 79 | 78 |
| 4 | 1.17 | 2.27 | 2.09 | 60 | 60 | 59 |
| 5 | 0.22 | 0.24 | 0.24 | 114 | 118 | 121 |

**Table 4: Average values of fracture length (m) and strike (°) for the domains in the NE.**

|      | Avg. length (mm) | | Avg. strike (°) | |
| --- | --- | --- | --- | --- |
| Gen. | NE 1 | NE 2 | NE 1 | NE 2 |
| 1 | 4.33 | 4.13 | 144 | 137 |
| 2 | 1.64 | 1.62 | 111 | 123 |
| 4 | 1.77 | 0.77 | 64 | 58 |
| 5 | 0.28 | 0.19 | 76 | 55 |

**Table 5: Evolution of the fracture network in SW1.**

| Gen. | Avg. length (m) | I | Y | X | E | Penta | Hexa |
| --- | --- | --- | --- | --- | --- | --- | --- |
| 1 | 3.06 | 49 | 6 | 3 | 25 | 0 | 0 |
| 1 - 2 | 4.69 | 22 | 17 | 0 | 59 | 0 | 0 |
| 1 - 3 | 0.59 | 92 | 595 | 278 | 145 | 1 | 0 |
| 1 - 4 | 0.56 | 84 | 628 | 303 | 151 | 2 | 3 |
| 1 - 5 | 0.20 | 30 | 3194 | 849 | 250 | 96 | 9 |

**Table 6: Evolution of the fracture network in SW2.**

| Gen. | Avg. branch length (m) | I | Y | X | E | Penta | Hexa | Hepta | Okta |
|---|---|---|---|---|---|---|---|---|---|
| 1 | 7.98 | 2 | 0 | 0 | 6 | | | | |
| 1 - 2 | 4.51 | 49 | 10 | 0 | 49 | | | | |
| 1 - 3 | 0.54 | 178 | 577 | 357 | 163 | 6 | | | |
| 1 - 4 | 0.50 | 157 | 612 | 425 | 173 | 12 | 4 | | |
| 1 - 5 | 0.20 | 18 | 3057 | 1000 | 258 | 123 | 20 | 2 | 1 |

**Table 7: Evolution of the fracture network in SW3.**

| Gen. | Avg. branch length (m) | I | Y | X | E | Penta | Hexa | Hepta |
|---|---|---|---|---|---|---|---|---|
| 1 | 3.06 | 48 | 7 | 2 | 25 | 0 | 0 | 0 |
| 1 - 2 | 3.17 | 52 | 7 | 2 | 34 | 0 | 0 | 0 |
| 1 - 3 | 0.59 | 203 | 448 | 262 | 145 | 8 | 0 | 0 |
| 1 - 4 | 0.54 | 179 | 492 | 329 | 158 | 15 | 4 | 0 |
| 1 - 5 | 0.20 | 12 | 2953 | 1057 | 275 | 102 | 16 | 2 |

**Table 8: Evolution of the fracture network in NE1.**

| Gen. | Avg. branch length (m) | I | Y | X | E | Penta |
|---|---|---|---|---|---|---|
| 1 | 4.33 | 19 | 4 | | 41 | |
| 1 - 2 | 0.79 | 78 | 65 | 82 | 54 | |
| 1 - 4 | 0.68 | 77 | 547 | 229 | 124 | 2 |
| 1 - 5 | 0.24 | 8 | 4135 | 705 | 283 | 12 |

**Table 9: Evolution of the fracture network in NE2.**

| Gen. | Avg. branch length [m] | I | Y | X | E | Penta |
|---|---|---|---|---|---|---|
| 1 | 4.13 | 21 | 24 | 3 | 131 | |
| 1 - 2 | 0.80 | 38 | 95 | 259 | 145 | |
| 1 - 4 | 0.45 | 25 | 193 | 347 | 184 | 2 |
| 1 - 5 | 0.16 | 7 | 1888 | 681 | 230 | 5 |

**Table 10: Differences of the fracture networks from manual and automatic tracing.**

|                       | SW 1  | SW 2  | SW 3  | NE 1 | NE 2  |
|-----------------------|-------|-------|-------|------|-------|
| Avg. branch length (m)| 0.03  | 0.03  | 0.03  | 0.09 | -0.05 |
| I                     | -136  | -99   | -50   | -11  | -10   |
| Y                     | -1056 | -1333 | -1026 | -965 | -629  |
| X                     | 136   | 243   | 172   | 423  | 233   |
| Penta                 | 34    | 36    | -1    | 10   | -3    |
| Hexa                  | 6     | 11    | -1    | 0    | 0     |
| Hepta                 | 0     | 0     | 2     | 0    | 0     |
| Okta                  | 0     | 1     | 0     | 0    | 0     |