# Peer review of "Mapping the fracture network in the Lilstock pavement, Bristol Channel, UK: manual versus automatic"

_Solid Earth, 2020_

## Referee Comment (RC1) · Roberto Emanuele Rizzo (Referee) · 10 Jun 2020

General Comments: This paper presents a detailed analysis of the fracture network visible on the wave-cut pavement of the Bristol Channel Limestone sequence, in UK. The authors, focusing their study to one specify stratigraphic layer, called the 'bench', analyse the results obtained by using manual and automatic extraction of fracture traces from drone photomontages. The results of the two methods are then compared and used to describe the network in terms of connectivity and deformation history that generated the present-day fracture network. Finally, they discuss quality of the outputs obtained via manual vs automatic, demonstrating that the manual method, while more

time-consuming, provides a better quality of data. This work covers the very interesting topic of manual vs automatic methods for extracting faults and fracture data from outcrops and, while I am very supportive to see it published, I think that it still needs major revisions before publication. In places, I found it quite hard to read; particularly, I found the Results section confusing as it goes back and forth discussing data obtained via manual and automatic. Another main issue that needs to be addressed is that some of the claims made are not always well justified. I have addressed all these points in the "Detailed Comments" below.

Detailed Comments:

Lines 147-148: Can you please provide an example for this, e.g. image/figure

Lines 150 – 151: I don't disagree, but I would not say that is "obvious"; for example someone else can claim that they choose to manually pick all fractures because this yields the best results, due to the topography of an outcrop or light exposure during image acquisition. How long is a "reasonable time"? How do you account for it? In this regards, can you please give an overall estimate on the time needed by the automatic process to extract all the fractures from one of the tiles (including all the steps for pre-processing an image and the number of trials needed before finding the right set of parameters to extract the fractures) and compare this on how long it takes manually? In my opinion this would be a very interesting information. In addition, for someone that has never done fracture tracing would learning and using the software make the job faster/easier or it would take longer than do it directly manually? Can you please comment?

Line 156: Can you please report which software have you used during the manual digitalisation of fractures? Was it one of the vector graphics editors (e.g., Adobe Illustrator, Corel Draw, Inkscape), or a geographic information system-based software?

Lines 170 – 173: You never mentioned before any "intensity" parameter; can you please clarify? In addition, when discussing the needs for applying an 'intensity threshold' do

you mean that this process is needed in order to make the automatically detected maps look similar to the original photograph from which you extracted the fractures? If this is the case, this sentence needs to be clarified.

Line 175: For completeness, can you please briefly explain the 'polygon sampling' strategy?

Lines 179 – 180: The sentence "2D fracture networks ..." needs a citation. Can you please clarify the meaning of "spatial graph"? In addition, can you please clarify the terms 'node' and 'edge'? In the context of fracture networks, these can mean different things, depending on whether the fractures themselves are viewed as graph vertexes (and therefore they are the 'nodes'), or if they are considered as links (the 'edges') between fracture intersections and terminations.

Lines 187 – 188: On lines 156 – 157 you mentioned that when manually tracing the fracture these are "traced as polylines". From the use of the prefix 'poly-' I have understood that one fracture trace is already made of many segments. So, can you please explain the need to further subdivide the fracture traces?

Lines 191 – 192: Can you please clarify if the correction for node degree > 3 was your improvement on the software, or it is already and option within of the original NetworkGT?

Lines 192 – 194: Can you please clarify and give more details on the use of the "spatial join function"? How does it work?

Lines 214 – 215: What is Passchier et al., interpretation? The paper is not published yet, so unless you clarify it is not possible to know how these features have been interpreted. Even if the paper was published, it would be more helpful if you could briefly explain this interpretation here, otherwise the reader would be forced to read another paper to understand what you mean.

Line 218: You can avoid the brackets here because you are directly mentioning the

authors.

Lines 223 – 224: Can you please explain why have you chosen to show exclusively fracture intensity maps? To my knowledge NetworkGT allows also for fracture density maps, why have you opted for not showing these?

Lines 228 – 229: This sentence is not clear. Do you mean that number of trace segments is higher than the actual count of fractures? Or do you mean that the automatic extraction of fractures overcount the total number of fractures compared to the manual interpretation?

Line 239: If not resolvable from the drone images, was the 'minimum length' measured directly on outcrop?

Lines 239 – 247: Are all these results relative to the manually traces fractures, the automatic, or both. Not clear. Moreover, can you please add few lines to describe more in details how all the statistical parameter that you show are useful to describe the fracture network? What having a positive kurtosis means? Similarly, can you please better explain what do you mean by symmetric and asymmetric branch distribution?

Line 246: Data of fracture trace length distribution are only shown for the automatic trace detection method, but not for the manually derived network. Is there a reason for this choice? Since you are discussing both methods, and relative results, I would advise to add a similar figure to Fig.7 where showing data from the manually traced network.

Lines 248 – 250: This sentence is not clear. Can you please review it? Are you referring to a limitation of the automatic extraction method?

Lines 250 – 253: For helping the reader to compare/contrast the results of automatic vs. manual method, I would suggest adding to Fig.5 and Fig.6 (which currently only show results for the automatic extraction) the fracture maps obtained by manually picking fractures. I acknowledge that these figures are shown later in the manuscript (figures

13, 14 and 15), but having the fracture maps produced with the two methods one next to each other would be ideal.

Lines 223 – 261 (all section 4.1): The whole section is a bit confusing and needs to be reviewed. It is never clear if you are referring to the results obtained by the manual of the automatic method. For clearness, I would suggest to first describe the results obtained by one method, followed by those obtained by the other, but I do not want to impose any style to your paper. As long as you make it clear which method you are referring to, find a way that you like better.

Lines 275 – 277: I understand that you are referring to the plot of fracture length vs. strike, however as written here it is not very clear. Please review.

Line 287: Is the fault to the Southeast corner of the tile NE2, or to the southeast to the whole outcrop? Please clarify.

Lines 288 – 289: This sentence is rather confusing. Can you please review it? Particularly, can you please clarify the meaning of 'perceived appearance'?

Lines 290 – 291: There is a 'IN' missing between 'domains' and 'the SW'.

Line 314: Should be 'strike at angle'.

Line 390: I understand that you might don't want to engulf the paper with too many figures, however, because you discuss largely around Figure S1, in my opinion this should be incorporated in the main paper, rather than be relegated to supplementary.

Line 396: Would it be more appropriate saying 'automatically extracted fracture network' rather than 'generated'? Usually the term generated is associated to Discrete Fracture Network (DFN) models.

Lines 401 – 402: Point (i), as written, it can be interpreted that the fractures are in the 'code', while the algorithm extracts these features from an image. Please amend. Point (ii), can you please expand on this point?

Lines 402 – 404: As written these sentences are very difficult to understand. Please review them.

Line 404 – 405: Can you please give more details about this procedure? It is not clear to me if you have completely discarded erosional features from the network created when manually tracing the fractures. In addition, the sentence needs to be reviewed: missing 'AS' between 'not' and 'fractures'.

Line 405 – 407: Can you please provide a full description of what do you mean by 'sensitivity'? This term has not been used before in the manuscript, therefore needs to be fully explained. In addition, I would avoid the use of vague adjectives like 'too high' or 'too low'. How much? Can you please quantify?

Line 409: As per my last comment before, can you please quantify 'Slightly smaller'.

Lines 411 – 413: Rather confusing sentence. Do you mean that the dissimilarities between automatic and manual extraction of fractures are comparable to differences between two manual interpretations? Can you provide any evidence for this? Otherwise can you cite appropriate works?

Lines 418 – 421: Please review these three sentences as they are rather unclear. It might be necessary to add a 'the' between 'requires' and 'expertise'. What do you mean by 'several generations are possible for a single fracture'? That a single fracture can be overprinted by a series of tectonic events?

Lines 422 – 423: Can you please clarify? Do you mean that pre-existing fractures can cause distortions in the orientation of later-formed fractures? How widespread such distortions need to be for not be considered just noise in the data (particularly if you have hundreds or thousands of fracture data)?

Line 426: Do you mean that fracture length is not a useful parameter to assign one fracture to a specific set?

Lines 428 – 430: Are you still referring to the fracture set denominated Gen. 1? Not

clear.

Lines 430 – 431: This sentence is not very clear and needs to be reviewed. What do mean by 'fractures as different appearances'?

Lines 432 – 433: Can you please explain which one is this 'larger structure' that you are referring to? Not clear to me.

Lines 436 – 438: In the sentence before this you have argued that gen.1 and gen. 2 can be seen as belonging to one fracture set, however here you assume, without proving it, that these are instead two different fracture sets? Please either refer to a work that shows that gen1 and gen2 were indeed formed at different times, or please provide a full explanation for your assumption.

Lines 442 – 444: A citation is needed when you mention a mechanical cause for the lack of gen3 fracturing in the NE area.

Line 449: what do you mean by 'complete distance'? Whole area?

Lines 462 – 466: All these claims are not supplemented by sufficient evidences or by citing relevant works. How was the paleo-stress oriented at time when gen.4 was formed? Are there veins filling gen1 and gen2 fractures that provide cementation for the fractures? If this is the case, why are you mentioning it just now? Otherwise, do you mean that fluids circulating through gen1 and gen2 fractures caused further cementation in the host rocks near the fractures? If this is the case, you should provide evidences or cite relevant works.

Lines 467 – 468: This sentence needs to be reviewed as it is not clear.

Lines 476 – 477: Can you please further explain the meaning of 'decreasing skewness'? How does this statistical parameter relate to the geometry of the fracture network?

Lines 4879 – 481: I feel that it needs to be made clearer how the variations in the count

of Y nodes and I nodes are related.

Line 483 – 484: How can you establish that a representative domain has been sampled? Can you show examples using your study case? It can be easy to argue that on outcrop the possibility of sampling a 'complete fracture network' are relatively scarce and relegated to few ideal outcrops.

Line 502: What do you mean by 'undirected fractures'? Can you please clarify?

Lines 504 –505: Can you please provide examples that can prove this claim? And, can you please produce a conceptual model that exemplifies the described process?

Line 511: Do you mean that all subareas show comparable branch lengths? As now written is a bit unclear.

Conclusions: While I like bullet points in the conclusion to show the main findings, I also feel that you should have few sentences that wrap up and recap your work. In addition, some of the points listed are a bit vague, specifically point 3 and point 5, as written, do not add any value to the work. So please either reformulate or delete them.

Figure 1. Last sentence is not clear. Do you mean areas the labels show the areas where you have acquired the data?

Figure 4. Please add that P21 indicates 'Fracture Intensity'.

Figures 5 and 6: This is just a personal taste, so you can ignore it, but I would find useful if you can show in these figures the location of the analysed tiles.

Figure 7. Since this plot refers to the automatic extraction method, can you please include in the caption that these are branch lengths and not trace (whole) fracture lengths? AS mentioned in the comments, I would suggest integrating this figure with the length distributions in the manually extrapolated network.

Figure 8. Similar to previous comment. Please indicate that these orientations refer to branches.

[Figure]

---

## Referee Comment (RC2) · William Dunne (Referee) · 20 Jun 2020

Summary Comments: When revised this manuscript will likely become a well cited methodology paper that is being published in a very timely manner. The basic structure and the data of this contribution of certainly of sufficient quality to publish. However, the document needs to achieve a stronger alignment of its actual text with its purpose of comparing the effectiveness of manual and automatically derived fracture trace maps and related fracture parameters or information about fracture development from a UAV-derived composite digital map of a very large fracture trace population. The comparison is not sufficiently developed and even lost, as in the Conclusions, in the narrative of the

document. Key matters that need to be addressed are (1) actual presentation of data related to time required to perform each method; (2) a more effective utilization of fracture intensity (P21 in this case) as a discriminator of quality and/or match between the two data sets, including a careful explanation and use of the attributes that compose P21; (3) avoid two unsupported interpretations in the Discussion; and (4) rebuild the Conclusions so that they are actually about the purpose of the manuscript and highlight key outcomes related to that purpose. All of these changes should be achievable and will add real value to the contribution, such that is should become a strong document valued by the research community.

The work is based on an excellent coastal platform exposure of thin-bedded Jurassic limestones at Lilstock, Somerset, England on the south side of the Bristol channel that has been the focus of a number of previous studies about fracture patterns in rocks.

Major Comments: Section 3.2, Line 150 and elsewhere in the text - Repeatedly and with good reason, the point is made that the manual tracing of the fractures in the digital maps is very time consuming and limits a researcher's ability to effectively utilize the large data sets that can now be created through tools such as UAVs and digital imaging equipment, so the development of effective automatic characterizing protocols will greatly enhance the amount and quality of information for analysis by researchers. Yet, the manuscript does not offer any data to justify this statement. For example, stating the amount of time needed to create each of the five manual samples vs. the five digital samples should be simple and effective. Further, then a comment/short narrative could be added into the discussion about the amount of time that would be needed to characterize the entire "bench" automatically and how, for such a modest amount of time, one would have a much richer data set to tackle........ Consequently, the manuscript should be revised to explicitly document the difference in time usage between the two methods, and then should consider the implications of having the quicker, more powerful automatic approach in the discussion.

Line 156 - It is very important to explicitly state that the manual data set is derived the

digital imagery and was not collected in the field (Correct?). This point in the text is a good place to present this point clearly.

Lines 224 to 225, Fig. 4, and S1-P21 difference illustration - This comparison of fracture intensity between the manual and automatic data gathering approaches is presented as being a primary tool for relating the information and quality from the two data sets. Yet, this comparison is significantly underexplained. For example: (1) Why is a P21 comparison such an effective choice for comparing the two data sets? (2) Just how good is the match, particularly as nothing is said to explain and/or characterize the difference illustration in the S1 illustration? (3) Why is no basic explanation provided of what a reader is seeing in Fig. 4, such as the use of meter-square sample areas, or consideration of the sensitivity from P21 varying spatially from 0 to 18 m-1? (4) Why the lack of an actual narrative comparing the two sets of imagery qualitatively, particularly if locations exist where the match is poor and that needs explanation to, for example, show the overall strength of the approach?

Lines 226 to 227 - An expectation is presented that a larger number of traces would generate a greater fracture intensity (P21) than a smaller number of traces. This expectation is not well rooted. Intensity P21 is a function of total trace length in a sample area, and that is a function of both the number of fracture traces AND their individual lengths (the authors show an understanding of this point in Lines 234/235, but have not utilized that understanding here). So, a fracture population with more traces may, in fact, be less intense because its traces are individually shorter than the other population with fewer but longer traces. Therefore, any expectation of differences in P21 between the two fracture trace populations would need to consider both the number of traces and some aggregate representation of the population of lengths, such as the mean length. Further, if the number of traces is thought to be the key parameter, a much more detailed presentation about the total number of traces in each sample window for each of the two sampling procedures is needed. Overall, the underlying logic of this comparison needs to be better developed and then more completely explained,

if utilized.

Line 367 - Given that the discussion covers "classification into fracture generations" and "network analysis", it is not at all clear why "Passchier et al. (XXXX)" needs to be cited, particularly as it is a very shaky citation with no status or occurrence in the references.

Sinuosity values vary so little from one sample window to another and with respect to the sequence of fracture development in the sample window patterns, would it not be better to eliminate all description/discussion of sinuosity from the manuscript, so as to simplify and focus it?

Lines 411 to 413 - An important comparison of difference is made in this sentence, Yet, no evidence or citation from other work is provided to support that this comparison is correct. So, the statement is an unsupported speculation and really needs to be better than that for the purposes of this manuscript.

Lines 419 to 421 - Here is another key point that is incompletely developed and explained. Particular examples of "human bias" should be identified with explanation. Then the highlighted text can be eliminated and replaced with text that has greater meaning and clarity. Further, the replacement text will need to be a few sentences rather than just one sentence, given the importance of this point.

Line 516 and Conclusions - Given the title of the manuscript and the setup of the abstract and introduction, these conclusions show a surprising lack of content related to comparing manual and automatic methodologies. The conclusion should be reorganized to begin with comparison outcomes (e.g., time usage, P21 comparison, managing manual input into automatic interpretation by parameter selection, etc.). It should delete any text related to superiority of manual to automatic unless a substantial addition is made to the manuscript about that matter. Then it can list particular outcomes for the samples of this particular fracture pattern in this particular location and geological setting, which are not the central focus of the contribution of the manuscript. Lines

423 to 425 - Again, examples with identification in figures are needed to support and document this point.

Comments: Line 21 - As this paper has a focus on methodology, the abstract should briefly present an explanation here as to why automatic assignment of fractures into generations cannot yet be done.

Lines 31-32 - The fracture geometries are set and not "evolving" so the explanation for the connectivity increase needs to be replaced/improved.

Lines 51 to 53 - This sentence should be moved to the end of this section (Line 59), so that this paper's purpose and approach is stated completely first.

Line 52 - What is the status of this companion paper (it is not in the reference list)? Can it be cited or does reference need to be made an unpublished source? Or?

Line 84 - Joints and not jointing are unfilled.

Line 104 - "v" - The significance of the radial pattern to the NE is not provided, so this text is superfluous to the later part of this point. If the radial pattern has significance, it should probably be a separate "vi".

Lines 132 to 140 - Text revised to create a narrative that more contrast imagery results for flight altitudes of 100m or 10m vs. 25m, including the removal of the green highlighted text.

Lines 133 to 137 - This highlighted text as written breaks up the narrative flow. It should likely be added to the end of the text on Line 127.

Lines 145 to 148 - This text should be moved up to the end of Line 127 to complete the general description of methodology. Thus, the subsection will finish with the determination of the optimal flight height and the creation of the overview.

Line 154 and elsewhere in the text - Please remember that "this" and "these" are neither pronouns or nouns.

Line 172 - The intensity belongs to the product, the fractures, and not to the process in this case (clearly, fracturing can be intense, but that is not the intended meaning here, as the description is of a "final fracture population".).

Line 201 - Again, can this contribution actually be cited? What is its status? If it does not have an accepted or published status, what can be used as a citation in its place?

Line 214 - Again, can this contribution actually be cited? What is its status? If it does not have an accepted or published status, what can be used as a citation in its place?

Line 222, Section 4.1 - This section would benefit from a revised title and the addition of two subsection headings to facilitate reader understanding: (1) The entire section is about P21 so change the title to be explicit about trace intensities rather than just traces. (2) 4.1.1 Intensity comparison between two methods (3) 4.1.2 Characterization of intensities for automated data - these two subsection titles clearly separate the purposes of these two portions of the text.

Line 224 - Not everyone knows P21, so here is a good location to simply and explicitly define the term. Note: P21 tends to be our best available measure of fracture abundance for natural rock outcrops and does have wide usage, but it is not universally known.

Lines 227 to 231 - If this text is meant to explain why the P21 intensity for the automatically collected data as compared to the manually collected data, it does not achieve that outcome. What is this text attempting to say? It is not clear?

Lines 231 to 233 - This sentence is not about the P21 comparison between manual and automate, but rather a description of the differences in P21 for the automated data set. It should be the first sentence of the next paragraph, which is about spatial variations in P21 for the automated data sets.

Lines 239 to 241 - The description of maximum length needs to allow for censoring by the sample window perimeters. The identified maximum lengths cannot be stated

to be the actual maximum length of any trace that is sampled in a window because of censoring. Now, it is possible to consider the maximum length of traces that are fully contained in a sample window, but that needs to be stated explicitly.

Line 254 and Figure "18" - To preserve order of figure citation, Figure 18 should become Figure 8, and Figures 8 to 17 should be renumbered.

Line 259 and new Figure 9 (old Figure 8) - This figure illustrates length-weight fracture-trace abundance as a function of orientation, so it does not show "fracture strike" in any simple manner. The text is revised in this line to better describe what is being illustrated.

Lines 264 to 265 - This sentence is an interpretation of observations before the observations are fully presented, so it is out of place and should be deleted.

Lines 265 to 266 - Redundant and unneeded sentence

Line 281 - How are these "denser clusters" recognized or statistically defined? A reader must be able to use the criterion/criteria to identify the clusters themselves. If reproducible methodology does not exist, the statement about orientations modes for clusters should be deleted.

Line 288 and Line 289 - text revised in these two locations to more completely and correctly describe the pattern characteristic of the fractures in the NE1 sample window, and in the NE windows vs. the SW windows, respectively.

Line 289 - text revised to more simply and clearly describe the lack of a relative age relationship.

Lines 291 to 293 - Sentence revised to more clearly identify and state differences with citation of Table 4 being placed in the parentheses at the end of the sentence to not distract from sentence meaning.

Line 301 - Adding the word "evolution" to this sentence is important for clarity and

meaning.

Throughout the text – "mapping boundary" should be "map boundary"

Lines 369 to 370 - Are these two sentences needed here? Their contents are not used immediately. Their content should be added where it is needed further into the document.

Lined 373 - Cite published work that supports the lack of sampling bias and similarity of results for exposure of this quality for different operators. Solid Earth has a publication about this matter, for example.

Line 382 - "excessive" requires some more explanation and/or examples to assist reader comprehension

Lines 388 to 395 - Not convinced that this discussion of the comparison of P21 between the manually and automatically acquired datasets is particularly useful because it considers number of traces separately from the distribution of tracelengths, which is somewhat arbitrary.

Lines 411 to 413 - An important comparison of difference is made in this sentence, Yet, no evidence or citation from other work is provided to support that this comparison is correct. So, the statement is an unsupported speculation and really needs to be better than that for the purposes of this manuscript.

Lines 426 to 438 - This paragraph needs an introductory sentence to establish its purpose and why three different characteristics are being "juxtaposed" in one paragraph. An example sentence is offered.

Lines 452 to 453 - This one sentence paragraph is not really needed as written. A clause is added to the opening sentence about generation 5 to preserve overall narrative flow.

Line 460 to 461 - Text revised and added, so as to achieve greater clarity with a more

complete and needed explanation for readers.

Line 464 to 466 - Text revised to more clearly explain situation and to more simply state examples of possible causes.

Lines 467 to 473 - Recommend the elimination of this text because sinuosity is not a distinguishing characteristic for this study.

Lines 504 to 505 - Interpretation is non-unique and unsupported, so it should not be included in the manuscript.

Please see the annotated PDFs for the main text and figures for additional detailed comments about the syntax.

Captions: Figure 1 – Lines 724-725 states "The study areas on "the bench" that are mapped in detail are marked in 725 red." Yet, the only red in the illustration are the red lines for faults. Is the color incorrectly identified or is something missing from this illustration?

Figure 2 - Line 731 - the object in 2a to 2e should be identified in the caption. Also, 2a to 2f should have scale bars even if they are approximate.

Figure 4 - Line 737 - Please note suggested addition to the text at the end of the caption.

Figure 7 - the labels for the X and Y axes of the three sets of graphs should be larger, so that they are easier to read when viewing the entire figure.

Figure 9 (10) caption improved by explicitly identifying that data belong to manually collected set and providing a better description of the rose diagrams.

Figure 10 (11) Caption - Revise to match revised Figure 9 (10) caption. Eliminate the addition sentence because the network cannot be described as being "oriented NW-SE". The pattern for NE2 can be described as having a strong length-weighted orientation mode that trends NW-SE, but even that information is likely not needed in
the caption here.

Figure 11 (12) - the key at the bottom of the figure should use larger dots so that the colors are easy for readers to distinguish. While these solid circles will be much larger than the actual ones in the plot, that is not an issue because assigning the colors easily to the generations for the readers is the goal.

Figure 13 (14) caption - make the change in the first sentence of the caption for Figures 14(15), 15(16), 16(17), and 17(18).

Tables - Orientation data in all tables - These data should be rounded off to the nearest integer. The use of two significant figures to the right of the decimal point is false precision, particularly for the manually traced lines.

Table 1 – Title expanded to explicitly identify that the characteristics stated in the table related to the automatically generated fracture network

Please also note the supplement to this comment:
https://se.copernicus.org/preprints/se-2020-67/se-2020-67-RC2-supplement.pdf
* * *
[Figure]

**Summary Comments:**

When revised this manuscript will likely become a well cited methodology paper that is being published in a very timely manner. The basic structure and the data of this contribution of certainly of sufficient quality to publish. However, the document needs to achieve a stronger alignment of its actual text with its purpose of comparing the effectiveness of manual and automatically derived fracture trace maps and related fracture parameters or information about fracture development from a UAV-derived composite digital map of a very large fracture trace population. The comparison is not sufficiently developed and even lost, as in the Conclusions, in the narrative of the document. Key matters that need to be addressed are (1) actual presentation of data related to time required to perform each method; (2) a more effective utilization of fracture intensity (P21 in this case) as a discriminator of quality and/or match between the two data sets, including a careful explanation and use of the attributes that compose P21; (3) avoid two unsupported interpretations in the Discussion; and (4) rebuild the Conclusions so that they are actually about the purpose of the manuscript and highlight key outcomes related to that purpose. All of these changes should be achievable and will add real value to the contribution, such that is should become a strong document valued by the research community.

The work is based on an excellent coastal platform exposure of thin-bedded Jurassic limestones at Lilstock, Somerset, England on the south side of the Bristol channel that has been the focus of a number of previous studies about fracture patterns in rocks.

**Major Comments**:

Section 3.2, Line 150 and elsewhere in the text - Repeatedly and with good reason, the point is made that the manual tracing of the fractures in the digital maps is very time consuming and limits a researcher's ability to effectively utilize the large data sets that can now be created through tools such as UAVs and digital imaging equipment, so the development of effective automatic characterizing protocols will greatly enhance the amount and quality of information for analysis by researchers. Yet, the manuscript does not offer any data to justify this statement. For example, stating the amount of time needed to create each of the five manual samples vs. the five digital samples should be simple and effective. Further, then a comment/short narrative could be added into the discussion about the amount of time that would be needed to characterize the entire "bench" automatically and how, for such a modest amount of time, one would have a much richer data set to tackle........ Consequently, the manuscript should be revised to explicitly document the difference in time usage between the two methods, and then should consider the implications of having the quicker, more powerful automatic approach in the discussion.

Line 156 - It is very important to explicitly state that the manual data set is derived the digital imagery and was not collected in the field (Correct?). This point in the text is a good place to present this point clearly.

Lines 224 to 225, Fig. 4, and S1-P21 difference illustration - This comparison of fracture intensity between the manual and automatic data gathering approaches is presented as being a primary tool for relating the information and quality from the two data sets. Yet, this comparison is significantly underexplained. For example: (1) Why is a P21 comparison such an effective choice for comparing the two data sets? (2) Just how good is the match, particularly as nothing is said to explain and/or characterize the difference illustration in the S1 illustration? (3) Why is no basic explanation provided of what a reader is seeing in Fig. 4, such as the use of meter-square sample areas, or consideration of the sensitivity from P21 varying spatially from 0 to 18 m$^{-1}$? (4) Why the lack of an actual narrative comparing the two sets of imagery qualitatively, particularly if locations exist where the match is poor and that needs explanation to, for example, show the overall strength of the approach?

Lines 226 to 227 - An expectation is presented that a larger number of traces would generate a greater fracture intensity (P21) than a smaller number of traces. This expectation is not well rooted. Intensity P21 is a function of total trace length in a sample area, and that is a function of both the number of fracture traces AND their individual lengths (the authors show an understanding of this point in Lines 234/235, but have not utilized that understanding here). So, a fracture population with more traces may, in fact, be less intense because its traces are individually shorter than the other population with fewer but longer traces. Therefore, any expectation of differences in P21 between the two fracture trace

**Fig. 1.**

**Supplement:**

[revised manuscript text omitted]

---

## Author Comment (AC1) · 10 Jul 2020

We thank Roberto Emanuele Rizzo for the detailed and helpful comments on the manuscript.

A main criticism was that the manuscript was quite hard to read, particularly in the results section where it goes back and forth discussing the data obtained from manual and automatic methods. In the new version of the manuscript we have taken special care to clarify which methods we are referring to in the narrative. We revised the results section accordingly and further divided the section into sub-sections for a clearer structuring therein. Throughout the manuscript, we further improved the narrative for

a better readability. Another major point of criticism was that some of the claims presented in the manuscript were not always well justified. We agree that some of our claims required further explanations and citations and provided those for the addressed cases whenever possible. For the cases where we were not able to provide a sufficient amount of evidence or citations, or when the latter would raise questions out of the scope of this study, we removed the claims from the manuscript accordingly. Detailed comments on the individual points raised by the reviewer are provided below.

Lines 147-148: Can you please provide an example for this, e.g. image/figure

Reply: The section was moved a few lines up according to the suggestion of the other reviewer. We added a figure showing examples of the ground truthing to the supplement (new S1).

Lines 150 – 151: I don't disagree, but I would not say that is "obvious"; for example someone else can claim that they choose to manually pick all fractures because this yields the best results, due to the topography of an outcrop or light exposure during image acquisition.

Reply: We agree that "obvious" was not the best choice of words. The "obvious" was replaced according to the suggestion of the other reviewer to clarify, that we refer to the need of a more rapid technique.

How long is a "reasonable time"? How do you account for it?

Reply: The initial choosing of words did not provide clear information, we have revised the sentence for clarity and provide quantitative estimations. More information about the time required and extrapolated to the complete outcrop was added to the section below.

In this regards, can you please give an overall estimate on the time needed by the automatic process to extract all the fractures from one of the tiles (including all the steps for preprocessing an image and the number of trials needed before finding the

right set of parameters to extract the fractures) and compare this on how long it takes manually? In my opinion this would be a very interesting information. In addition, for someone that has never done fracture tracing would learning and using the software make the job faster/easier or it would take longer than do it directly manually? Can you please comment?

Reply: We agree that these points are very interesting for the reader. This was also one of the main comments of the other reviewer, therefore, we added a narrative in the methods parts explaining the required time under different aspects for each of the two methods in detail, incorporating the tracing itself and the time required for the removal of artifacts. Further points were added in the discussion giving an estimate what time would be required to trace the whole dataset using either technique. Also, we have added a short explanation to highlight the pros and cons for someone who has never done fracture tracing for both techniques. As explained in the manuscript, the dataset requires the interpreter to make judgment calls on several occasions, thus this task might be challenging for someone who has never done fracture tracing before. With the help of the automatic trace extraction, the results would be unbiased and easier to reproduce by another interpreter. Timewise, an unexperienced interpreter must either face the learning curve of the GIS software (or any other software used to manually trace the fractures) or MATLAB for which the automatic trace extraction code is written. Depending on personal preferences, either learning curve might be steeper. For small datasets and an unexperienced interpreter, it might be faster to do everything manually, but if larger datasets are to be processed, the time invested for learning the automatic trace extraction is compensated by the time saved during the application of the method.

Line 156: Can you please report which software have you used during the manual digitalization of fractures? Was it one of the vector graphics editors (e.g., Adobe Illustrator, Corel Draw, Inkscape), or a geographic information system-based software?

Reply: This information was accidentally removed during an earlier revision of the manuscript. We now clearly mention the use of ArcGIS at this point.

Lines 170 – 173: You never mentioned before any "intensity" parameter; can you please clarify? In addition, when discussing the needs for applying an 'intensity threshold' do you mean that this process is needed in order to make the automatically detected maps look similar to the original photograph from which you extracted the fractures? If this is the case, this sentence needs to be clarified.

Reply: The "intensity" in this case refers to the chosen parameter combination during the automatic extraction. We have slightly revised the sentences to clarify and cited Prabhakaran et al. (2019) for further information on the topic for the interested reader.

Line 175: For completeness, can you please briefly explain the 'polygon sampling' strategy?

Reply: We added a narrative to briefly explain the polygon sampling strategy as discussed in Nyberg et al. 2018.

Lines 179 – 180: The sentence "2D fracture networks ..." needs a citation. Can you please clarify the meaning of "spatial graph"? In addition, can you please clarify the terms 'node' and 'edge'? In the context of fracture networks, these can mean different things, depending on whether the fractures themselves are viewed as graph vertexes (and therefore they are the 'nodes'), or if they are considered as links (the 'edges') between fracture intersections and terminations.

Reply: We added a citation as suggested (Sanderson et al. 2019) and also added some sentences to better explain the meaning of "spatial graph" and our use of the terms 'node' and 'edge' with respect to their use in literature.

Lines 187 – 188: On lines 156 – 157 you mentioned that when manually tracing the fracture these are "traced as polylines". From the use of the prefix 'poly-' I have understood that one fracture trace is already made of many segments. So, can you please explain the need to further subdivide the fracture traces?

Reply: The term "polyline" is used according to its definition within the ArcGIS environment: "A Polyline object is a shape defined by one or more paths, in which a path is a series of connected segments." Therefore, it is correct that one manually mapped fracture trace consists of a polyline made of many segments that share the ID of the polyline. However, NetworkGT only allows single segments (stored with an individual ID) as input, therefore we had to dissolve the polylines first. We have added a narrative to point this out for the reader along with a more detailed explanation.

Lines 191 – 192: Can you please clarify if the correction for node degree > 3 was your improvement on the software, or it is already and option within of the original NetworkGT?

Reply: In the manuscript we state that nodes with a degree > 4 (X-node) are not supported/implemented in the code of NetworkGT and, thus, returned as error. To fix these resulting errors, we developed our own method to correct these errors resulting from the application of NetworkGT. We revised the sentence to make this point clear.

Lines 192 – 194: Can you please clarify and give more details on the use of the "spatial join function"? How does it work?

Reply: The spatial join function is provided in the ArcGIS Analysis Toolbox and can be used to join features based on different parameters of their spatial location. We added a few sentences to give more details on the topic and provide a better explanation on how we applied this function to our data.

Lines 214 – 215: What is Passchier et al., interpretation? The paper is not published yet, so unless you clarify it is not possible to know how these features have been interpreted. Even if the paper was published, it would be more helpful if you could briefly explain this interpretation here, otherwise the reader would be forced to read another paper to understand what you mean.

Reply: In Passchier et al. (in prep.) we introduce the initial interpretations of the generations on a larger scale of the outcrop, which helped to identify the criteria on which we

base our interpretation of the generations in this manuscript. However, the criteria we used and present in this work are self-sustained, explained in detail in section 3.4 and do not require the citation of Passchier et al. (in prep). We now refer to this citation as a companion paper only once and earlier in the manuscript and rewrote the narrative to clarify, that the interpretation of the generations is based on the criteria explained in this manuscript.

Line 218: You can avoid the brackets here because you are directly mentioning the authors.

Reply: We revised the sentence according to the suggestions of the other reviewer. In the revised sentence, the brackets are now necessary.

Lines 223 – 224: Can you please explain why have you chosen to show exclusively fracture intensity maps? To my knowledge NetworkGT allows also for fracture density maps, why have you opted for not showing these?

Reply: In the initial version of the manuscript we only presented fracture intensity because fracture intensity shows the total persistence of fracture segments within the domains, while fracture density only gives the amount of fractures segments per unit area. Fracture segment density was assumed to be of lesser interest because we already stated that automatic trace extraction produces more trace segments than manual tracing (of whole fractures and splitting them at the nodes), therefore, higher densities for the automatic extraction are just the logical consequence. To provide a better comparison of the methods presented in this manuscript we revised the section following the suggestion of the other reviewer and now also incorporate fracture segment densities throughout the domains. We have added two figures comparing the results for both methods, further material to the supplement, a new subsection introducing the results of P20 for both methods and a narrative to explain why we now investigate fracture segment density and intensity with implications for the two methods.

Lines 228 – 229: This sentence is not clear. Do you mean that number of trace segments is higher than the actual count of fractures? Or do you mean that the automatic extraction of fractures overcount the total number of fractures compared to the manual interpretation?

Reply: The automatic code generates segments, while a manual interpreter maps the complete fracture from tip to tip which represents the path along several connected segments. We revised the sentence for clarity and now better explain the difference of the results and how to process the data to make a comparison possible by splitting the manually traced fracture traces into segments analogue to the ones generated automatically.

Line 239: If not resolvable from the drone images, was the 'minimum length' measured directly on outcrop?

Reply: The section was rewritten according to the suggestion of the other reviewer. Now it is clarified that we are referring to a cutoff length within the sampling windows. These short segments are the result of the tracing and cannot be verified because they are below the resolution of the ortho-mosaics.

Lines 239 – 247: Are all these results relative to the manually traces fractures, the automatic, or both. Not clear. Moreover, can you please add few lines to describe more in details how all the statistical parameter that you show are useful to describe the fracture network? What having a positive kurtosis means? Similarly, can you please better explain what do you mean by symmetric and asymmetric branch distribution?

Reply: We revised the whole section and divided it into sub-sections to make it more clear to which results (manual or automatic) we are referring to. The section describing the statistical parameters has been extended to provide more detailed explanations of the parameters and their meaning or implication for the network.

Line 246: Data of fracture trace length distribution are only shown for the automatic trace detection method, but not for the manually derived network. Is there a reason

for this choice? Since you are discussing both methods, and relative results, I would advise to add a similar figure to Fig.7 where showing data from the manually traced network.

Reply: To follow the initial narrative of the manuscript, we stated that the results of both methods are comparable anyway and, thus, we only used the results of one method to describe the network. Now, that we have revised the narrative to an even larger focus on the comparison of both methods, we have added a similar figure presenting the manually traced segments as advised and also added a section that compares both methods using the results presented in the figures.

Lines 248 – 250: This sentence is not clear. Can you please review it? Are you referring to a limitation of the automatic extraction method?

Reply: Yes, this is what we are referring to. The sentence was revised according to the suggestion of the other reviewer for clarification.

Lines 250 – 253: For helping the reader to compare/contrast the results of automatic vs. manual method, I would suggest adding to Fig.5 and Fig.6 (which currently only show results for the automatic extraction) the fracture maps obtained by manually picking fractures. I acknowledge that these figures are shown later in the manuscript (figures13, 14 and 15), but having the fracture maps produced with the two methods one next to each other would be ideal.

Reply: We agree that a direct comparison of the networks resulting from the techniques is interesting for the reader. However, this would require another very large figure for the networks and the branches and nodes to be visible, and break the narrative at this point of the manuscript. We present a figure that directly compares all networks using P21 in which the networks are shown in the background of the plots and can be compared there with the advantage of different P21 values pointing the reader directly to locations where the methods have different results. Furthermore, we have added examples of two areas which are compared in detail using P20 along with supplementary

material that shows P20 with the networks plotted in the background in the supplement. For the interested reader, the shapefiles are also provided as supplementary material and can compared using a GIS.

Lines 223 – 261 (all section 4.1): The whole section is a bit confusing and needs to be reviewed. It is never clear if you are referring to the results obtained by the manual of the automatic method. For clearness, I would suggest to first describe the results obtained by one method, followed by those obtained by the other, but I do not want to impose any style to your paper. As long as you make it clear which method you are referring to, find a way that you like better.

Reply: We agree that the initial version of the section was a bit confusing. We revised the whole section for clarity in our narrative and divided it into several subsections to better compare both methods, following the suggestion of the other reviewer. Within the subsections we also revised the narrative to avoid confusion between the two methods and now state clearly which method we are referring to.

Lines 275 – 277: I understand that you are referring to the plot of fracture length vs. strike, however as written here it is not very clear. Please review.

Reply: We revised the sentence to clarify that we refer to the relationship of strike and length of single fractures. Also, according to the comment of reviewer 2, we rewrote the sentence to avoid confusion with statistically defined clusters in the plots.

Line 287: Is the fault to the Southeast corner of the tile NE2, or to the southeast to the whole outcrop? Please clarify.

Reply: The sentence was revised to clarify that we are referring to the relative position of the fault to the sample window.

Lines 288 – 289: This sentence is rather confusing. Can you please review it? Particularly, can you please clarify the meaning of 'perceived appearance'?

Reply: We revised the sentence and deleted the term "perceived appearance" to avoid

confusion. Now we state clearly that we refer to the generation 3 which is present in the network of the domains in the SW but not in the NE.

Lines 290 – 291: There is a 'IN' missing between 'domains' and 'the SW'.

Reply: We added the 'in'.

Line 314: Should be 'strike at angle'.

Reply: We replaced "in" with "at".

Line 390: I understand that you might don't want to engulf the paper with too many figures, however, because you discuss largely around Figure S1, in my opinion this should be incorporated in the main paper, rather than be relegated to supplementary.

Reply: We agree with this point and added the supplement S1 as figure to the manuscript within the revised section 4.1.

Line 396: Would it be more appropriate saying 'automatically extracted fracture network' rather than 'generated'? Usually the term generated is associated to Discrete Fracture Network (DFN) models.

Reply: We agree and changed the wording accordingly to avoid confusion

Lines 401 – 402: Point (i), as written, it can be interpreted that the fractures are in the 'code', while the algorithm extracts these features from an image. Please amend. Point (ii), can you please expand on this point?

Reply: We revised (i) to avoid confusion. Now it is clear that the code extracts the fractures we refer to. Furthermore, we expanded (ii) by providing more explanations and details in the following sentences

Lines 402 – 404: As written these sentences are very difficult to understand. Please review them.

Reply: We revised the sentences for clarity. As suggested by the other reviewer, we

also added a new figure with more detailed explanations to the manuscript to provide examples of the cases discussed in these sentences.

Line 404 – 405: Can you please give more details about this procedure? It is not clear to me if you have completely discarded erosional features from the network created when manually tracing the fractures. In addition, the sentence needs to be reviewed: missing 'AS' between 'not' and 'fractures'.

Reply: In this sentence we refer to wrong positives in the automatic extraction caused by a rough surface and a too high sensitivity (in terms of the chosen parameter combination) of the code. We added a more detailed explanation of the issue and rewrote the sentence to clarify.

Line 405 – 407: Can you please provide a full description of what do you mean by 'sensitivity'? This term has not been used before in the manuscript, therefore needs to be fully explained. In addition, I would avoid the use of vague adjectives like 'too high' or 'too low'. How much? Can you please quantify?

Reply: We revised the sentence and replaced "sensitivity" to clarify that we refer to the parameter combination chosen during the automatic tracing, what is explained with more details earlier in the manuscript. The sentence and complete manuscript were revised to avoid vague adjectives as suggested, using qualifications instead.

Line 409: As per my last comment before, can you please quantify 'Slightly smaller'.

Reply: We replaced "slightly smaller" by stating a quantity.

Lines 411 – 413: Rather confusing sentence. Do you mean that the dissimilarities between automatic and manual extraction of fractures are comparable to differences between two manual interpretations? Can you provide any evidence for this? Otherwise can you cite appropriate works?

Reply: Yes, this is what we wanted to express. We revised the sentence for clarity and added a citation of a work that compared manual interpretations of fracture networks

to back our statement.

Lines 418 – 421: Please review these three sentences as they are rather unclear. It might be necessary to add a 'the' between 'requires' and 'expertise'. What do you mean by 'several generations are possible for a single fracture'? That a single fracture can be overprinted by a series of tectonic events?

Reply: The sentences were revised according to the suggestions of the other reviewer for clarity. Now it is clear that we wanted to say that several generations can be assigned to a single fracture trace during the interpretation of the generations. Furthermore, we added a more detailed section below that includes a new figure to better explain the narrative besides providing more detailed explanations of the points discussed in the section.

Lines 422 – 423: Can you please clarify? Do you mean that pre-existing fractures can cause distortions in the orientation of later-formed fractures? How widespread such distortions need to be for not be considered just noise in the data (particularly if you have hundreds or thousands of fracture data)?

Reply: Yes, this is what we wanted to express. We rephrased the sentence to make this clear for the reader. In our cases, we have mainly observed this occurrence for old fractures that terminated at their tips within the sampling window. While this might be of minor relevance for the interpretation of generations, it is necessary to be mentioned for the analysis of the fracture network evolution with respect to its connectivity. There, we see that the tips of the old fractures (I nodes) are successively connected to other fractures until barely any isolated tips remain in the network. Therefore, we interpret this connecting of the tips as a common occurrence in this fracture network.

Line 426: Do you mean that fracture length is not a useful parameter to assign one fracture to a specific set?

Reply: Yes, this circumstance is caused by the censoring of the long fractures by our

map domains. We revised the sentence for clarity and added further explanations to point out that length as a parameter is biased in the presented case.

Lines 428 – 430: Are you still referring to the fracture set denominated Gen. 1? Not clear.

Reply: Yes, we are. We revised the sentence for clarity.

Lines 430 – 431: This sentence is not very clear and needs to be reviewed. What do mean by 'fractures as different appearances'?

Reply: We revised the sentence according to the suggestion of the other reviewer. Now we clearly state that we refer to the trend of the fractures in this case.

Lines 432 – 433: Can you please explain which one is this 'larger structure' that you are referring to? Not clear to me.

Reply: We refer to the fault as shown in fig, 1b. We have added the reference to the figure and "fault" for clarification. With the revised figure caption of figure 1, this is now clearer.

Lines 436 – 438: In the sentence before this you have argued that gen.1 and gen. 2 can be seen as belonging to one fracture set, however here you assume, without proving it, that these are instead two different fracture sets? Please either refer to a work that shows that gen1 and gen2 were indeed formed at different times, or please provide a full explanation for your assumption.

Reply: We revised the previous sentence according to the other reviewer. The explanation was expanded to further clarify, that we stuck to our initial interpretation because we cannot make a reliable decision in this case simply based on the abutting criteria, which allow several interpretations: either an interpretation as two consecutive generations, or as one generation in which the geometry of the abutting fractures simply represents the order of which fractures belonging to one generation developed. However, either interpretation leads the same result in the following analysis of the network

development because the relative order in which the fractures developed remains the same.

Lines 442 – 444: A citation is needed when you mention a mechanical cause for the lack of gen3 fracturing in the NE area.

Reply: The sentence was revised according to the suggestion of the other reviewer. The mentioning of a "mechanical cause" was been removed and we simply refer to the possibility of other reasons that are not within the scope of this study.

Line 449: what do you mean by 'complete distance'? Whole area?

Reply: Yes, we revised the sentence for clarity.

Lines 462 – 466: All these claims are not supplemented by sufficient evidences or by citing relevant works. How was the paleo-stress oriented at time when gen.4 was formed? Are there veins filling gen1 and gen2 fractures that provide cementation for the fractures? If this is the case, why are you mentioning it just now? Otherwise, do you mean that fluids circulating through gen1 and gen2 fractures caused further cementation in the host rocks near the fractures? If this is the case, you should provide evidences or cite relevant works.

Reply: We revised the section accordingly to the suggestions of the other reviewer. Now, we explain the situation more clearly and more simply state examples of possible causes without the need of additional citations that lead to details which are out of the scope of this study.

Lines 467 – 468: This sentence needs to be reviewed as it is not clear.

Reply: The sentence was removed due to the elimination of sinuosity in this manuscript as suggested by the other reviewer.

Lines 476 – 477: Can you please further explain the meaning of 'decreasing skewness'? How does this statistical parameter relate to the geometry of the fracture network?

Reply: We added a sentence to briefly explain the reference, now also provide a clear link to the figures and tables showing the data. More details are now presented earlier in the revised section 4.1.

Lines 4879 – 481: I feel that it needs to be made clearer how the variations in the count of Y nodes and I nodes are related.

Reply: This part of section is supposed to discuss trends in the data visible throughout our domains and not intended to focus on the relationships of nodes. The appearance and relationships of different node types is discussed in detail a few lines below, still within the same section.

Line 483 – 484: How can you establish that a representative domain has been sampled? Can you show examples using your study case? It can be easy to argue that on outcrop the possibility of sampling a 'complete fracture network' are relatively scarce and relegated to few ideal outcrops.

Reply: This is the point we wanted to make at this position to highlight the necessity of a complete interpretation of the network. We present the criteria based on which we have selected our domains in section 2.2 and added further text at this part of the discussion to debate what aspects need to be considered when the domains are supposed to be representative for the complete network at this point.

Line 502: What do you mean by 'undirected fractures'? Can you please clarify?

Reply: We referred to generation 5 fractures which do not follow a clear orientation mode in all areas. We removed "undirected" to avoid confusion and revised the statement to one of a more general nature.

Lines 504 –505: Can you please provide examples that can prove this claim? And, can you please produce a conceptual model that exemplifies the described process?

Reply: This claim was non-unique and unsupported and not necessary to aid the narrative of the manuscript. Therefore, we have removed it from the manuscript according to the suggestion of the other reviewer.

Line 511: Do you mean that all subareas show comparable branch lengths? As now written is a bit unclear.

Reply: Yes, we revised the sentence for clarity.

Conclusions: While I like bullet points in the conclusion to show the main findings, I also feel that you should have few sentences that wrap up and recap your work. In addition, some of the points listed are a bit vague, specifically point 3 and point 5, as written, do not add any value to the work. So please either reformulate or delete them.

Reply: We added a introduction to the conclusion to recap our work presented in the manuscript. To account for the comment of the other reviewer on this chapter, we have also revised the bullet points. Former point 3 was reformulated and point 5 deleted.

Figure 1. Last sentence is not clear. Do you mean areas the labels show the areas where you have acquired the data?

Reply: The figure was revised earlier, and the changes of the captions were lost. We updated the caption, it now states that the map domains in which we traced the fractures are marked as the yellow (not red) squares in the figure.

Figure 4. Please add that P21 indicates 'Fracture Intensity'.

Reply: This clarification was added to the caption.

Figures 5 and 6: This is just a personal taste, so you can ignore it, but I would find useful if you can show in these figures the location of the analysed tiles.

Reply: The location of the analyzed tiles is shown in figure 1. This is now clarified by the revision of the caption of figure 1. Therefore, we do not include their locations additionally in the figures as suggested, because this would a repetition and furthernone

more reduce the size of the figures with the important content at this place, leading to a worse readability.

Figure 7. Since this plot refers to the automatic extraction method, can you please include in the caption that these are branch lengths and not trace (whole) fracture lengths? AS mentioned in the comments, I would suggest integrating this figure with the length distributions in the manually extrapolated network.

Reply: The caption was revised accordingly, and an additional figure showing the length distributions of the manually traced segments was added to the manuscript as suggested.

Figure 8. Similar to previous comment. Please indicate that these orientations refer to branches.

Reply: We now state explicitly that we are referring to branches in the caption.

---

## Author Comment (AC2) · 10 Jul 2020

We thank William Dunne for the detailed and very constructive comments on the manuscript.

A major point of his comments is that the document needs to achieve a stronger alignment of its actual text with its purpose of comparing the manual and automatic methods, because the comparison is not sufficiently developed. We agree and carefully revised the manuscript to better develop this point throughout the whole narrative, particularly in the sections comparing the two methods.

[Figure]

Further key matters that needed to be addressed are the following ones:

(1) actual presentation of data related to time required to perform each method.

Reply: In the revised version of the manuscript, we extended the narrative at several points to compare the different times required for both methods, discuss different parameters that influence the required time and provide estimates to map the complete outcrop using both techniques.

(2) a more effective utilization of fracture intensity (P21 in this case) as a discriminator of quality and/or match between the two data sets, including a careful explanation and use of the attributes that compose P21

Reply: We revised and extended the sections presenting and discussing P21. We now provide more details on the method itself, explain the attributes composing P21, how it can be applied to our dataset and what the results indicate with respect to the different methods. We have added an additional figure to show the differences of P21 for both methods and discuss them in detail. To further back the P21 analysis, we also included a preceding section where we introduce and discuss P20 (fracture trace densities) with additional figures as preceding section to the P21 section. Overall, we provide a more detailed explanation on the Pij system and compare and analyze the results of P20 and P21 with respect to the different methods and the structure of the data of fracture trace segments.

(3) avoid two unsupported interpretations in the Discussion;

Reply: We removed any unsupported interpretations throughout the manuscript and provided further explanations and citations for interpretations that are supported but were lacking the necessary arguments in the previous version of the manuscript.

(4) rebuild the Conclusions so that they are actually about the purpose of the manuscript and highlight key outcomes related to that purpose.

Reply: We revised the conclusions and added a short introduction to recap our work.

[Figure]

The bullet points were extended and rearranged, to first evaluate the raw data, then compare the aspect of time required for both methods followed by the differences and similarities (P20, P21) and pros and cons of both methods. Once the main key findings with respect to the different methods are addressed, more general points are listed that highlight the particular findings for the particular fracture network throughout the domains.

Detailed comments on the individual points raised by the reviewer are provided below.

Major Comments:

Section 3.2, Line 150 and elsewhere in the text - Repeatedly and with good reason, the point is made that the manual tracing of the fractures in the digital maps is very time consuming and limits a researcher's ability to effectively utilize the large data sets that can now be created through tools such as UAVs and digital imaging equipment, so the development of effective automatic characterizing protocols will greatly enhance the amount and quality of information for analysis by researchers. Yet, the manuscript does not offer any data to justify this statement. For example, stating the amount of time needed to create each of the five manual samples vs. the five digital samples should be simple and effective. Further, then a comment/short narrative could be added into the discussion about the amount of time that would be needed to characterize the entire "bench" automatically and how, for such a modest amount of time, one would have a much richer data set to tackle........ Consequently, the manuscript should be revised to explicitly document the difference in time usage between the two methods, and then should consider the implications of having the quicker, more powerful automatic approach in the discussion.

Reply: We agree that these points are very interesting for the reader. Therefore, we added a narrative in the methods parts explaining the required time under different aspects for each of the two methods in detail, incorporating the tracing itself and the time required for the removal of artifacts. Further points were added in the discussion giving

an estimate what time would be required to trace the whole dataset using either technique and discuss advantages and disadvantages of both methods with respect to the required time in different scenarios. Also, we have added a short explanation to highlight the pros and cons for someone who has never done fracture tracing following the suggestion of the other reviewer. Following the revisions throughout the manuscript, we revised the conclusions accordingly to better present the most important findings with respect to the time required using both methods.

Line 156 - It is very important to explicitly state that the manual data set is derived the digital imagery and was not collected in the field (Correct?). This point in the text is a good place to present this point clearly.

Reply: We agree and now state explicitly, that the data was derived from the digital imagery and that we used ArcGIS as requested by the other reviewer.

Lines 224 to 225, Fig. 4, and S1-P21 difference illustration - This comparison of fracture intensity between the manual and automatic data gathering approaches is presented as being a primary tool for relating the information and quality from the two data sets. Yet, this comparison is significantly underexplained. For example: (1) Why is a P21 comparison such an effective choice for comparing the two data sets? (2) Just how good is the match, particularly as nothing is said to explain and/or characterize the difference illustration in the S1 illustration? (3) Why is no basic explanation provided of what a reader is seeing in Fig. 4, such as the use of meter-square sample areas, or consideration of the sensitivity from P21 varying spatially from 0 to 18 m-1? (4) Why the lack of an actual narrative comparing the two sets of imagery qualitatively, particularly if locations exist where the match is poor and that needs explanation to, for example, show the overall strength of the approach?

Reply: We agree that the narrative was underdeveloped int the section and revised it carefully. Besides P21 we now also provide examples of P20. Both are now properly introduced along with explanations why we use them and what their results tell us about

the network. Explanations and examples are added for the new and existing figures to back the narrative and make it more comprehensive for the readers.

Lines 226 to 227 - An expectation is presented that a larger number of traces would generate a greater fracture intensity (P21) than a smaller number of traces. This expectation is not well rooted. Intensity P21 is a function of total trace length in a sample area, and that is a function of both the number of fracture traces AND their individual lengths (the authors show an understanding of this point in Lines 234/235, but have not utilized that understanding here). So, a fracture population with more traces may, in fact, be less intense because its traces are individually shorter than the other population with fewer but longer traces. Therefore, any expectation of differences in P21 between the two fracture trace populations would need to consider both the number of traces and some aggregate representation of the population of lengths, such as the mean length. Further, if the number of traces is thought to be the key parameter, a much more detailed presentation about the total number of traces in each sample window for each of the two sampling procedures is needed. Overall, the underlying logic of this comparison needs to be better developed and then more completely explained, if utilized.

Reply: We agree that our reasoning in the old version of the manuscript was not well presented one extended and clarified the statements, along with a more detailed narrative, additional figures and an extended Table 1. Besides P21 we now also provide examples of P20 (see the reply to the prior comment) to compare the different numbers of traces generated by both methods in the sample windows for a better comparison of the trace segments as they result from both methods. We now explain that we use P21 to compare the trace lengths present in the sample areas, which allows us to better compare the different traces generated by both methods with respect to their lengths representing the fractures, giving us better control over the interpretation of the results of both methods. This topic is now addressed within a whole sub-section to provide better and more extensive explanations on how we utilize Pij and P21 in that particular

section, and how we use it to compare the different methods.

Line 367 - Given that the discussion covers "classification into fracture generations" and "network analysis", it is not at all clear why "Passchier et al. (XXXX)" needs to be cited, particularly as it is a very shaky citation with no status or occurrence in the references.

Reply: In Passchier et al. (in prep.) we introduce the initial interpretations of the generations on a larger scale of the outcrop, which helped to identify the criteria on which we base our interpretation of the generations in this manuscript. However, the criteria we used and present in this work are self-sustained, explained in detail in section 3.4 and do not require the citation of Passchier et al. (in prep). We now refer to this citation as a companion paper only once and earlier in the manuscript and rewrote the narrative to clarify, that the interpretation of the generations is based on the criteria explained in this manuscript.

Sinuosity values vary so little from one sample window to another and with respect to the sequence of fracture development in the sample window patterns, would it not be better to eliminate all description/discussion of sinuosity from the manuscript, so as to simplify and focus it?

Reply: We agree that the differences of sinuosity, spatially or between the methods, are minor and not significant, thus, we eliminated sinuosity from the manuscript entirely, because it does not add valuable information.

Lines 411 to 413 - An important comparison of difference is made in this sentence, Yet, no evidence or citation from other work is provided to support that this comparison is correct. So, the statement is an unsupported speculation and really needs to be better than that for the purposes of this manuscript.

Reply: We infer that the automatic code at this stage represents a good option for creating an initial fracture trace map that only differs from a manual interpretation to a

degree that is comparable to the deviation of two manual interpretations of the same fracture network. This assumption is backed by e.g. Long et al. (2018), who compared different manual interpretations of fracture networks and is now cited in the revised version of the manuscript. The sentence was revised for clarity and a citation added to support our point.

Lines 419 to 421 - Here is another key point that is incompletely developed and explained. Particular examples of "human bias" should be identified with explanation. Then the highlighted text can be eliminated and replaced with text that has greater meaning and clarity. Further, the replacement text will need to be a few sentences rather than just one sentence, given the importance of this point.

Reply: We agree that is an important point. We extended the section and added an additional figure to provide several particular examples and more detailed explanations discussing human bias, e.g. several examples of non-unique interpretations to aid the narrative in the new section. The highlighted text was replaced by a more elaborate section.

Line 516 and Conclusions - Given the title of the manuscript and the setup of the abstract and introduction, these conclusions show a surprising lack of content related to comparing manual and automatic methodologies. The conclusion should be reorganized to begin with comparison outcomes (e.g., time usage, P21 comparison, managing manual input into automatic interpretation by parameter selection, etc.). It should delete any text related to superiority of manual to automatic unless a substantial addition is made to the manuscript about that matter. Then it can list particular outcomes for the samples of this particular fracture pattern in this particular location and geological setting, which are not the central focus of the contribution of the manuscript.

Reply: We have added a short introduction to the conclusion to recap our work presented in the manuscript. To account for this comment, we have also revised the bullet points. Former point 3 was reformulated and point 5 deleted. The bullet points were

extended and rearranged, to first evaluate the raw data, then compare the aspect of time required for both methods followed by the differences and similarities (P20, P21) and pros and cons of both methods. Once the main key findings with respect to the different methods are addressed, more general points are listed that highlight the particular findings for the particular fracture network throughout the domains. Throughout the conclusions we do avoid any text related to superiority of manual to automatic trace extraction that is not discussed in detail during the manuscript as advised.

Lines 423 to 425 - Again, examples with identification in figures are needed to support and document this point.

Reply: See comment to lines 419-421: We revised and supplemented this section by the addition of another figure to give more detailed examples and for a better documentation of the narrative.

Comments: Line 21 - As this paper has a focus on methodology, the abstract should briefly present an explanation here as to why automatic assignment of fractures into generations cannot yet be done.

Reply: We agree and added an explanation to the abstract to briefly explain the differences of the methods causing this circumstance.

Lines 31-32 - The fracture geometries are set and not "evolving" so the explanation for the connectivity increase needs to be replaced/improved.

Reply: We revised the sentence to clarify, that each domain has slightly different fracture network characteristics and greater connectivity occurs where the development of later, shorter fractures has been distorted less by the abundance of pre-existing, longer fractures as observed in our data.

Lines 51 to 53 - This sentence should be moved to the end of this section (Line 59), so that this paper's purpose and approach is stated completely first.

Reply: We moved the sentence accordingly.

Line 52 - What is the status of this companion paper (it is not in the reference list)? Can it be cited or does reference need to be made an unpublished source? Or?

Reply: In Passchier et al. (in prep.) we introduce the initial interpretations of the generations on a larger scale of the outcrop, which helped to identify the criteria on which we base our interpretation of the generations in this manuscript. However, the criteria we used and present in this work are self-sustained, explained in detail in section 3.4 and do not require the citation of Passchier et al. (in prep). We now refer to this citation as a companion paper only once and earlier in the manuscript and rewrote the narrative to clarify, that the interpretation of the generations is based on the criteria explained in this manuscript.

Line 84 - Joints and not jointing are unfilled. Reply: We replaced "jointing" with "joints".

Line 104 - "v" - The significance of the radial pattern to the NE is not provided, so this text is superfluous to the later part of this point. If the radial pattern has significance, it should probably be a separate "vi".

Reply: We assigned the radial pattern to an own bullet point, because the results highlight its significant impact on the fracture network connectivity. The position of the pattern we refer to in the domains and the bench is now clarified by referring to our domains NE1 and NE2.

Lines 132 to 140 - Text revised to create a narrative that more contrast imagery results for flight altitudes of 100m or 10m vs. 25m, including the removal of the green highlighted text.

Reply: All suggested revisions were implemented.

Lines 133 to 137 - This highlighted text as written breaks up the narrative flow. It should likely be added to the end of the text on Line 127.

Reply: We moved the text accordingly.

[Figure]

Lines 145 to 148 - This text should be moved up to the end of Line 127 to complete the general description of methodology. Thus, the subsection will finish with the determination of the optimal flight height and the creation of the overview.

Reply: We moved the text accordingly.

Line 154 and elsewhere in the text - Please remember that "this" and "these" are neither pronouns or nouns.

Reply: We reviewed the manuscript and corrected this mistake throughout the whole text.

Line 172 - The intensity belongs to the product, the fractures, and not to the process in this case (clearly, fracturing can be intense, but that is not the intended meaning here, as the description is of a "final fracture population".).

Reply: We agree and replaced "fracturing" with "fracture" for clarity.

Line 201 - Again, can this contribution actually be cited? What is its status? If it does not have an accepted or published status, what can be used as a citation in its place?

Reply: In Passchier et al. (in prep.) we introduce the initial interpretations of the generations on a larger scale of the outcrop, which helped to identify the criteria on which we base our interpretation of the generations in this manuscript. However, the criteria we used and present in this work are self-sustained, explained in detail in section 3.4 and do not require the citation of Passchier et al. (in prep). We now refer to this citation as a companion paper only once and earlier in the manuscript and rewrote the narrative to clarify, that the interpretation of the generations is based on the criteria explained in this manuscript.

Line 214 - Again, can this contribution actually be cited? What is its status? If it does not have an accepted or published status, what can be used as a citation in its place?

Reply: In Passchier et al. (in prep.) we introduce the initial interpretations of the gener-

ations on a larger scale of the outcrop, which helped to identify the criteria on which we base our interpretation of the generations in this manuscript. However, the criteria we used and present in this work are self-sustained, explained in detail in section 3.4 and do not require the citation of Passchier et al. (in prep). We now refer to this citation as a companion paper only once and earlier in the manuscript and rewrote the narrative to clarify, that the interpretation of the generations is based on the criteria explained in this manuscript.

Line 222, Section 4.1 - This section would benefit from a revised title and the addition of two subsection headings to facilitate reader understanding: (1) The entire section is about P21 so change the title to be explicit about trace intensities rather than just traces. (2) 4.1.1 Intensity comparison between two methods (3) 4.1.2 Characterization of intensities for automated data - these two subsection titles clearly separate the purposes of these two portions of the text.

Reply: We agree that the section required a revision and rewrote the text to more clearly compare both methods and took special care to avoid possible confusion of which method we are referring to respectively. Since we also added an analysis using P20 the manuscript, the new sections were titled as follows: 4.1 "Fracture trace segments", in which we discuss general statistics of the segments created by both methods along with a new figure as suggested by the other reviewer. The new subsections are titled 4.1.1 "Fracture trace segment densities", which compares P20 for both methods, 4.1.2 "Fracture trace segment intensities", which compares P21 for both methods was revised to better explain P21 as suggested in an earlier comment. Previously also included in 4.1 were the results of the network topology analysis of the automatic traces. They are now separated more clearly from the previous comparisons in a new section 4.2, that precedes the manual interpretation and network analyses.

Line 224 - Not everyone knows P21, so here is a good location to simply and explicitly define the term. Note: P21 tends to be our best available measure of fracture abundance for natural rock outcrops and does have wide usage, but it is not universally

known.

Reply: We revised the complete section (see replies to earlier comments) and now properly introduce the Pij system, in particular P21 and P20 to the readers along with more detailed explanations of why and how we use both to compare the segments of both tracing/extraction methods.

Lines 227 to 231 - If this text is meant to explain why the P21 intensity for the automatically collected data as compared to the manually collected data, it does not achieve that outcome. What is this text attempting to say? It is not clear?

Reply: This text was supposed to explain that the automatic trace extraction is expected to result in a larger number of segments, because fracture traces are segmented at intersections, while a manual interpreter traces complete fractures from tip to tip. We revised the sentences for claritiy and moved them out of the P21 section into the section providing more general differences of the networks resulting from the different methods to avoid confusion (also due to the restructuring of section 4.1). More clear comparisons of P21 are now provided in the new section, along with a new figure (previous supplement) to further help the narrative and better compare both methods of fracture trace mapping, also with an extended table 1.

Lines 231 to 233 - This sentence is not about the P21 comparison between manual and automate, but rather a description of the differences in P21 for the automated data set. It should be the first sentence of the next paragraph, which is about spatial variations in P21 for the automated data sets.

Reply: We have revised and restructured section 4.1 according to an earlier comment. This sentence belongs to the paragraph dealing with the P21 analyses, which are primarily used to compare both methods. The sentence commented on is separated out at the end of the paragraph to avoid confusion with the comparisons between the methods, because it describes differences in domains and not between methods. It now serves as connection to the following section which discusses the spatial variation

within the domains.

Lines 239 to 241 - The description of maximum length needs to allow for censoring by the sample window perimeters. The identified maximum lengths cannot be stated to be the actual maximum length of any trace that is sampled in a window because of censoring. Now, it is possible to consider the maximum length of traces that are fully contained in a sample window, but that needs to be stated explicitly.

Reply: We agree, this is correct. We added sentence to clarify that maximum lengths may be censored by the sampling windows. This point is also examined later in the manuscript in the discussion chapter.

Line 254 and Figure "18" - To preserve order of figure citation, Figure 18 should become Figure 8, and Figures 8 to 17 should be renumbered.

Reply: We have added several new figures to the manuscript and updated the numbering accordingly. The figure referred to in this comment is presented earlier as suggested in the revised version of the manuscript.

Line 259 and new Figure 9 (old Figure 8) - This figure illustrates length-weight fracture trace abundance as a function of orientation, so it does not show "fracture strike" in any simple manner. The text is revised in this line to better describe what is being illustrated.

Reply: We agree and implemented the suggested revisions.

Lines 264 to 265 - This sentence is an interpretation of observations before the observations are fully presented, so it is out of place and should be deleted.

Reply: We agree and deleted the sentence.

Lines 265 to 266 - Redundant and unneeded sentence

Reply: We agree and deleted the sentence.

[Figure]

Line 281 - How are these "denser clusters" recognized or statistically defined? A reader must be able to use the criterion/criteria to identify the clusters themselves. If reproducible methodology does not exist, the statement about orientations modes for clusters should be deleted.

Reply: Our statement was based on a qualitative visual interpretation of the plot of fracture length vs orientation. We agree that this is not a good foundation to describe the distribution of gen. 5 and therefore deleted the sentence.

Line 288 and Line 289 - text revised in these two locations to more completely and correctly describe the pattern characteristic of the fractures in the NE1 sample window, and in the NE windows vs. the SW windows, respectively.

Reply: We agree and implemented the suggested revisions.

Line 289 - text revised to more simply and clearly describe the lack of a relative age relationship.

Reply: We agree and implemented the suggested revisions.

Lines 291 to 293 - Sentence revised to more clearly identify and state differences with citation of Table 4 being placed in the parentheses at the end of the sentence to not distract from sentence meaning.

Reply: We agree and implemented the suggested revisions.

Line 301 - Adding the word "evolution" to this sentence is important for clarity and meaning. Throughout the text – "mapping boundary" should be "map boundary"

Reply: We agree and added "Evolution" to the sentence and. We replaced "mapping boundary" with "map boundary" throughout the manuscript.

Lines 369 to 370 - Are these two sentences needed here? Their contents are not used immediately. Their content should be added where it is needed further into the document.

Reply: We agree and deleted the sentences at this part of the manuscript. References to Table 10 and the figure are placed at another position where they better complement the narrative further into the document.

Lined 373 - Cite published work that supports the lack of sampling bias and similarity of results for exposure of this quality for different operators. Solid Earth has a publication about this matter, for example.

Reply: Citations on this topic were already provided earlier in the manuscript. To better back our narrative in the lines commented on, we have also added the citations at this place.

Line 382 - "excessive" requires some more explanation and/or examples to assist reader comprehension

Reply: We added an explanation with examples to assist reader comprehension on the topic in the following line. Even more details on the topic are now discussed in the manuscript supported by an additional figure in chapter 5.2.

Lines 388 to 395 - Not convinced that this discussion of the comparison of P21 between the manually and automatically acquired datasets is particularly useful because it considers number of traces separately from the distribution of tracelengths, which is somewhat arbitrary.

Reply: In the revised version of the manuscript we now present P20 earlier in the manuscript to also consider segment density (number of traces/unit area) and provide a direct comparison of the numbers of traces in the revised Table 1. However, in the box-counting method for P21, the box considers the length of cut segments per unit area independently from the number of segments (graph edges) and the manually traced fractures have been pre-processed earlier to resemble the same data structure as the automatically traces ones. Therefore, we deem this to be a valid comparison. To avoid confusion, this is now clearly stated earlier in the manuscript.

Lines 411 to 413 - An important comparison of difference is made in this sentence, Yet, no evidence or citation from other work is provided to support that this comparison is correct. So, the statement is an unsupported speculation and really needs to be better than that for the purposes of this manuscript.

Reply: The sentence was revised according to the other reviewer and we added citation of a work that compared manual interpretations of fracture networks to back our claim. We infer that the automatic code at this stage represents a good option for creating an initial fracture trace map that only differs from a manual interpretation to a degree that is comparable to the deviation of two manual interpretations of the same fracture network (e.g. Long et al., 2018).

Lines 426 to 438 - This paragraph needs an introductory sentence to establish its purpose and why three different characteristics are being "juxtaposed" in one paragraph. An example sentence is offered.

Reply: We implemented the example sentence as suggested.

Lines 452 to 453 - This one sentence paragraph is not really needed as written. A clause is added to the opening sentence about generation 5 to preserve overall narrative flow.

Reply: We agree and implemented the revisions as suggested.

Line 460 to 461 - Text revised and added, so as to achieve greater clarity with a more complete and needed explanation for readers.

Reply: We agree and implemented the revisions as suggested.

Line 464 to 466 - Text revised to more clearly explain situation and to more simply state examples of possible causes.

Reply: We agree and implemented the revisions as suggested.

Lines 467 to 473 - Recommend the elimination of this text because sinuosity is not a

distinguishing characteristic for this study.

Reply: We agree, the text commented on and the whole aspect of sinuosity was eliminated from the manuscript as suggested in an earlier comment.

Lines 504 to 505 - Interpretation is non-unique and unsupported, so it should not be included in the manuscript.

Reply: We agree that this is indeed an unsupported statement and deleted the interpretation from the manuscript.

Please see the annotated PDFs for the main text and figures for additional detailed comments about the syntax.

Reply: We would like to thank the reviewer William Dunne for the detailed remarks on the syntax provided in the annotated PDF, we have corrected the syntax accordingly to the comments and suggested revisions provided therein.

Captions: Figure 1 – Lines 724-725 states "The study areas on "the bench" that are mapped in detail are marked in 725 red." Yet, the only red in the illustration are the red lines for faults. Is the color incorrectly identified or is something missing from this illustration?

Reply: The figure was revised prior to the version presented in the manuscript and the change of the captions was lost. We corrected the caption to state that the map domains are marked as the yellow (not red) squares in the figure.

Figure 2 - Line 731 - the object in 2a to 2e should be identified in the caption. Also, 2a to 2f should have scale bars even if they are approximate.

Reply: We added a reference to the persons in the images and scale bars for the images on the left.

Figure 4 - Line 737 - Please note suggested addition to the text at the end of the caption.

Reply: We implemented the suggested addition in the caption.

Figure 7 - the labels for the X and Y axes of the three sets of graphs should be larger, so that they are easier to read when viewing the entire figure.

Reply: We enlarged all labels on the axes for a better readability.

Figure 9 (10) caption improved by explicitly identifying that data belong to manually collected set and providing a better description of the rose diagrams.

Reply: We implemented the suggested revisions of the caption.

Figure 10 (11) Caption - Revise to match revised Figure 9 (10) caption. Eliminate the addition sentence because the network cannot be described as being "oriented NW-SE". The pattern for NE2 can be described as having a strong length-weighted orientation mode that trends NW-SE, but even that information is likely not needed in the caption here.

Reply: We revised the caption match the one of the associated figure, the numbering of the figures was updated to match the new order (now figures 14 and 16). In the second part of the caption, we were not referring to the orientation of the actual network, but the map domains themselves. We have revised the caption for clarity.

Figure 11 (12) - the key at the bottom of the figure should use larger dots so that the colors are easy for readers to distinguish. While these solid circles will be much larger than the actual ones in the plot, that is not an issue because assigning the colors easily to the generations for the readers is the goal.

Reply: We agree and enlarged the dots in the key of the figure.

Figure 13 (14) caption - make the change in the first sentence of the caption for Figures 14(15), 15(16), 16(17), and 17(18).

Reply: We made the suggested revisions for all mentioned figures and a remark was added for the NE domains that gen 3 fractures were not identified within these two

domains.

Tables - Orientation data in all tables - These data should be rounded off to the nearest integer. The use of two significant figures to the right of the decimal point is false precision, particularly for the manually traced lines.

Reply: We agree and rounded the orientation data in all tables off to the nearest integer.

Table 1 – Title expanded to explicitly identify that the characteristics stated in the table related to the automatically generated fracture network

Reply: We have added the characteristics of the manual interpretation to the table to provide a more direct comparison. The title of the table was revised according to the comment and the new content.